# Nonlinear receptive fields evoke redundant retinal coding of natural scenes

Dimokratis Karamanlis[1,2,7 ✉], Mohammad H. Khani[1,2,8], Helene M. Schreyer[1,2,8], Sören J. Zapp[1,2], Matthias Mietsch[3,4] & Tim Gollisch[1,2,5,6 ✉]

The role of the vertebrate retina in early vision is generally described by the efficient coding hypothesis[1,2], which predicts that the retina reduces the redundancy inherent in natural scenes[3] by discarding spatiotemporal correlations while preserving stimulus information[4]. It is unclear, however, whether the predicted decorrelation and redundancy reduction in the activity of ganglion cells, the retina's output neurons, hold under gaze shifts, which dominate the dynamics of the natural visual input[5]. We show here that species-specific gaze patterns in natural stimuli can drive correlated spiking responses both in and across distinct types of ganglion cells in marmoset as well as mouse retina. These concerted responses disrupt redundancy reduction to signal fixation periods with locally high spatial contrast. Model-based analyses of ganglion cell responses to natural stimuli show that the observed response correlations follow from nonlinear pooling of ganglion cell inputs. Our results indicate cell-type-specific deviations from efficient coding in retinal processing of natural gaze shifts.

Natural visual scenes contain strong positive stimulus correlations in both space and time[3]. According to the prominent efficient coding hypothesis[1,2], the retina's function is to encode stimulus information without wasting resources on signalling this inherent redundancy of natural scenes. Thus, to reduce the redundancy, the retina should decorrelate its output, the spiking activity of retinal ganglion cells, at least as much as the intrinsic noise in the system permits while retaining stimulus information[4,6]. In addition to this intuitive rationale, the popularity of the efficient coding hypothesis is based on its success in explaining characteristics of the early visual system, including centre-surround receptive fields[4] and the emergence and spatial organization of retinal cell types[7–10].

However, the decorrelation prediction of efficient coding has so far only been tested with stimuli that at most share some statistical similarities with natural scenes[11–14], such as static images, sometimes including object movement. Instead, the natural retinal input is dynamically structured by eye and head movements that rapidly shift the retinal image[5]. Such gaze shifts can induce robust response transients at fixation onset in neurons at the early stages of the visual system[15], thus shaping the encoding of natural scenes. Here we therefore sought to study whether retinal redundancy reduction and decorrelation hold for natural stimuli that include gaze shifts and whether stimulus correlations are efficiently discarded by the retina.

## Redundancy in natural-video responses

We recorded ganglion cell spiking activity from isolated marmoset retinas with multielectrode arrays in response to natural videos generated by shifting photographic images according to natural gaze traces (Fig. 1a). The traces had been measured from head-fixed marmosets viewing natural scenes[16] and contained both saccades and fixational eye movements. From the recordings, we functionally identified the four numerically dominant ganglion cell types of the primate retina, ON and OFF parasol cells, as well as ON and OFF midget cells, by their characteristic response kinetics, receptive-field sizes and the tiling of visual space by receptive fields of a given type (Fig. 1c, Extended Data Fig. 1).

Natural videos generated strong and reliable responses (Fig. 1b), which often displayed considerable correlations for pairs of neighbouring cells of the same type (Fig. 1d). Especially ON parasol cells frequently showed simultaneous firing-rate peaks (Fig. 1d, top). Correspondingly, pairwise correlations for ON parasol cells were nearly as high as the corresponding light-intensity correlation in the stimulus (Fig. 1e), thus showing almost no decorrelation. This included cell pairs with neighbouring receptive fields (typically distances below approximately 300 μm), as well as across larger distances. By contrast, OFF midget cells displayed a high degree of decorrelation (Fig. 1e), with firing events of neighbouring cells often occurring for distinct fixations (Fig. 1d, bottom). Correlations for pairs of OFF parasol and ON midget cells, respectively, lay between these two extremes. OFF parasol cells displayed more decorrelation than their ON counterparts but were more strongly correlated than OFF midget cells. Thus, the expected decorrelation is not seen in all ganglion cell types but ranges from strong decorrelation, as in OFF midget cells, to essentially no decorrelation, as in ON parasol cells.

Substantial positive correlations were also found for pairs of parasol cells with opposing contrast preference—that is, an ON cell and an

[1]University Medical Center Göttingen, Department of Ophthalmology, Göttingen, Germany. [2]Bernstein Center for Computational Neuroscience, Göttingen, Germany. [3]German Primate Center, Laboratory Animal Science Unit, Göttingen, Germany. [4]German Center for Cardiovascular Research, Partner Site Göttingen, Göttingen, Germany. [5]Cluster of Excellence "Multiscale Bioimaging: from Molecular Machines to Networks of Excitable Cells" (MBExC), University of Göttingen, Göttingen, Germany. [6]Else Kröner Fresenius Center for Optogenetic Therapies, University Medical Center Göttingen, Göttingen, Germany. [7]Present address: University of Geneva, Department of Basic Neurosciences, Geneva, Switzerland. [8]Present address: Institute of Molecular and Clinical Ophthalmology Basel, Basel, Switzerland. ✉e-mail: dimokaramanlis@gmail.com; tim.gollisch@med.uni-goettingen.de

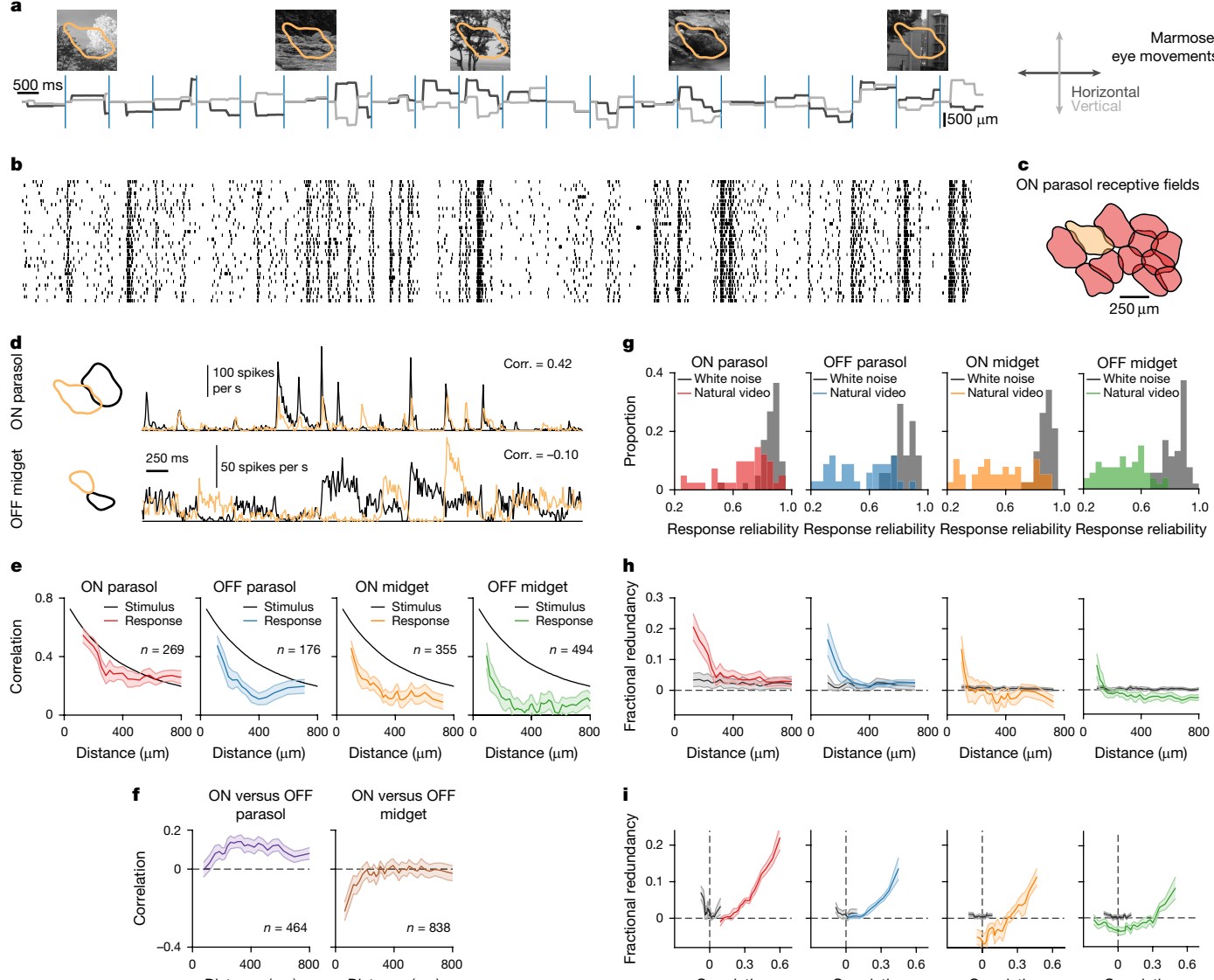

**Fig. 1 | Correlations and redundancy in primate ganglion cell responses to natural videos. a**, Marmoset-specific videos, generated by shifting natural images according to gaze traces recorded from head-fixed marmosets. Each image was presented for 1 s as marked by blue lines. The receptive field of a sample ON parasol ganglion cell is overlaid on the sample images shown. **b**, Spike raster of the sample ON parasol cell for 30 trials in response to the stimulus in **a**. **c**, Receptive-field mosaic of simultaneously recorded ON parasol cells from the peripheral marmoset retina, with the sample cell highlighted. **d**, Receptive fields and firing-rate profiles of two neighbouring ON parasol cells (top) and two neighbouring OFF midget cells (bottom) with resulting activity correlation coefficients (corr.). **e**, Correlation coefficients for ganglion cell pairs under the natural video as a function of receptive-field distance (number of pairs specified

in the figure). For reference, black lines show the correlation between stimulus pixels. **f**, Same as **e**, but for pairs of ON and OFF cells. **g**, Histograms of response reliability under natural videos and white noise, measured by the coefficient of determination between firing rates of even and odd repeats. Number of cells: $n = 41/34/38/53$ for ON parasol/OFF parasol/ON midget/OFF midget. Note that more trials for firing-rate evaluation were available under white noise, contributing to higher reliability values as compared to natural videos. **h**, Fractional redundancy as a function of receptive-field distance for cell pairs of the same type responding to the natural video (coloured traces) or white noise (grey). **i**, Relationship between correlation and fractional redundancy. For **e**,**f**,**h**,**i**, lines represent binned averages for pairs at similar $x$ coordinates (with 95% confidence intervals) for simultaneously recorded cell pairs, and data are from three retinas.

OFF cell (Fig. 1f)—indicating a common pattern in the co-activation of parasol cells within and across types. Pairs of ON and OFF midget cells, on the other hand, showed essentially no correlation or, at small distances, negative correlation, as should be expected when one cell responds to increases and the other to decreases in light intensity.

For the case of noiseless transmission channels, decorrelation is a direct prediction of the efficient coding hypothesis[2,6], as any statistical dependencies between the system's output components reduce the entropy of the joint output patterns and thus prevent the system from using its full coding capacity. In the presence of intrinsic noise at the system's input stage or during processing, a certain level of correlations

in the output may help preserve information in accordance with efficient coding[4,6], as correlated activity allows averaging to increase the signal-to-noise ratio. However, under the present stimulation conditions in the photopic range with natural contrast values well above detection threshold, the retina can be assumed in a low-noise regime, as evident in the reliable spiking responses (Fig. 1b) with Fano factors generally below unity over individual fixation periods (Extended Data Fig. 2e). Thus, averaging over cells offers minimal benefit for efficient information transmission.

To further check whether the observed cell-type-specific correlations could be consistent with compensating for noise, we measured

the response reliability for repeated presentations of the same video sequence by the coefficient of determination between firing-rate profiles for even versus odd stimulus repeats. We found that responses of parasol cells were at least as reliable as responses of midget cells. In particular, ON parasol cells generally displayed the highest level of reliability under natural stimulation among the four analysed types (Fig. 1g). Thus, the most strongly correlated cell type was also the most reliable one, which is inconsistent with the idea that correlations in parasol cells would arise to counteract noise in the signals that they encode.

In principle, correlations could also contribute to efficient stimulus encoding in the form of stimulus-independent so-called noise correlations[17,18]. However, noise correlations were small in our data and typically negligible compared to stimulus-induced correlations (Extended Data Fig. 2a–c), thus indicating that correlations are not part of a synergistic encoding scheme, but imply redundancy[17]. Note also that we here measured correlations in trial-averaged firing rates and thereby obtain a measure that is largely independent of noise correlations.

The deficiency in decorrelation in parasol cells indicates that their representation of natural scenes contains considerable redundancy. To directly assess whether this is indeed the case, we evaluated the fractional redundancy[12,19] of a given cell pair by quantifying the stimulus information provided by the joint responses of the pair and relating it to the single-cell information obtained from its constituent cells (Extended Data Fig. 2d). The fractional redundancy is zero if cells contribute independent information and takes positive values if the information carried by a cell pair falls below the sum of the single-cell information values, up to a maximum of unity if one of the cells adds no new information. Indeed, we found that fractional redundancy values could be substantial for our data, in particular for ON, but also for OFF parasol cells at short distances, indicating that more than 20% of single-cell information could be redundant (Fig. 1h). By contrast, OFF midget cell pairs displayed much less and often no redundancy. Moreover, for each cell type, the fractional redundancy was tightly connected to the measured correlation values (Fig. 1i), confirming response correlations as a source of redundancy. We also found that the spatiotemporal structure of natural stimuli is essential for the high redundancy values. Under a repeated presentation of spatiotemporal white noise, all four ganglion cell types had consistently low redundancy (Fig. 1h), which was also reflected in low cell-pair correlation values (Fig. 1i).

## Spatial contrast triggers correlations

The idea of retinal decorrelation is typically associated with the centre-surround receptive fields of ganglion cells[4]. Yet these considerations generally assume a linear receptive field that acts as a spatial stimulus filter, whereas ganglion cells often display nonlinear processing in the receptive field, which can lead to cell-type-specific sensitivity to spatial contrast on spatial scales below the receptive-field size[20]. Such spatial contrast, which is high when edges or textures are present in natural scenes, can particularly drive parasol cell responses in the macaque retina[21]. We therefore sought to identify whether sensitivity to spatial contrast directly influenced the pairwise response correlations.

In ON parasol cells, fixations with high spatial contrast led to stronger responses that were also more correlated than for fixations with comparable light intensity but low spatial contrast (Fig. 2a,b). These effects also existed in OFF parasol and ON midget cells, albeit to a lesser degree, but not in OFF midget cells (Fig. 2b). For pairs of ON and OFF parasol cells, we also observed stronger correlations for high-spatial-contrast fixations, but not for pairs of ON and OFF midget cells (Fig. 2c,d). Thus, spatial stimulus structure can promote response correlations for certain cell pairs (Fig. 2e, top). This seems to be mediated by nonlinear processing in ganglion cell receptive fields, as the same analysis with predictions of linear–nonlinear (LN) models, fitted to the ganglion cells and capturing the encoding properties of their linear receptive

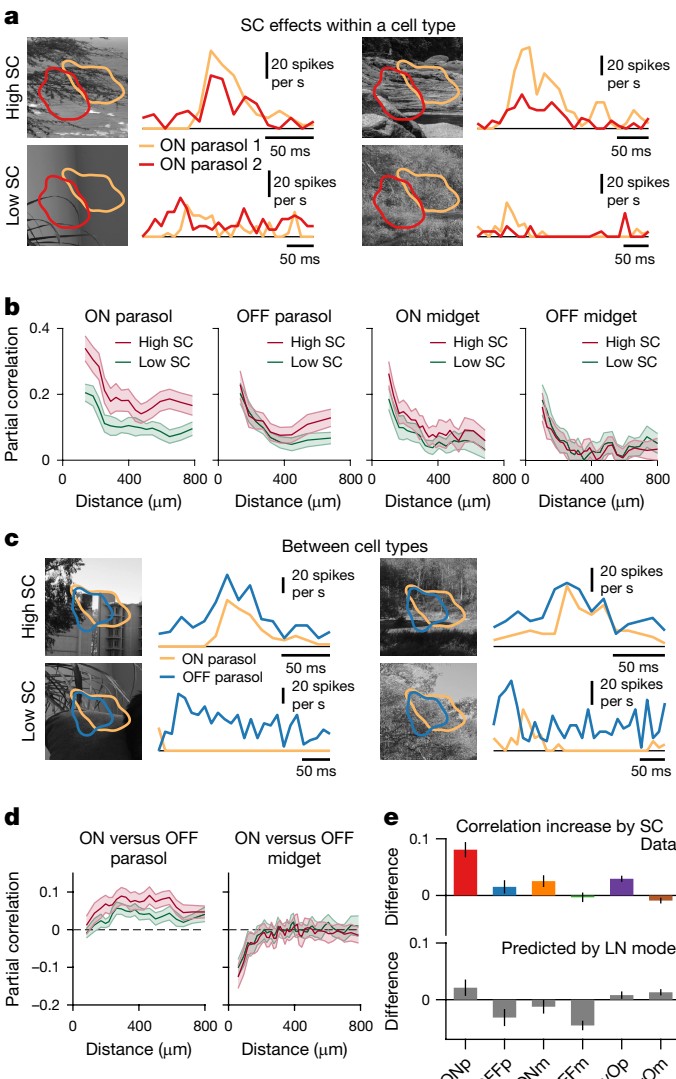

**Fig. 2 | Spatial contrast in natural videos leads to concerted responses within and across ganglion cell types of primate retina. a**, Responses of two neighbouring ON parasol cells to fixations with similar light intensity but either high (top) or low spatial contrast (SC) (bottom). **b**, Partial correlations for each cell type (mean ± 95% confidence interval), separating the pairwise correlations into contributions from fixation periods with high versus low spatial contrast. **c**, Same as **a** for a pair of ON and OFF parasol cells. **d**, Same as **b**, but between types of different response polarity. **e**, Median differences between high- and low-spatial-contrast partial correlations across types (top) and predicted differences calculated with fitted LN models (bottom). Increases in correlation due to spatial contrast were statistically significant (one-sided Wilcoxon sign-rank test) for ON parasol (ONp, $P = 3.5 \times 10^{-27}$), OFF parasol (OFFp, $P = 8.1 \times 10^{-5}$), ON midget (ONm, $P = 7.4 \times 10^{-9}$) and ON versus OFF parasol (OvOp, $P = 4.8 \times 10^{-20}$) cell pairs, but not for OFF midget (OFFm, $P = 0.93$) or ON versus OFF midget (OvOm, $P = 0.99$; here, correlations even decreased slightly but significantly) cell pairs (cell pair numbers shown in Fig. 1e,f). Error bars are median ± robust confidence interval (95%), and data are from three retinas.

fields, failed to reproduce the spatial-contrast-dependent correlation differences observed across cell types (Fig. 2e, bottom).

## Comparison of marmoset and mouse retina

To assess whether spatial-contrast-dependent correlations generalize across species, we also recorded from ganglion cells in the isolated mouse retina (Extended Data Figs. 3–5), to which we presented

natural videos generated by pairing horizontal gaze traces recorded from freely moving mice[22] with natural images from a standard database. We functionally identified (Extended Data Fig. 3) the four types of alpha ganglion cells, which are among the most accessible and widely studied mouse ganglion cell types[23–25]. These cells can be identified by their characteristic visual response properties (Extended Data Fig. 4). Moreover, transient and sustained alpha cell types seem to be orthologs of the primate parasol and midget cell types, respectively, as indicated by transcriptome analysis[26]. Our analysis of correlations and redundancy showed striking similarities between marmoset and mouse ganglion cells (Extended Data Fig. 5). In particular, we found substantial pairwise correlations under the natural video for certain types, but not others. Sustained-OFFα cells were strongly decorrelated, whereas the other three cell types displayed sizeable pairwise correlations. Positive correlations also occurred across ON and OFF types for transient alpha cells.

The pairwise correlations were tightly linked to redundancy in the retinal output, and the high response reliability (Extended Data Fig. 5b) with Fano factors mostly below unity (Extended Data Fig. 3g) again indicated a low-noise regime. As in the marmoset retina, the cells with the highest correlation and redundancy (Extended Data Fig. 5d,f), here transient-OFFα cells, displayed much more reliable responses than the most decorrelating ones (Extended Data Fig. 3h), here sustained-OFFα cells. Moreover, stronger correlations were generally associated with higher spatial contrast, in particular at short retinal distances, and this spatial-contrast-dependence of pairwise correlations was not well captured by LN models (Extended Data Fig. 5h–j). Thus, the cell-type-dependent deficiency in redundancy reduction during natural videos and the correlation-boosting characteristics of high-spatial-contrast fixations seem to be general phenomena across species.

## Subunit models for natural scenes

To investigate how spatial contrast influences response correlations, we aimed at capturing the spatial-contrast sensitivity of the cells under natural stimuli in a computational model. We used a subunit model, which partitions the receptive field of a ganglion cell into smaller subunits whose outputs are nonlinearly summed. The subunits are thought to correspond to bipolar cells that provide excitatory input to ganglion cells[27–29]. To overcome challenges of previous approaches for fitting subunit models to experimental data[14,28,30], such as the reliance on white-noise stimulation, we developed a new parameterized subunit model, which we call the subunit grid model (Fig. 3b). The model contains a set of identical subunits with centre-surround receptive fields and semiregular spacing for each ganglion cell. It can be efficiently fitted to responses obtained under flashed sinusoidal gratings of varying orientation and spatial frequency (Extended Data Fig. 6), which are potent stimuli for driving ganglion cells.

The obtained subunit models showed component differences between ganglion cell types (Fig. 3c,d). For example, subunit nonlinearities of ON parasol cells had particularly high thresholds, showing stronger rectification than for OFF parasol cells, the opposite of what was expected from findings in the macaque retina[21]. Midget cells generally also showed substantial rectification, consistent with findings in the peripheral macaque retina[31,32], but OFF midget cells additionally displayed a more linear regime around the origin of the subunit nonlinearity. Obtained subunit diameters for OFF parasol cells (around 30–40 μm; Extended Data Fig. 6i) and OFF midget cells (around 20 μm) roughly matched data on dendritic field sizes of the putative presynaptic bipolar cells in the peripheral marmoset retina[33] (around 30 μm for type 3 diffuse bipolar cells and 15–20 μm for flat midget bipolar cells, respectively). For ON midget cells, on the other hand, subunits were often surprisingly large (around 50 μm), indicating that they do not represent individual bipolar cell inputs,

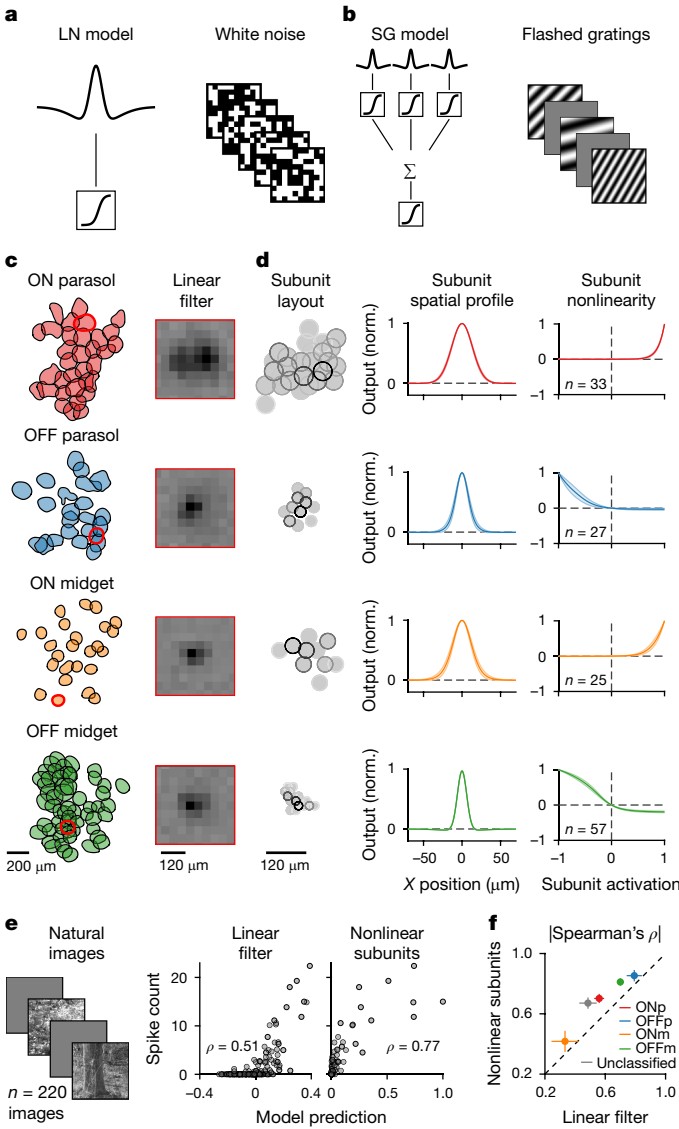

**Fig. 3 | The subunit grid model captures the nonlinear receptive field and responses to natural images. a**, Left, schematic LN model, depicting a centre-surround spatial filter (top, here shown in one spatial dimension) and a subsequent nonlinear transformation (bottom). Right, sample stimulus frames of the spatiotemporal white noise used for fitting the LN model. **b**, Left, schematic subunit grid (SG) model, which sums nonlinearly transformed signals from several identical centre-surround subunits. Right, sample stimulus frames of the flashed gratings, used for fitting the SG model. **c**, Receptive-field mosaics of different ganglion cell types (left), with highlighted sample cells (red outlines) and their linear spatial filters (right). Darker pixels in spatial filters denote larger (positive) values. **d**, Obtained subunit layouts for the sample cells (left), spatial profiles of subunits (middle) and subunit nonlinearities (right). For the subunit maps, each circle corresponds to the $2\sigma$ Gaussian contour of the subunit centre, with the saturation of the outline denoting the subunit weight. Spatial profiles and nonlinearities are shown as mean and 95% confidence interval across all cells of the same type. **e**, Sample natural images that were flashed onto the retina (left). Output from the linear filter of the LN model (middle) and from the summed nonlinear subunits of the SG models (right) plotted against natural image responses for a single cell. $\rho$ denotes the Spearman correlation. **f**, Comparison of model performance (measured by the absolute value of the Spearman correlation as in (**e**)) for ON parasol (ONp, $n = 63$), ON midget (ONm, $n = 31$), OFF parasol (ONp, $n = 67$) and OFF midget (OFFm, $n = 79$) cells from three retinas. Grey dot marks all cells that were unclassified ($n = 193$) but reliable. Error bars are median ± robust confidence interval (95%). Norm., normalized.

potentially because subunit size was not well constrained by the data for these cells. Alternatively, this could reflect several midget bipolar cells combining to form individual subunits[30,34] or signals from type-6 diffuse bipolar cells, which also provide input to ON midget cells[35] and which have dendritic diameters of 40–80 μm in the marmoset retina[33]. Cell-type-specific differences in nonlinear components were also prominent in the mouse retina (Extended Data Fig. 7). Besides rectification in most cell types, we observed prominent saturation of subunit signals in transient-OFFα cells. This subunit nonlinearity, found particularly for dorsal transient-OFFα cells (Extended Data Fig. 8), is consistent with increased sensitivity to spatial homogeneity[36]. For most cell types, subunit models captured responses to flashed natural images better than linear receptive fields for both marmoset (Fig. 3e,f) and mouse retina (Extended Data Fig. 7). Thus, cell-type-specific models of the nonlinear receptive field can reflect retinal processing of spatial contrast in naturalistic stimuli, indicating that the different nonlinear characteristics may help explain differences in spatial-contrast-driven correlations between ganglion cell types.

To extend the analysis to dynamic stimuli, we added temporal filters to both the centre and surround of the subunits and fitted spatiotemporal subunit grid models to ganglion cell responses under sinusoidal gratings flickering in rapid succession (Fig. 4a). The obtained spatiotemporal models captured natural video responses for different types of ganglion cells, in both marmoset and mouse (Fig. 4b,c). Subunit grid models improved over simple LN models for most cell types, except for mouse sustained- and transient-OFFα cells, by reproducing additional response peaks. Response predictions of the subunit grid model also outperformed those of alternative subunit identification schemes, such as spike-triggered non-negative matrix factorization[28] and spike-triggered clustering[30] (Extended Data Figs. 9 and 10).

Moreover, the obtained models reproduced the measured response correlations well. In particular, cell-type-specific correlations predicted by subunit grid models were much closer to the data than for LN models (Fig. 4e), which tended to overestimate these correlations as previously reported[11]. The lower predicted response correlations by the subunit grid model might seem counterintuitive: subunits confer sensitivity to spatial contrast, which, as we have seen, boosts correlations in the data (Fig. 2). However, this discrepancy can be explained by the inclusion of surround suppression in the subunit grid model through the subunit surround. LN models, particularly when fitted to white-noise stimuli, may underestimate the strength of the receptive-field surround[37].

## Nonlinearities drive correlations

To distinguish the effects of surround suppression and spatial-contrast sensitivity conveyed by the subunits, we compared the subunit grid models to difference-of-Gaussians (DoG) LN models fitted directly to the flickering grating responses, the same stimulus used for the subunit grid. DoG LN models showed better response predictions than white-noise-fitted LN models but were still outperformed by subunit grid models for certain cell types (Fig. 4d). Pairwise correlations estimated by DoG LN models matched those of the subunit grid models (Fig. 4e), confirming that surround suppression is essential for reducing the overestimation of correlations by the standard LN model and capturing the correct range of response correlations. However, the DoG LN model lacks the required spatial nonlinearities, and we therefore used the subunit grid model to investigate the observed dependence of response correlations on spatial stimulus structure.

To investigate how the nonlinear receptive field contributes to the response correlations, we separated the fixations for each cell pair according to how important nonlinear spatial processing was for determining the cells' responses (Fig. 5a). Specifically, we tagged those fixations as nonlinear for which the predictions of the (spatially nonlinear) subunit grid model differed most from the predictions of the (spatially linear) DoG LN model. For these 'maximally differentiating fixations',

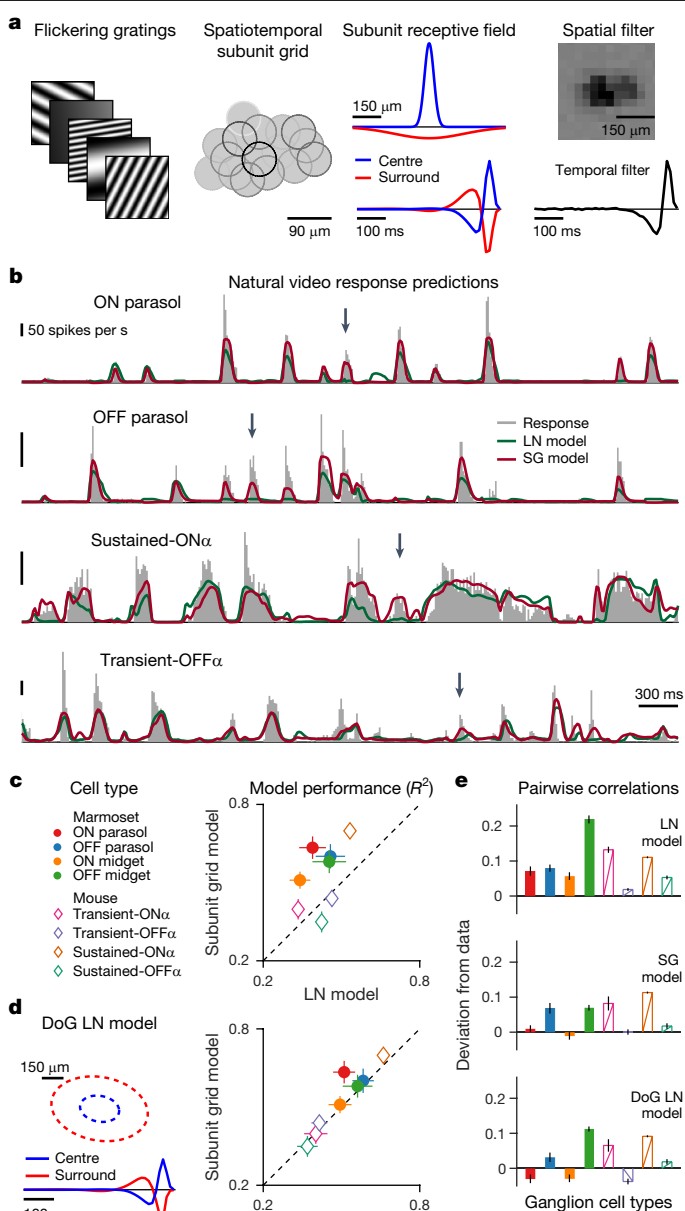

**Fig. 4 | The subunit grid model captures the retinal output under natural videos. a**, Left, schematic of the flickering-gratings stimulus. Middle, subunit layout for a single ON parasol cell, shown as in Fig. 3d, and corresponding centre and surround spatial and temporal components. Right, LN model components determined with white-noise stimulation, matching roughly the union of subunit centres and their temporal profile. **b**, LN and SG model predictions (coloured traces) of ganglion cell responses (grey histograms) to natural videos for sample cells of different types in marmoset and mouse retina. Arrows mark response peaks captured by the SG model but not by the LN model. **c**, Median model performance for all identified types in marmoset and mouse. **d**, Left, schematic depiction of DoG LN model, showing elliptical outlines of centre and surround and corresponding temporal components. Right, comparison of median model performance between DoG LN and SG models for different cell types. **e**, Deviations of pairwise correlation coefficients for model predictions from the actual correlations in the data. For **c**–**e**, error bars are median ± 95% robust confidence interval and number of cells (and corresponding cell pairs) are ON parasol $n = 41$ (269), OFF parasol $n = 34$ (176), ON midget $n = 37$ (330), OFF midget $n = 53$ (494), transient-ONα $n = 18$ (57), transient-OFFα $n = 43$ (315), sustained-ONα $n = 88$ (1923), sustained-OFFα $n = 76$ (889) from three marmoset and four mouse retina pieces (eight pieces for LN model evaluation with $n$ values given in Extended Data Fig. 5).

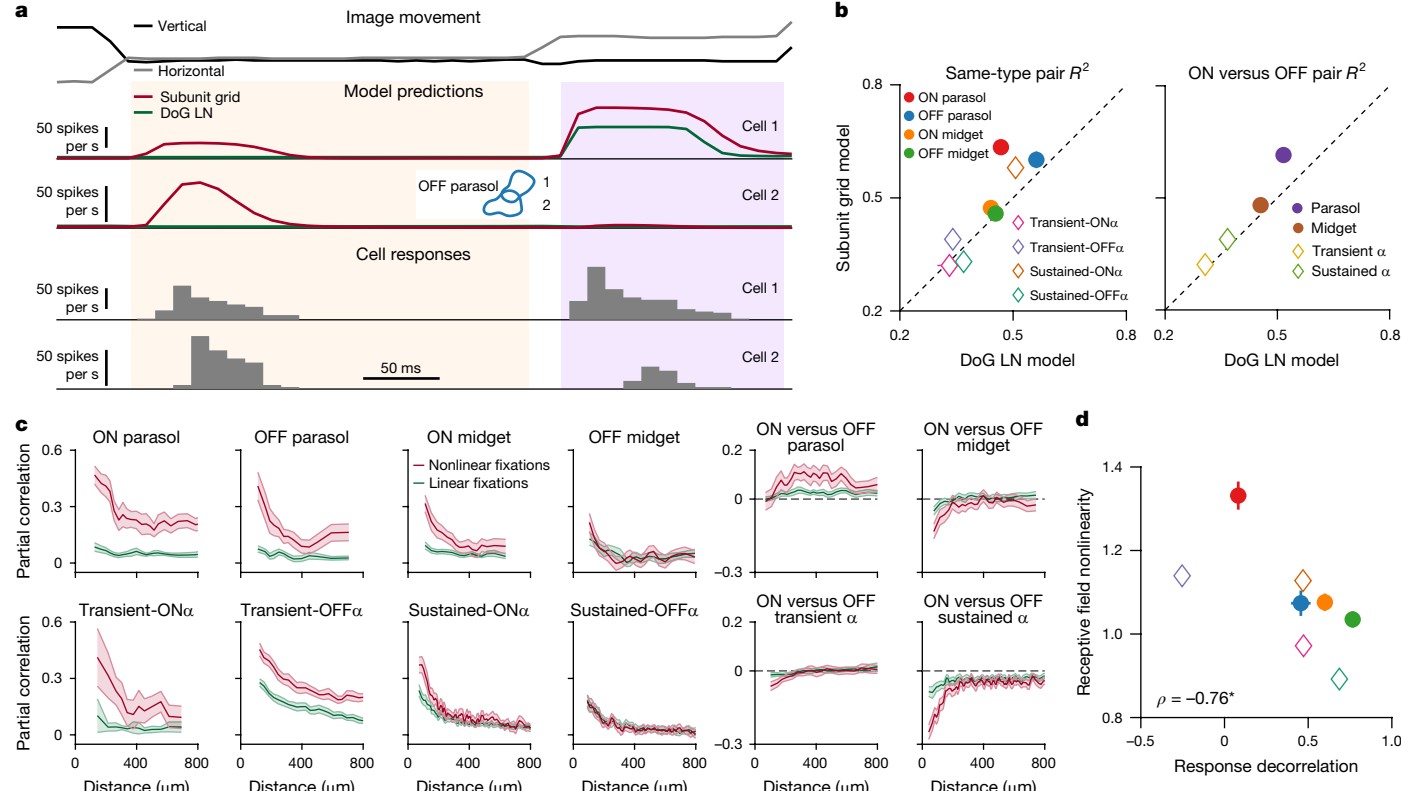

**Fig. 5 | Correlated activity results from fixations that evoke nonlinear responses. a**, Top, sample image trajectory for the presentation of a single image. Middle, corresponding model predictions for two neighbouring OFF parasol cells (receptive fields in inset) for the DoG LN and the subunit grid models. Shaded areas mark two consecutive fixations, the first classified as differentiating (the two model predictions diverge) and the second as non-differentiating (model predictions align). Bottom, responses of the two cells during the same period. **b**, Comparison of average model performances ($R^2$) for the subunit grid model and the DoG LN model during the top 20% maximally differentiating fixations. **c**, Contributions of linear and nonlinear fixations to the total pairwise correlations for the natural video (mean with 95% confidence intervals). **d**, Relationship between receptive-field nonlinearity, calculated as

the ratio of subunit grid over DoG LN model performance during maximally differentiating fixations, and overall pairwise response decorrelation, calculated as the difference between stimulus and response correlations relative to the stimulus correlations. For cells with zero DoG LN performance, the ratio was set to the maximum value measured across cells of the same type. * denotes significant Spearman correlation ($p = 0.037$, two-sided permutation test). For **b** and **d**, error bars are median ± 95% robust confidence interval and number of cell pairs are ON parasol $n = 269$, OFF parasol $n = 176$, ON midget $n = 355$, OFF midget $n = 494$, transient-ONα $n = 63$, transient-OFFα $n = 315$, sustained-ONα $n = 2040$, sustained-OFFα $n = 889$ from three marmoset and four mouse retina pieces (eight pieces for response decorrelation with $n$ values given in Extended Data Fig. 5).

the subunit grid model displayed superior model predictions compared to the DoG LN models for certain cell types, such as ON and OFF parasol cells in the marmoset and sustained-ONα cells in the mouse (Fig. 5b), for which capturing receptive-field nonlinearities thus mattered most. These cell types had also shown strong spatial-contrast dependence of the pairwise response correlation (Fig. 2b and Extended Data Fig. 5i).

Thus, to determine whether the nonlinear receptive field alone might explain the difference in contributions of high- and low-contrast stimulus segments to the response correlations, we split the set of all fixations into those in which predictions of a linear and a nonlinear receptive field matched best ('linear fixations') and those in which the two predictions most diverged ('nonlinear fixations'). We then assessed the relative importance of the linear and nonlinear fixations for the correlated spiking activity by calculating the contribution of each subset of fixations to the total correlation. Indeed, for parasol cells in the marmoset and nonlinear alpha cells in the mouse, nonlinear fixations contributed the most to the overall pairwise correlations, as indicated by the larger partial correlations for this set of fixations (Fig. 5c), whereas partial correlations for linear fixations were much more similar between cell types. Moreover, it is the nonlinear fixations that were responsible for the positive correlations of ON and OFF parasol cells (Fig. 5c). By contrast, linear cells, such as marmoset OFF midget cells and mouse sustained-OFFα cells, displayed more balanced

partial correlations between linear and nonlinear fixations, indicating that responses during both sets of fixations contributed equally to the total correlation for these cells. We therefore conclude that fixations containing salient spatial structure, which drives particularly the nonlinear components of receptive fields, elicit concerted responses for specific types of retinal ganglion cells. Thus, across both marmoset and mouse, ganglion cells with stronger receptive-field nonlinearities tend to perform less stimulus decorrelation during stimulation with natural gaze dynamics (Fig. 5d).

## Discussion

We provide direct evidence that redundancy reduction in the retina is violated in a cell-type-specific manner under natural stimuli that include gaze dynamics. Although some ganglion cell types displayed substantial decorrelation and redundancy reduction, others showed highly correlated activity. The correlations led to redundant representations and were particularly pronounced when the stimulus shifted to a new fixation that contained high spatial contrast. This concerted activity originated in nonlinear processing in the receptive fields of retinal ganglion cells, a processing feature that has been absent in many considerations of efficient coding and redundancy reduction in the retina[4,7,9,19]. Under the global changes in spatial stimulus patterns

induced by gaze shifts, nonlinear receptive fields become simultaneously activated in a way that is not effectively suppressed by surround mechanisms. This co-activation occurs for a range of distances, as well as across preferred contrast polarity, thus even creating seemingly paradoxical positive correlations between ON and OFF cells.

The correlated activity of parasol ganglion cells in the primate retina and transient alpha cells in mouse challenges the efficient coding hypothesis. Although complete decorrelation is predicted by efficient coding only when there is no noise before the output stage of the system[4], it seems unlikely that the observed high correlations directly counteract noise for efficient signal transmission. First, stimulation with temporal dynamics from natural gaze shifts drives responses with high reliability and signal-to-noise ratio. Second, the most correlated cell types show particularly reliable responses compared to the least correlated ones. Thus, a cell-type-specific role of correlations for counteracting noise is not supported. It remains possible, however, that correlations could support coding efficiency for large populations of several ganglion cell types. For example, correlations between ON parasol cells might counteract noise in midget cells for a joint efficient stimulus encoding, in particular because noise may be shared between parasol and midget cells[38,39]. However, it seems unclear whether their distinct downstream pathways and their differences in conduction velocity[40] may support a joint coding scheme of parasol and midget cells.

The analysis of decorrelation and redundancy does not hinge on the specific stimulus aspects represented by the cells' activity or whether cells can be described by a linear receptive field. If the task were, for example, to encode high-frequency spatial contrast without redundancy, lateral inhibition that is as sensitive to spatial contrast as centre excitation could decorrelate responses even with nonlinear receptive fields. However, ganglion cell receptive-field surrounds may differ substantially from the centre in their spatial nonlinearities[31,41]. To further investigate the relationship between spatial nonlinearities and redundancy reduction, it would thus be interesting to analyse how spatially nonlinear ganglion cell models should be structured to optimize coding efficiency[42].

Earlier investigations of correlated retinal activity had often focused on spontaneous activity or artificial stimuli[38,43–46], such as white noise. Notably, in the context of artificial stimuli, nonlinearities of receptive fields had previously been associated with strengthening decorrelation[14], in contrast to our finding with natural stimuli. Studies of salamander retina with natural stimuli had also observed considerable correlations[12,47], although not connected to spatial nonlinearities or gaze dynamics. Also on the basis of salamander retina, fixational eye movements had been proposed to contribute to decorrelation[48], yet our data show strong correlations for stimuli that contained measured fixational eye movements.

Correlated retinal activity has been indicated to play a role in increasing spatial resolution[43] and error correction[47]. Because retinal circuit nonlinearities have been associated with computations underlying visual feature detection[49,50], we hypothesize that the response correlations in nonlinear cell types aid in signalling the detection of a relevant visual feature in natural scenes. For example, we found that mammalian direction-selective (DS) retinal ganglion cells, which are a prime example of feature detectors, have strongly nonlinear receptive fields, and their strong pronounced pairwise response correlations could even exceed stimulus correlations (Extended Data Fig. 11). Correlations may be particularly important for tagging the relevant feature, such as local spatial contrast or the preferred motion signal, and distinguishing it from changes in illumination of the receptive field. Although a single neuron's firing rate might be confounded by light intensity or other stimulus dimensions to which the neuron is sensitive, the feature of interest may be isolated by combining the activity from groups of neurons[51]. Further insight into the functional consequences of correlated activity may come from assessing their dependence on stimulus context, such as average light level. Spatial nonlinearities in ganglion cell receptive fields, for example, may decrease at lower light levels[52], which should result in decreased stimulus-induced correlations, and noise correlations may become more prevalent[18].

Our observations of stronger nonlinearities in marmoset ON than OFF parasol cells differ from previous findings that OFF parasol cells are the more nonlinear ones in the macaque[21,53]. In our data, the stronger nonlinearities of ON parasol cells were observed in the subunit models fitted to flashed gratings, as well as in those fitted to flickering gratings; in the stronger improvements of response predictions for natural stimuli when nonlinearities were included; and in responses to reversing gratings (for example, Extended Data Fig. 9b). It seems feasible that this represents a species difference between macaque and marmoset but could also depend on experimental conditions, such as illumination level or stimulation of the receptive-field surround[21,31,52,54]. Generally, differences in nonlinearities between ON and OFF channels in the retina seem to be species and cell-type dependent[25,55,56] and may reflect differences in visual tasks[57].

Efficient coding is often considered a natural assumption for sensory systems because of the need to preserve energy associated with neuronal activity[58]. However, whether the retinal output is energy-efficient in vivo has been debated[59]. Moreover, feature detection might have different requirements than general information transmission, such as robustness or future prediction[60], which could lead to deviations from efficient coding. In this context, the retinal code may multiplex correlated nonlinear responses containing feature information with decorrelated baseline activity. Our findings indicate that the retinal output can maintain efficiency in various stimulus contexts while being robust for feature detection. Energy constraints could also be addressed by other mechanisms, such as making responses transient, allowing the visual system to detect important features promptly. Thus, the different information channels of the retina may balance energy conservation and robust feature detection on the basis of their respective visual tasks.

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

## Methods

### Tissue preparation and electrophysiology

We recorded spiking activity from retinas of three adult male marmoset monkeys (*Callithrix jacchus*), 12, 13 and 18 years of age, using a single piece of retina from each animal. No previous determination of sample size was used. The retinal tissue was obtained immediately after euthanasia from animals used by other researchers, in accordance with national and institutional guidelines and as approved by the institutional animal care committee of the German Primate Center and by the responsible regional government office (Niedersächsisches Landesamt für Verbraucherschutz und Lebensmittelsicherheit, permit number 33.19-42502-04-20/3458). After enucleation, the eyes were dissected under room light, and the cornea, lens and vitreous humour were carefully removed. The resulting eyecups were then transferred into a light-tight container containing oxygenated (95% $O_2$ and 5% $CO_2$) Ames' medium (Sigma-Aldrich) supplemented with 4 mM D-glucose (Carl Roth) and buffered with 20–22 mM $NaHCO_3$ (Merck Millipore) to maintain a pH of 7.4. The container was gradually heated to 33 °C, and after at least an hour of dark adaptation, the eyecups were dissected into smaller pieces. All retina pieces used in this study came from the peripheral retina (7–10 mm distance to the fovea). The retina was separated from the pigment epithelium just before the start of each recording. All reported marmoset data are from pieces for which a 5% contrast full-field modulation at 4 Hz produced at least a 10 spikes per second modulation in the average ON parasol spike rate. This ensured high quality and light sensitivity of the analysed retina pieces (Extended Data Fig. 12).

We also recorded spiking activity from 12 retina pieces of eight wild-type female mice (C57BL/6J) between 7 and 15 weeks old (except for one 23-week-old mouse). No previous determination of sample size was used. All mice were housed on a 12-hour light/dark cycle. The ambient conditions in the animal housing room were kept at around 21 °C (20–24 °C) temperature and near 50% (45–65%) humidity. Experimental procedures were in accordance with national and institutional guidelines and approved by the institutional animal care committee of the University Medical Center Göttingen, Germany. We cut the globes along the ora serrata and then removed the cornea, lens and vitreous humour. The resulting eyecups were hemisected to allow two separate recordings. On the basis of anatomical landmarks, we performed the cut along the horizontal midline and marked dorsal and ventral eyecups. Before the start of each recording, we isolated retina pieces from the sclera and pigment epithelium.

For both marmoset and mouse retina recordings, we placed retina pieces ganglion-cell-side-down on planar multielectrode arrays (Multichannel Systems; 252 electrodes; 10 or 30 μm electrode diameter, either 60 or 100 μm minimal electrode distance) with the help of a semipermeable dialysis membrane (Spectra Por) stretched across a circular plastic holder (removed before the recording). The arrays were coated with poly-D-lysine (Merck Millipore). For some marmoset recordings, we used 60-electrode perforated arrays[61]. Dissection and mounting were performed under infrared light (using LEDs with peak intensity at 850 nm) on a stereo-microscope equipped with night-vision goggles. Throughout the recordings, retina pieces were continuously superfused with oxygenated Ames' solution flowing at 8–9 ml min$^{-1}$ for the marmoset or 5–6 ml min$^{-1}$ for the mouse retina. The solution was heated to a constant temperature of 33–35 °C through an inline heater in the perfusion line and a heating element below the array.

Extracellular voltage signals were amplified, bandpass filtered between 300 Hz and 5 kHz and digitized at 25 kHz sampling rate. We used Kilosort[62] for spike sorting. To ease manual curation, we implemented a channel-selection step from Kilosort2 by discarding channels that contained only a few threshold crossings. We curated the output of Kilosort through phy, a graphical user interface for visualization and selected only well-separated units with clear refractory periods in the autocorrelograms. In a few cases, we had to merge units with temporally misaligned templates; we aligned the spike times by finding the optimal shift through the cross-correlation of the misaligned templates.

### Visual stimulation

Visual stimuli were sequentially presented to the retina through a gamma-corrected monochromatic white organic LED monitor (eMagin) with 800 × 600 square pixels and 85 Hz (marmoset) or 75 Hz (mouse) refresh rate. The monitor image was projected through a telecentric lens (Edmund Optics) onto the photoreceptor layer, and each pixel's side measured 7.5 μm on the retina or 2.5 μm for some marmoset recordings in which we used a different light-projection setup[61]. All stimuli were presented on a background of low photopic light levels, and their mean intensity was always equal to the background. To estimate isomerization rates of photoreceptors, we measured the output spectrum of the projection monitors and the irradiance at the site of the retina and combined this information with the absorbance profile[63] and peak sensitivities of the opsins (543–563 nm for different marmoset M-cones and 499 nm for marmoset rods[64,65]; 498 nm for mouse rods) and with the collecting areas of photoreceptors, using 0.5 μm$^2$ for mouse rods[66], 0.37 μm$^2$ for marmoset cones and 1.0 μm$^2$ for marmoset rods, applying here the values from macaque cones and rods[67,68]. For the marmoset, the background light intensity resulted in approximately 3,000 photoisomerizations per M-cone per second and approximately 6,000 photoisomerizations per rod per second, and for the mouse, approximately 4,000 photoisomerizations per rod per second. We fine-tuned the focus of stimuli on the photoreceptor layer before the start of each experiment by visual monitoring through a light microscope and by inspection of spiking responses to contrast-reversing gratings with a bar width of 30 μm.

### Receptive-field characterization

To characterize functional response properties of the recorded ganglion cells, we used a spatiotemporal binary white-noise stimulus (100% contrast) consisting of a checkerboard layout with flickering squares, ranging from 15 to 37.5 μm on the side in different recordings. The stimulus update rate ranged from 21.25 to 85 Hz. Each stimulus cycle consisted of a varying training stimulus and a repeated test stimulus, with 18–55 cycles presented in total. The training stimulus duration ranged from 45 to 144 s in different experiments. The test stimulus consisted of a fixed white-noise sequence ranging from 16 s to 18 s, which we used here to determine noise entropies and noise correlations.

We calculated spike-triggered averages (STAs) over a 500 ms time window and extracted spatial and temporal filters for each cell as previously described[69]. In brief, the temporal filter was calculated from the average of spatial STA elements whose absolute peak intensity exceeded 4.5 robust standard deviations of all elements. The robust standard deviation of a sample is defined as 1.4826 times the median absolute deviation of all elements, which aligns with the standard deviation for a normal distribution. The spatial receptive field was obtained by projecting the spatiotemporal STA on the temporal filter. We also calculated spike-train autocorrelation functions under white noise, using a discretization of 0.5 ms. For plotting and subsequent analyses, all autocorrelations were normalized to unit sum.

For each cell, a contour was used to summarize the spatial receptive field. We upsampled the spatial receptive field to single-pixel resolution and then blurred it with a circular Gaussian of $\sigma = 4$ pixels. We extracted receptive-field contours using MATLAB's 'contourc' function at 25% of the maximum value in the blurred filter. In some cases, noisy STAs would cause the contour to contain points that lay further away from the actual spatial receptive field. Thus, we triaged the contour points and removed points that exceeded 20 robust standard deviations of all distances between neighbours of the points that were used to define the contour. This process typically resulted in a single continuous area without holes. The centre of each receptive field was defined as the

median of all contour points, and its area was determined by the area enclosed by the contour.

## Ganglion cell type identification

We used responses to a barcode stimulus[70] to cluster cells into functional types in each single recording. The barcode pattern had a length of 12,750 (or 12,495) μm and was generated by superimposing sinusoids of different spatial frequencies ($f$) with a $1/f$ weighting. The constituent sinusoids had spatial frequencies between 1/12,750 (or 1/12,495) and $1/120$ μm$^{-1}$ (separated by 1/12,750 or 1/12,495 μm$^{-1}$ steps, respectively) and had pseudorandom phases. The final barcode pattern was normalized so that the brightest (and dimmest) values corresponded to 100% (and −100%) Weber contrast from the background. The pattern moved horizontally across the screen at a constant speed of 1,275 (or 1,125) μm s$^{-1}$, and the stimulus was repeated 10–20 times. Obtained spike trains were converted into firing rates using 20 ms time bins and Gaussian smoothing with $\sigma$ = 20 ms. We quantified cell reliability with a symmetrized coefficient of determination ($R^2$), as described previously[36]. We only included cells with a symmetrized $R^2$ value of at least 0.1 and that were not putative DS cells (see below).

We used average responses to the barcode stimulus to generate a pairwise similarity matrix, as described previously[70]. We defined the similarity between each pair of cells as the peak of the cross-correlation function (normalized by the standard deviations of the two signals) between the spike rate profiles of the two cells. To obtain a final similarity matrix, we multiplied the entries of the barcode similarity matrix with the entries of three more similarity matrices, obtained from receptive-field response properties. The first two were generated by computing pairwise correlations between both the temporal filters and the autocorrelation functions of each cell. The third one used receptive-field areas and was defined as the ratio of the minimum of the two areas over their maximum (Jaccard index).

We converted the combined similarity matrix to a distance matrix by subtracting each entry from unity. We then computed a hierarchical cluster tree with MATLAB's 'linkage' function, using the largest distance between cells from two clusters as a measure for cluster distance (complete linkage). The tree was used to generate 20–50 clusters; we chose the number depending on the number of recorded cells. This procedure yielded clusters with uniform temporal components and autocorrelations and with minimally overlapping receptive fields but typically resulted in oversplitting functional ganglion cell types. Thus, we manually merged clusters with at least two cells on the basis of the similarity of properties used for clustering and on receptive-field tiling. To incorporate cells that were left out of the clustering because of the barcode quality criterion, we expanded the clusters obtained after merging. An unclustered cell was assigned to a cluster if its Mahalanobis distance from the centre of the cluster was at most 5 but at least 10 for all other clusters. Our method could consistently identify types with tiling receptive fields forming a mosaic over the recorded area. This was generally the case for parasol and midget cells in the marmoset and the different alpha cells in the mouse retina, which are the ones primarily analysed in this work. Note, though, that mosaics are typically incomplete because of missed cells whose spikes were not picked up by the multielectrode array (or not sufficiently strong to allow reliable spike sorting). As is common with this recording technique, missed cells are more frequent for some cell types than for others, and this recording bias renders, for example, midget cell mosaics less complete than those of parasol cells. The present analyses, however, do not rely on the recovery of complete mosaics, as the pairwise investigations of correlation and redundancy only require sufficient sampling of pairs at different distances.

## Matching cell types to mouse ganglion cell databases

We validated the consistency of cell type classification by examining cell responses to a chirp stimulus[24], which was not used for cell clustering.

Light-intensity values of the chirp stimulus ranged from complete darkness to the maximum brightness of our stimulation screen. The stimulus was presented 10–20 times. For the mouse retina, the parameters of the chirp stimulus matched the original description, which allowed us to compare cell responses to calcium traces in a database of classified retinal ganglion cells[24]. To convert spike rates to calcium signals, we convolved our spiking data with the calcium kernel reported in the original paper. We then computed correlations to the average traces of each cluster in the database.

For some mouse experiments, we also used responses to spot stimuli[23]. In brief, we flashed one-second-long spots over the retina at different locations and with five different spot diameters (100, 240, 480, 960 and 1,200 μm). Between spot presentations, illumination was set to complete darkness, and the spots had an intensity of 100–200 photoisomerizations per rod per second. For each cell, we estimated a response centre by identifying which presented spot location yielded the strongest responses when combining all five spot sizes. We only used cells whose estimated response centre for the spots lay no further than 75 μm from the receptive-field centre as determined with white-noise stimulation. To calculate similarities to the cell types in the available database[23], we concatenated firing-rate responses to the five spot sizes and then calculated correlations with the available database templates. We also applied saccade-like shifted gratings to detect image-recurrence-sensitive cells in mouse retinas as previously described[36,71]. These cells correspond to the transient-OFFα cells in the mouse retina.

## Natural videos, LN model predictions and response correlations

For the marmoset retina, we constructed natural videos in similar fashion as previously done for the macaque retina[30,72]. In brief, the videos consisted of 347 grayscale images that were shown for 1 s each and jittered according to measurements of eye movements obtained from awake, head-fixed marmoset monkeys[16] (graciously provided by J. L. Yates and J. F. Mitchell; personal communication). These eye movement data had been collected at a scale of about 1.6 arcmin per pixel, which roughly corresponds to 2.67 μm on the marmoset retina, using a retinal magnification factor of 100 μm deg$^{-1}$ (ref. 73). To align the sampled traces with the resolution of our projection system, we adjusted the pixel size of the gaze traces to 2.5 μm when presenting the natural video. We furthermore resampled the original 1,000 Hz gaze traces to produce a video with a refresh rate of 85 Hz. The presented natural video consisted of 30–35 cycles of varying training and repeated test stimuli. Test stimuli consisted of 22 distinct natural images, using the original grayscale images (graciously provided by J. L. Yates and J. F. Mitchell; personal communication) viewed by the marmosets during eye movement tracing (mean intensity −10% relative to background; 38% average contrast, calculated as the standard deviation across all pixels for each image). Each test image was paired with a unique movement trajectory given by the marmoset eye movements. For each training stimulus cycle, we presented 40 images out of the 325 remaining images (sampled with replacement), each paired with a unique movement trajectory. These 325 images were obtained from the van Hateren database[74], were multiplicatively scaled to have the same mean intensity as the background and had an average contrast of 45%.

For the mouse retina, we applied a similar procedure. In brief, the videos consisted of the same 325 images from the van Hateren database as used for the marmoset training stimulus, shown for 1 s each and jittered according to the horizontal gaze component[22] of freely moving mice (graciously provided by A. Meyer; personal communication). We resampled the original 60 Hz gaze traces to produce a video with a refresh rate of 75 Hz. For our recordings, the one-dimensional gaze trajectory was randomly assigned to one of four orientations (0, 45, 90 or 135 degrees) for each 1 s image presentation. The amplitude of the original movement was transformed into micrometre on the retina using a retinal magnification factor of 31 μm deg$^{-1}$ for the mouse.

All images were multiplicatively scaled to have the same mean intensity as the background. Test stimuli consisted of 30–35 cycles of 25 distinct natural images, paired with unique movement trajectories. The training stimuli consisted of batches of 35 images out of the remaining 300 (sampled with replacement), each paired with a unique movement trajectory.

For the model-based analyses of the responses to the natural videos, we applied a temporal binning corresponding to the stimulus update frequency (85 Hz for the marmoset and 75 Hz for the mouse) and used the spike count in each bin. To extract firing rates for the test stimuli, we averaged the binned responses over repeats. Furthermore, to eliminate cells with noisy responses, we only used cells for subsequent analyses with a symmetrized $R^2$ of at least 0.2 between even and odd trials of the test set.

All model predictions for natural videos used the stimulus training part for estimating an output nonlinearity and the test part for evaluation of model performance. For the LN model, we obtained the spatiotemporal stimulus filter (decomposed into a spatial and a temporal filter as explained in 'Receptive-field characterization') from the spatiotemporal white-noise experiments but estimated the nonlinearity from the natural-video data. To do so, we projected the video frames onto the upsampled spatial filter (to single-pixel resolution) and then convolved the result with the temporal filter. The output nonlinearity was obtained as a histogram (40 bins containing the same number of data points across the range of filtered video-stimulus signals) containing the average filtered signal and the average corresponding spike count. To apply the nonlinearity to the test data, we used linear interpolation of histogram values. We estimated model performance using the coefficient of determination between model prediction and measured firing rate to obtain the fraction of explained variance ($R^2$). Negative values were clipped to zero.

We calculated video response correlations, using the trial-averaged firing rates of the test stimulus, as the Pearson correlation coefficient between the firing rates for each cell pair of the same type (as well as across specific types). We performed the same analyses for model predictions and for calculating correlations inherent to the test stimulus, where we calculated pairwise correlations of the light intensity of 5,000 randomly selected pixels. Decorrelation was defined for each cell pair as the difference between stimulus and response correlation relative to stimulus correlation for a pixel distance matching that of the actual cell pair distance. To generate correlation–distance curves (Fig. 1e), we sorted cell pairs by ascending distance and averaged pair correlations over groups of 20–60 pairs (depending on cell type, using fewer pairs per bin when the number of available cells was small).

## Spike-train information and fractional redundancy

To estimate whether pairwise correlations led to coding redundancy, we quantified the information contained in ganglion cell spike trains by measuring entropies of response patterns in temporal-frequency space by evaluating the Fourier transforms of the response patterns[75,76]. For temporal patterns that are sufficiently long compared to the time scales of correlations, this approach allows treating the different frequency modes independently and approximating, through the central limit theorem, the empirical distribution of Fourier components by normal distributions whose entropies can be analytically computed[75,76]. This greatly reduces the sampling problem of information-theoretic evaluations encountered by direct methods of computing entropies through empirical frequencies of different response patterns[77].

Here we applied the method to spike-train responses (binned at 0.4 ms) from the repeated parts of the natural video (or white noise) and divided them into 0.8-s-long non-overlapping sections separately for each stimulus trial. This process yielded 27 (or 31) sections for the marmoset (or mouse) natural video and 20–22 sections for white noise for each trial (around 55 white-noise trials for marmoset and 40 for mouse recordings). The selection of the section length aimed at having

comparable numbers of sections per trial and trials per section (both around 30) in the natural video analysis to mitigate bias effects from limited data in the calculation of information rates. For each section ($s$) and trial ($t$), we performed a Fourier transform to obtain complex-valued frequency coefficients that were then separated into real-valued cosine ($c_{cos}$) and sine ($c_{sin}$) coefficients for each frequency ($f$). For a single-cell analysis, we then estimated signal ($H_{signal}$) and noise ($H_{noise}$) entropies by computing the variance of those coefficients either over sections of a given trial or over trials for a given section, respectively, and then averaging the variances over the remaining dimension (that is, trials or sections, respectively):

$$H_{signal} = \frac{1}{2}\log_2[2\pi e(V_{cos}^s + V_{sin}^s)]$$

$$H_{noise}(f) = \frac{1}{2}\log_2[2\pi e(V_{cos}^t + V_{sin}^t)]$$

where, for example, $V_{sin}^s = \langle \mathrm{Var}(c_{sin}(f))_s \rangle_t$ denotes the variance of sine coefficients (subscript 'sin') over sections (superscript 's'), averaged over trials.

The frequency-resolved information rate was calculated as the difference of signal and noise entropies, normalized by the duration of the applied response sections to obtain information per time. The total information rate was then obtained as the sum over frequencies. For this sum, we applied an upper cutoff at 200 Hz, because signal and noise entropies had converged to the same baseline level by then.

For estimating the information content of a cell pair, we proceeded analogously, but instead of computing the variances of the sine and cosine coefficients directly, we first gathered the sine and cosine coefficients from both cells to compile the corresponding 4 × 4 covariance matrix over sections (or trials) and averaged the covariance matrices over trials (or sections). We then used the four eigenvalues ($\lambda_k$) of each averaged covariance matrix to calculate the response entropy for a cell pair (separately for signal and noise):

$$H_{pair}(f) = \frac{1}{2}\log_2\left[2\pi e\left(\sum_{k=1}^{4} \lambda_k(f)\right)\right]$$

Information rates were again obtained as the difference between signal and noise entropy, summed over frequencies and normalized by the duration of the response sections. To check for bias from finite data in the calculation of information rates[78], we also computed information rates for different fractions of the full dataset but observed little systematic dependence on the size of the data fraction. This is for two reasons. First, the possibility to obtain entropies analytically only after estimating the variances of the Fourier components greatly limits the sampling problem, and second, the comparable numbers of trials and sections used in the estimation of entropies mean that any residual bias is of similar scale for signal and noise entropies, thus leading to at least partial cancellation.

To obtain the contributions of individual frequency bands to the information rates, we used the same approach as above separately for each frequency component without summation over frequencies.

Fractional redundancy for a cell pair ($i$, $j$) was calculated on the basis of a previous definition[12,19] as the difference between the sum of single-cell information values ($I_i$ and $I_j$) and pair information ($I_{ij}$) normalized by the minimum single-cell information:

$$C_{ij} = \frac{I_i + I_j - I_{ij}}{\min(I_i, I_j)}$$

Other definitions of redundancy, in particular in the context of efficient coding, are based on a comparison of the actual information passed through an information channel (here corresponding to the

joint responses of the cell pair and their information rate $I_{ij}$) and the channel capacity: that is, the maximum information that the channel could supply[2,79]. In practice, however, channel capacity is difficult to assess and requires fundamental assumptions about the neural code and attainable firing rates. Instead, the comparison of $I_{ij}$ to the sum of single-cell information rates, as used here, can be thought of as capturing whether the capacity as specified by the constraints of the individual cells' response characteristics is fully exhausted by the joint responses. Thus, fractional redundancy is sensitive to inefficient use of the channel capacity that stems from correlation but not from inefficient coding by individual cells.

### Analysis of fixations and spatial contrast

To investigate the effects of spatial contrast on response correlations, we divided the test part (repeated image sequences) of the natural video into distinct fixations by detecting saccadic transitions. To do so, we first marked each time point when a new image was presented as a transition. In each image presentation, we calculated the distance between consecutive positions to estimate the instantaneous eye velocity and used MATLAB's 'findpeaks' function to obtain high-velocity transitions. We constrained peak finding for the marmoset (and mouse) to a minimum peak time interval of 47 (and 53) ms and a minimum amplitude of 10 (and 300) deg s$^{-1}$. This process yielded 80 fixations for the marmoset and 68 fixations for the mouse video. Fano factors were computed for individual fixations. To reduce effects of nonstationary activity, we included here only cells with a positive symmetrized coefficient of determination between the firing-rate profiles of the first half and second half of trials. To mitigate noise from fixations with no or little activity, we excluded, for each cell, fixations with fewer than three spikes on average and report the average Fano factor over fixations, weighted by the mean spike count.

For each video frame and each ganglion cell, spatial contrast was calculated as described previously[36] using the standard deviation of pixels inside the cell's receptive field, weighted by the receptive-field profile. For each fixation, we assigned to each cell the median spatial contrast of all frames during the fixation period. We also assigned a linear activation per fixation, estimated by filtering video frames with the spatial filter obtained from white noise and taking the median over all fixation frames.

To reduce effects of the light level on the analysis of spatial contrast, we aimed at separating the fixations into high-spatial-contrast and low-spatial-contrast groups while balancing the linear activation between the groups. For a pair of cells, we therefore sorted all fixations of the test set by the average linear activation across both cells and paired neighbouring fixations in this sorted list. This led to 40 pairs (34 for the mouse), and for each pair, we assigned the fixation with the higher spatial contrast to the high-spatial-contrast group and the other fixation to the low-spatial-contrast group. To expand the pairwise correlation ($r_{pair}$) into high- and low-spatial-contrast parts, we split the numerator of the Pearson correlation coefficient so that $r_{pair} = r_{high} + r_{low}$, with

$$r_{high} = \frac{\sum_{i \in high} (x_i - \bar{x})(y_i - \bar{y})}{(N_{frames} - 1)\, \sigma_X\, \sigma_Y}$$

with $x$ and $y$ corresponding to the responses of the two cells and $i$ indexing the frames of the natural video, the sum here running over the frames from high-spatial-contrast fixations and $N_{frames}$ denoting the total number of frames. Mean ($\bar{x}$, $\bar{y}$) and standard deviation ($\sigma_X$, $\sigma_Y$) values correspond to the length of the entire test part of the video.

### Extraction of DS ganglion cells

To identify DS ganglion cells in the mouse retina, we used drifting sinusoidal gratings of 100% contrast, 240 μm spatial period and a temporal frequency of 0.6 Hz, moving along eight different, equally spaced

directions. We analysed cell responses as previously described[36]. Cells with a mean firing rate of at least 1 Hz and a direction selectivity index (DSI) of at least 0.2 (significant at 1% level) were considered putative DS cells. The DSI was defined as the magnitude of the normalized complex sum $\sum_\theta r_\theta e^{i\theta} / \sum_\theta r_\theta$, with $\theta$ specifying the drift direction and $r_\theta$ the average (across trials) spike count during the grating presentation for direction $\theta$ (excluding the first grating period). The preferred direction was obtained as the argument of the same sum. The statistical significance of the DSI was determined through a Monte Carlo permutation approach[28,36].

To separate ON from ON–OFF DS cells, we used a moving-bar stimulus. The bar (width: 300 μm, length: 1,005 μm) had 100% contrast and was moved parallel to the bar orientation in eight different directions with a speed of 1,125 μm s$^{-1}$. We extracted a response profile to all bar directions through singular value decomposition, as previously described[24] and calculated an ON–OFF index to determine whether cells responded only to the bar onset (ON) or to both onset and offset (ON–OFF). Cells with an ON–OFF index (computed as the difference of onset and offset spike-count responses divided by their sum) above 0.4 were assigned as ON DS cells and were grouped into three clusters on the basis of their preferred directions.

### Flashed gratings

Depending on the experiment, we generated 1,200 to 2,400 different sinusoidal gratings with 25 or 30 different spatial frequencies ($f$), with half-periods between 15 and 1,200 μm, roughly logarithmically spaced. For each grating, we generated 12 or 10 equally spaced orientations ($\theta$) and four or eight equally spaced spatial phases ($\varphi$). For a given grating, the contrast value for each pixel with ($x$, $y$) coordinates were generated according to the following equation:

$$C(x, y) = \sin(2\pi f(x\cos\theta + y\sin\theta) + \varphi)$$

Gratings were presented as 200 ms flashes on the retina, separated by a 600 or 800 ms grey screen. The order of presentation was pseudorandom. We collected spike-count responses to the flashes by counting spikes during stimulus presentation for the marmoset or 20 ms after stimulus onset up to 20 ms after stimulus offset for the mouse. We used tuning surfaces to summarize responses (Extended Data Fig. 6), which we generated by averaging responses over trials and spatial phases for each frequency–orientation pair. In the mouse recordings, in which we typically collected four to five trials per grating, we calculated symmetrized $R^2$ values for the spike counts, and we only used cells with an $R^2$ of at least 0.2 for further analyses. In marmoset recordings, we typically collected one to two trials per grating, and we thus used no exclusion criterion.

### DoG subunits

The subunit grid model consists of DoG subunits, and fitting its parameters to data is facilitated by an analytical solution of the DoG activation by a grating. The latter was obtained by considering the grating activations of both centre and surround elliptical Gaussians on the basis of previous calculations[80], as described in the following. The DoG receptive field was defined with these parameters: standard deviations $\sigma_x$ and $\sigma_y$ at the $x$ and $y$ axis, the orientation of the $x$ axis $\theta_{DoG}$, the spatial scaling for the subunit surround $k_s$ and a factor determining the relative strength of the surround $w_s$. Concretely, the response of a DoG receptive field ($r_{DoG}$) centred at ($x_o$, $y_o$) to a parametric sinusoidal grating ($f$, $\theta$, $\varphi$) is

$$r_{DoG}(f, \theta, \varphi; x_o, y_o, \sigma_x, \sigma_y, \theta_{DoG}, k_s, w_s)$$
$$= A_{DoG}(f; \sigma_x, \sigma_y, \theta_{DoG}, k_s, w_s) \times \cos\Theta_{DoG}(f, \theta, \varphi; x_o, y_o)$$

with the amplitude $A_{DoG}$ given by

$$A_{\text{DoG}}(f; \sigma, k_s, w_s) = e^{-2\pi\sigma_{\text{DoG}}^2 f^2} - w_s e^{-2\pi(k_s\sigma_{\text{DoG}})^2 f^2}$$

with

$$\sigma_{\text{DoG}} = \sqrt{\sigma_y^2 \sin^2(\theta + \theta_{\text{DoG}}) + \sigma_x^2 \cos^2(\theta + \theta_{\text{DoG}})}$$

The receptive-field phase $\Theta_{\text{DoG}}$ is given by

$$\Theta_{\text{DoG}}(f, \theta, \varphi; x_o, y_o) = 2\pi f \sqrt{x_o^2 + y_o^2} \cos\left(\theta - \tan^{-1}\frac{y_o}{x_o}\right) + \varphi - \pi/2$$

## DoG LN model
We fitted parameterized DoG LN models to the measured grating responses. The full model combined the DoG receptive-field activation with an output nonlinearity, for which we chose a logistic function $N(x) = (1 + e^{-x})^{-1}$. The model response ($R$), denoting the modelled neuron's firing rate, was thus given by

$$R = aN(\beta_{\text{DoG}}r_{\text{DoG}} + \gamma_{\text{DoG}})$$

where $\beta_{\text{DoG}}$ and $\gamma_{\text{DoG}}$ are parameters determining the steepness and threshold of the output nonlinearity and $a$ is a response scaling factor.

All model parameters ($x_o, y_o, \sigma_x, \sigma_y, \theta_{\text{DoG}}, k_s, w_s, \beta_{\text{DoG}}, \gamma_{\text{DoG}}, a$) were optimized simultaneously by minimizing the negative Poisson log-likelihood, using constrained gradient descent in MATLAB with the following constraints: $\sigma_x, \sigma_y > 7.5\,\mu m$, $-\pi/4 < \theta_{\text{DoG}} < \pi/4$, $1 < k_s < 6$, $a > 0$. Each trial was used independently for fitting.

## Subunit grid model
We fitted all subunit grid models with 1,200 potential subunit locations, placed on a hexagonal grid around a given receptive-field centre location. The centre was taken as the centre of a fitted DoG model. The subunits were spaced 16 μm apart. Each subunit had a circular DoG profile with a standard deviation of $\sigma$ (centre Gaussian) and centred at $(x_{os}, y_{os})$, and its activation in response to a grating was given by

$$r_s(f, \theta, \varphi; x_{os}, y_{os}, \sigma, k_s, w_s) = A_s(f; \sigma, k_s, w_s) \times \cos\Theta_s(f, \theta, \varphi; x_{os}, y_{os})$$

where both amplitude and phase are given by the DoG receptive-field formulas with $\sigma_x = \sigma_y = \sigma$.

The full response model was

$$R_{\text{SG}} = G\left(\sum_{s=1}^{N_{\text{sub}}} w_s N(\beta r_s + \gamma)\right)$$

where $N(x) = (1 + e^{-x})^{-1}$ is a logistic function, $\beta$ and $\gamma$ are parameters determining the steepness and threshold of the subunit nonlinearity, $N_{\text{sub}}$ is the number of subunits with non-zero weights, $w_s$ are positive subunit weights, and $G$ is a Naka–Rushton output nonlinearity $G(x) = ax^n/(x^n + k^n) + b$, with non-negative parameters $\theta_{\text{out}} = (a, b, n, k)$.

## Fitting and model selection
We optimized subunit grid models using the stochastic optimization method ADAM[81] with the following parameters: batch size = 64, $\beta_1 = 0.9$, $\beta_2 = 0.999$, $\varepsilon = 10^{-6}$. For the learning rate ($\eta$), we used a schedule with a Gaussian profile of $\mu = N_{\text{epochs}}/2$ and $\sigma = N_{\text{epochs}}/5$: this led to a learning rate that was low in the beginning of the training, peaked midway and was lowered again towards the end. Peak learning rate was set to $\eta_{\max} = 0.005$. The number of epochs ($N_{\text{epochs}}$) was fixed for all cells to $4 \times 10^5/N_{\text{trials}}$, with $N_{\text{trials}}$ representing the number of all grating presentations used for fitting, which typically resulted in 50–150 epochs.

To enforce parameter constraints during fitting, such as non-negativity, we used projected gradient descent. We also aimed at regularizing the parameter search in a way that non-zero subunit weights were penalized more strongly when other subunits with non-zero

weights were spatially close. We therefore introduced a density-based regularizer that controlled the coverage of the receptive field with a flexible number of subunits (Extended Data Fig. 6).

Concretely, the cost function we minimized was

$$-\frac{1}{N_{\text{sp}}}\ln L(\mathbf{s_G}, \mathbf{r_G}; \theta_{\mathbf{p}}, \mathbf{w}) + \lambda \sum_{s=1}^{N_{\text{sub}}} w_s \sum_{i \neq s} \frac{w_i}{d_{si}^2}$$

with $N_{\text{sp}}$ being the total number of spikes, $L$ the Poisson likelihood, $\mathbf{s_G}$ the vector of all grating parameters used, $\mathbf{r_G}$ the corresponding spike-count response vector, $\theta_{\mathbf{p}} = (\sigma, k_s, w_s, \beta, \gamma, \theta_{\text{out}})$ all the shared model parameters, and $\mathbf{w} = (w_1, \ldots, w_{N_{\text{sub}}})$ the vector containing all subunit weights. $\lambda$ controls the regularization strength, which depends on the pairwise subunit distances $d_{si}$.

After the end of the optimization, we pruned subunit weights with small contributions or weights that ended up outside the receptive field. To do so, we first set to zero every weight smaller than 5% of the maximum subunit weight. We then fitted a two-dimensional Gaussian to an estimate of the receptive field, obtained by summing subunit receptive fields weighted by the subunit weights. The weight corresponding to any subunit centre lying more than $2.5\sigma$ outside that Gaussian was set to zero. To ensure proper scaling of the output nonlinearity after weight pruning, we refitted the output nonlinearity parameters along with a global scaling factor for the weights.

We typically fitted six models per cell with different regularization strengths $\lambda$ ranging from $10^{-6}$ to $5 \times 10^{-4}$. To select for the appropriate amount of regularization, we only accepted models that yielded at least three subunits and had a low receptive-field coverage (less than 3; see below). If no eligible model was fitted, cells were excluded from further analyses. Among the remaining models, we selected the one that minimized the Bayesian information criterion, which we defined for the subunit grid model as

$$N_{\text{sub}}\ln(N_{\text{data}}) - 2\ln(L)$$

where $N_{\text{sub}}$ is again the number of subunits with non-zero weights, $N_{\text{data}}$ is the number of grating-response pairs used to fit the model and $L$ the likelihood of the fitted model. The selected model balanced good prediction performance and realistic receptive-field substructure. Note, though, that the actual size and layout of the subunits might not be critical to obtain good model performance[31] as long as appropriate spatial nonlinearities are included, and that the subunit nonlinearities are generally better constrained by the data than the subunits themselves (Extended Data Fig. 6).

## Parameter characterization of the subunit grid model
To summarize how densely subunits covered a cell's receptive field, we defined a measure for the subunit coverage. It was calculated as the ratio A/B, where A was the subunit diameter ($4\sigma$ of the centre Gaussian) and B was the average distance between subunit centre points. For a particular cell, the average subunit distance was calculated as the average over all nearest-neighbour distances, weighted by each pair's average subunit weight. If fewer than three subunits had non-zero weights in the model, no coverage value was computed.

To plot and characterize subunit nonlinearities, we first added an offset so that an input of zero corresponded to zero output. We then scaled the nonlinearities so that the maximum value over the input range [−1, 1] was unity. Following offsetting and scaling, we calculated nonlinearity asymmetries to quantify the response linearity of subunits as $(1 − M)/(1 + M)$, where $M$ is the absolute value of the minimum of the nonlinearity over the input range [−1, 1].

## Natural images and response predictions
We flashed a series of 220 (or 120) natural images to the retina, as described previously[36]. We used images from the van Hateren database,

which were cropped to their central 512 × 512 pixel square and presented over the multielectrode array at single-pixel resolution. All images were multiplicatively scaled to have the same mean intensity as the background. Interspersed with the natural images, we also presented artificial images. The images were generated as black-and-white random patterns at a single-pixel level and then blurred with Gaussians of eight different spatial scales[29], but the corresponding responses were not analysed as part of this study. All images were flashed for 200 ms, separated by either 600 or 800 ms of grey background illumination. Images were flashed in a randomized order, and we typically collected eight trials per image. Average spike counts were calculated in the same way as in the case of flashed gratings, and only cells with symmetrized $R^2$ of at least 0.2 were used for further analyses.

To calculate response predictions for models built with white noise, we used the output of spatial filters applied to the natural images. The filters were upsampled to match the resolution of the presented images and normalized by the sum of their absolute values. For models obtained from responses to flashed gratings, DoG receptive fields were instantiated at single-pixel resolution, and the natural images were then projected onto the DoG receptive fields. For the subunit model, the subunit filter outputs were passed through the fitted subunit nonlinearity and then summed while applying the subunit weights. The performance for each model was calculated as the Spearman rank correlation $\rho$ between the model output (without an explicit output nonlinearity) and cell responses to the natural images[28].

### Flickering gratings and spatiotemporal DoG LN models

We generated 3,000 (or 4,800) different gratings with 25 (or 30) different spatial frequencies, between 7.5 and 1,200 µm half-periods, roughly logarithmically spaced. For each grating, we generated 20 orientations and six (or eight) spatial phases. The gratings were presented in pseudorandom sequence, updated at a 85 (or 75) Hz refresh rate. Every 6,120 (or 3,600) frames, we interleaved a unique sequence of 1,530 (or 1,200) frames that was repeated throughout the recording to evaluate response quality.

We fitted a spatiotemporal DoG LN model to the grating responses. The temporal filters spanned a duration of 500 ms and were modelled as a linear combination of ten basis functions. The response delay was accounted for with two square basis functions for each of the two frames before a spike. The remaining eight were chosen from a raised cosine basis, with peaks ranging from 0 to 250 ms before a spike.

Concretely, the spatiotemporal DoG model had the form

$$R = aN(\mathbf{r}_C^T \mathbf{k}_{Ct} + \mathbf{r}_S^T \mathbf{k}_{St} + b)$$

where $N(x) = (1 + e^{-x})^{-1}$ is a logistic function, $\mathbf{k}_{Ct}$ and $\mathbf{k}_{St}$ are separate temporal filters for the centre and the surround, $b$ determines the baseline activation, and $a$ is a response scaling factor. The vectors $\mathbf{r}_C$ and $\mathbf{r}_S$ contain DoG receptive-field activations for 500 ms before a particular frame and were calculated as for the flashed gratings. The model was fitted with nonlinear constrained optimization, with DoG constraints identical to the case of flashed gratings and $a > 0$.

### Spatiotemporal subunit grid model

We also fitted a spatiotemporal subunit grid model to the grating responses. Our strategy was similar to the case of flashed gratings. We fitted each subunit grid model with 1,200 subunit locations, placed in a hexagonal grid around a given receptive-field centre location. The centre was taken as the fitted centre of the DoG model. The subunits were spaced 16 µm apart. The model response ($R$) was given by

$$R = G\left(\sum_{s=1}^{N_{sub}} w_s N(\mathbf{r}_C^T \mathbf{k}_{Ct} + \mathbf{r}_S^T \mathbf{k}_{St} + \gamma)\right)$$

where $w_s$ are non-negative subunit weights, $N(x)$ is a logistic function, $\mathbf{k}_{Ct}$ and $\mathbf{k}_{St}$ are separate temporal filters for the centre and the surround shared across all subunits, and $\gamma$ determines the nonlinearity threshold. We used a Naka–Rushton output nonlinearity $G(x) = ax^n/(x^n + k^n)$, with non-negative parameters $\boldsymbol{\theta}_{out} = (a, n, k)$. The vectors $\mathbf{r}_c$ ($\mathbf{r}_s$) contain Gaussian centre (surround) subunit activations for 500 ms before a particular frame and for each subunit. The parameters required to fit DoG subunits are the standard deviation of the centre, the scaling for the subunit surround and a factor determining the relative strength of the surround.

We used stochastic gradient descent with the ADAM optimizer to fit spatiotemporal models. The parameters were the same as in the flashed-grating models, except for the batch size = 2,000 and $\eta_{max} = 0.02$. We used the same learning schedule for $\eta$ and the same regularization to control for subunit density as in the case of flashed gratings.

### Natural video predictions of grating-fitted models

To obtain natural video predictions for models built from flickering gratings, we instantiated receptive fields, as well as subunit filters, at single-pixel resolution. Again, we projected video frames on centre and surround filters separately, convolved each result with the corresponding temporal filter and summed the two outputs for obtaining the final filter output. For subunit grid models, the subunit nonlinearity fitted from the gratings was applied to the linear subunit outputs, which were then summed with the non-negative weights to obtain the final activation signal. The training part of the video was used for estimating an output nonlinearity using maximum likelihood (under Poisson spiking) for both the DoG LN and the subunit grid models. The nonlinearity had the same parametric form as in the model fit with gratings. Unlike the models applied to natural images, in which the Bayesian information criterion was applied, we here used the training set to select the appropriate regularization strength by finding the maximum of the log-likelihood among the eligible models (with at least three subunits and a receptive-field coverage below 3). If no eligible model was fitted, cells were excluded from further analyses. Model performance was estimated for the test set using the coefficient of determination between model prediction and measured firing rate as a fraction of explained variance ($R^2$). Negative values of $R^2$ were clipped to zero.

To better differentiate DoG and subunit grid model performance, we selected fixations on the basis of model predictions. For each cell pair, we selected the 20% of the fixations for which the deviation in the predictions of the two models, averaged over the two cells, was largest. For a single cell and a single fixation, the deviations were calculated as the absolute value of model differences normalized by the cell's overall response range (maximum minus minimum during the test part of the video) and averaged over all frames of the fixation. Performance of both models ($R^2$) was then compared to the frame-by-frame neural response on these fixations and averaged over the two cells. The selection of maximally differentiating fixations does not favour either model a priori, because it is only based on how much model predictions differ and not on their performance in explaining the data.

Similar to the spatial contrast analysis, we expanded the pairwise correlation ($r_{pair}$) into linear and nonlinear contributions by splitting the numerator of the Pearson correlation coefficient so that $r_{pair} = r_{nonlinear} + r_{linear}$. For a pair of cells, we sorted all fixations (in descending order) by the average deviation of model predictions. We assigned the first half of the fixations to the nonlinear group and the remaining ones to the linear group.

### Reporting summary

Further information on research design is available in the Nature Portfolio Reporting Summary linked to this article.

## Data availability

The recorded spiking responses to natural stimuli and gratings have been made publicly available at https://gin.g-node.org/gollischlab/Karamanlis_Gollisch_2023_RGC_spiketrains_natural_movies_and_subunit_models (https://doi.org/10.12751/g-node.ejk8kx). For visual stimulation, nature images from the van Hateren database can be downloaded from an openly accessible repository at https://pirsquared.org/research/vhatdb/full/. Source data are provided with this paper.

## Code availability

The code to fit subunit grid models to grating data is available at https://github.com/dimokaramanlis/subunit_grid_model. The modified Kilosort code for spike sorting is available at https://github.com/dimokaramanlis/KiloSortMEA. The phy software is available at https://github.com/cortex-lab/phy.

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

**Acknowledgements** We thank J. Yates and J. Mitchell for providing eye movement data from marmosets and corresponding natural images; A. Meyer for providing mouse horizontal gaze traces; members of the Gollisch lab for assistance during marmoset data collection; F. Rieke and J. Anguera for advice on experiments with the primate retina; and N. Sirmpilatze for comments on the manuscript. This work was funded by the European Research Council under the European Union's Horizon 2020 research and innovation programme (grant no. 724822) and by the Deutsche Forschungsgemeinschaft (German Research Foundation), project IDs 432680300 (SFB 1456, project B05) and 515774656. D.K. was supported by a Boehringer Ingelheim Fonds fellowship.

**Author contributions** D.K. designed the experiments and collected and analysed data with supervision from T.G. M.H.K., H.M.S., S.J.Z. and M.M. assisted with marmoset data collection. S.J.Z. performed spike-triggered non-negative matrix factorization analyses. D.K. and T.G. wrote the paper with input from all authors.

**Competing interests** The authors declare no competing interests.

**Additional information**
**Correspondence and requests for materials** should be addressed to Dimokratis Karamanlis or Tim Gollisch.

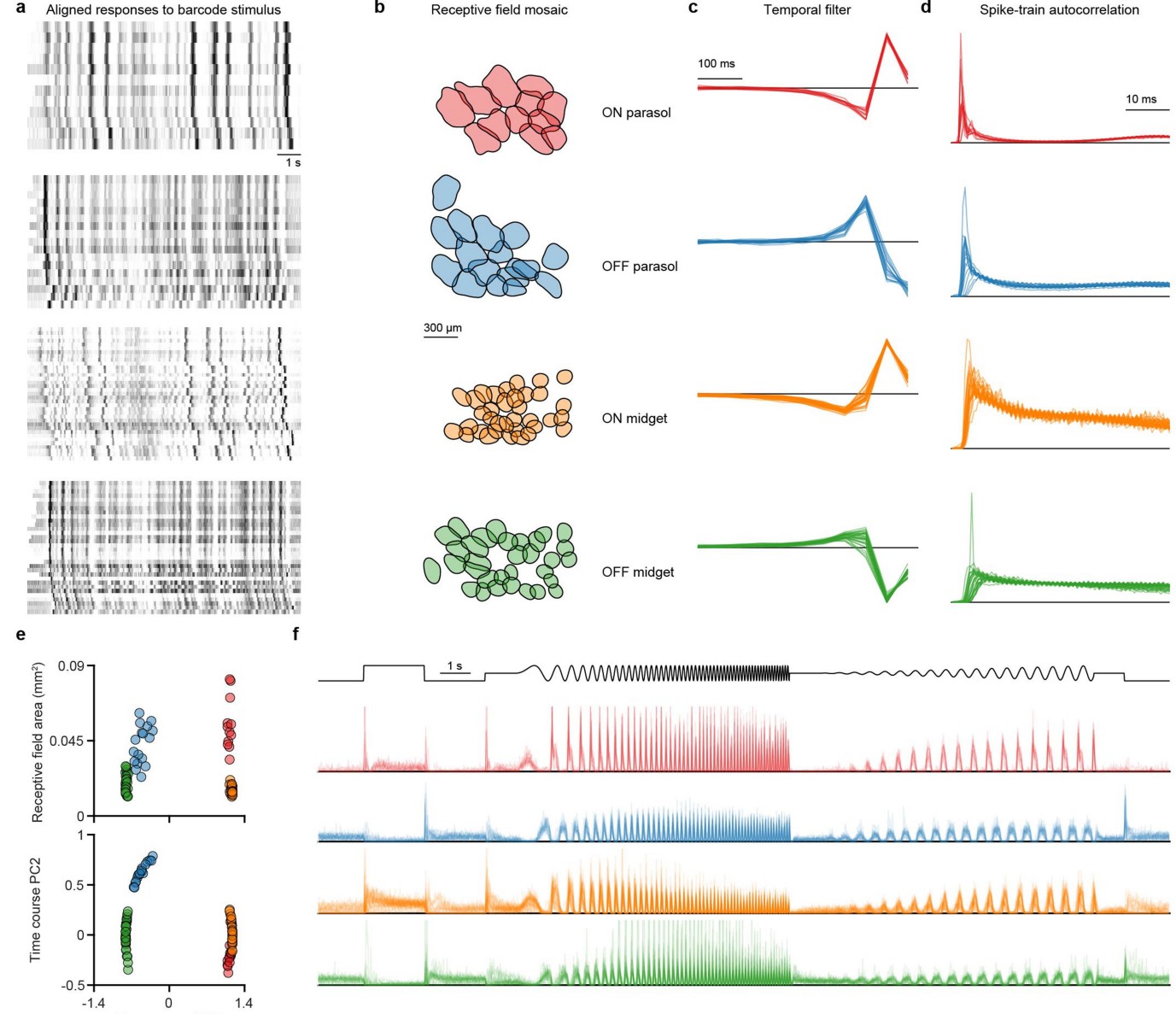

**Extended Data Fig. 1 | Cell type identification in the marmoset retina.**
The retina was stimulated with a barcode stimulus. Cell responses were then
clustered along with information from white-noise stimulation (receptive field
size, temporal filter, autocorrelation). **a**, Responses to the barcode stimulus of
four identified clusters for a single retina. These responses were aligned to a
seed cell (first row) to show the match. **b**, Receptive-field mosaics of the four
identified clusters. **c**, Temporal filters. **d**, Spike-train autocorrelations (bin size
is 0.5 ms). **e**, Clustering of identified cell types, shown by projections into

two-dimensional parameter spaces. Receptive field area was calculated from
the estimated contours. Time course PC1 and PC2 measure the projections of
the cells' temporal filters onto the first two principal components of all temporal
filters in the same retina piece. **f**, Responses of the identified cells (colored
traces) to a chirp stimulus (black trace on top, depicting the applied light
intensity over time). PSTHs are calculated with 10-ms bins and normalized to
unit sum for plotting.

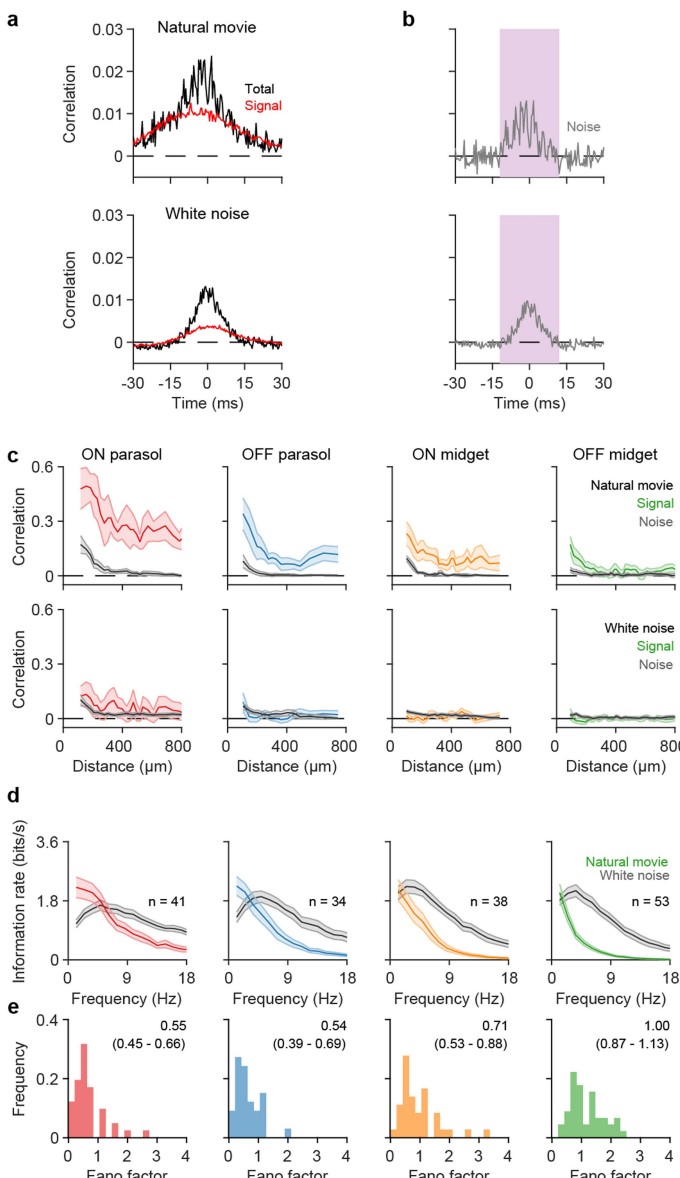

**Extended Data Fig. 2 | Signal and noise correlations, information rates, and Fano factors in the marmoset retina. a**, Total cross-correlation (black) for a pair of ON parasol cells and a shuffle predictor of signal correlation (red) for the natural movie (top) and white noise (bottom), using the repeated stimulus sections of both stimuli. The shuffle predictor was generated by averaging cross-correlations calculated after randomly permuting trials of the repeated stimulus (natural movie or white noise) 15 times. **b**, Noise correlations were estimated by subtracting the shuffle predictor from the total correlation. Shaded area marks the area used for summarizing noise correlations. **c**, For each cell type, the area under the noise cross-correlation curve is plotted against receptive field distance for natural movie (top) and white noise (bottom) data. **d**, Cell-type-specific information rates for different response frequencies under the natural movie (colored) or spatiotemporal white noise (grey). Colored lines represent averages for pairs at similar x-coordinates (with 95% confidence intervals) within the same ganglion cell type. **e**, Fano factor histograms for cell responses to the natural movie. Fano factors were averaged over fixations for each cell, and for each cell type, the median across cells (with 95% robust confidence intervals in parentheses) is reported in the figure. Data from 3 retina pieces.

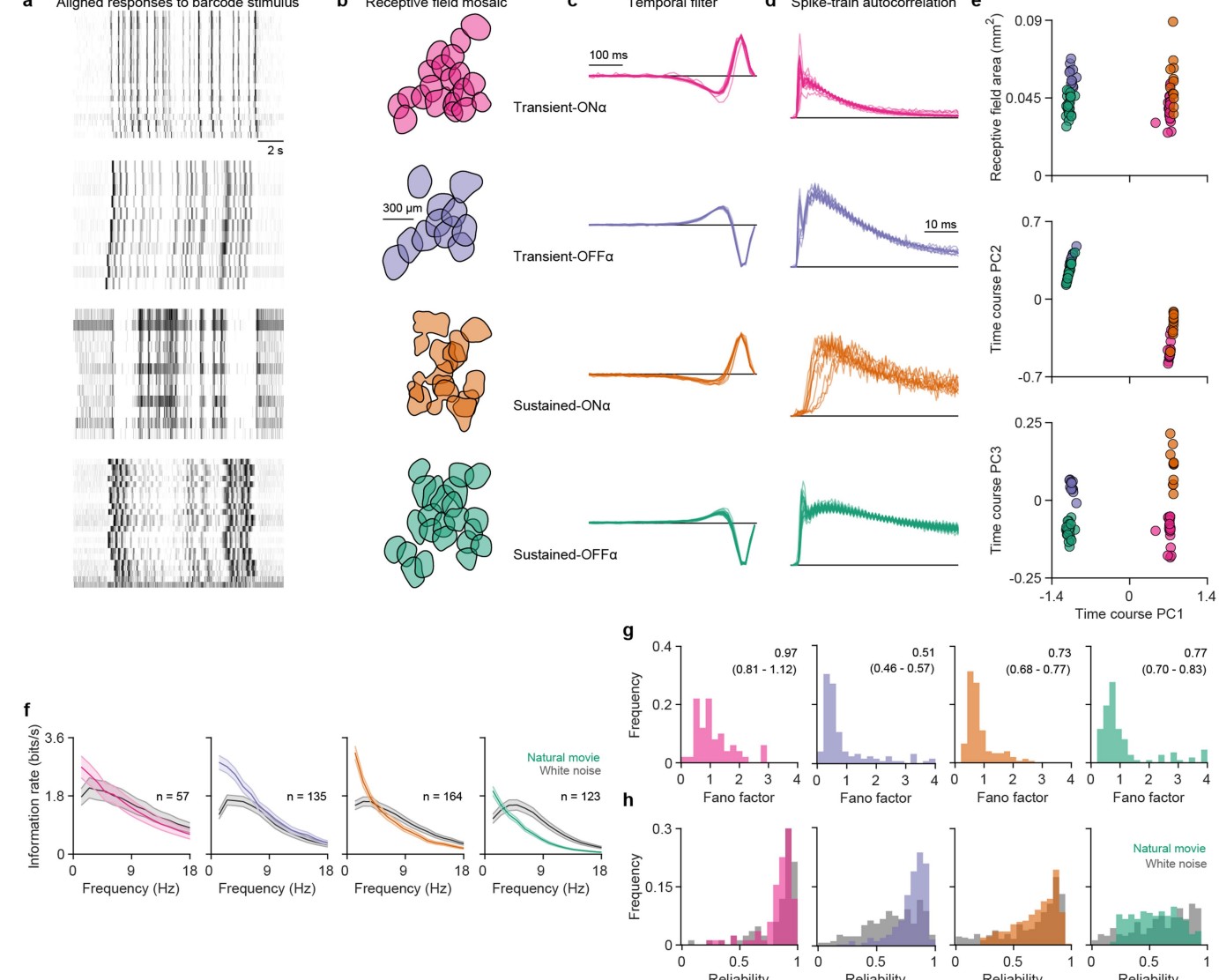

**Extended Data Fig. 3 | Cell type identification in the mouse retina.**
The retina was stimulated with a barcode stimulus. Cell responses were then clustered along with information from white-noise stimulation (receptive field size, temporal filter, autocorrelation). **a**, Responses to the barcode stimulus of four identified clusters. These responses were aligned to a seed cell (first row) to show the match. **b**, Receptive-field mosaics of the four identified clusters. **c**, Temporal filters. **d**, Spike-train autocorrelations (bin size is 0.5 ms). **e**, Clustering of identified cell types, shown in similar fashion as in Extended Data Fig. 1e. **f**, Cell-type-specific information rates for different response frequencies under

the natural movie (colored) or spatiotemporal white noise (grey). Colored lines represent averages for pairs at similar x-coordinates (with 95% confidence intervals) within the same ganglion cell type. **g**, Fano factor histograms for cell responses to the natural movie, calculated for each cell by considering spike counts for individual fixations. The reported Fano factors are medians across cells (95% robust confidence intervals in parentheses). **h**, Histograms of response reliability under natural movies and white noise, measured by the coefficient of determination between firing rates of even and odd stimulus repeats. Data for **f-h** are from 8 retina pieces.

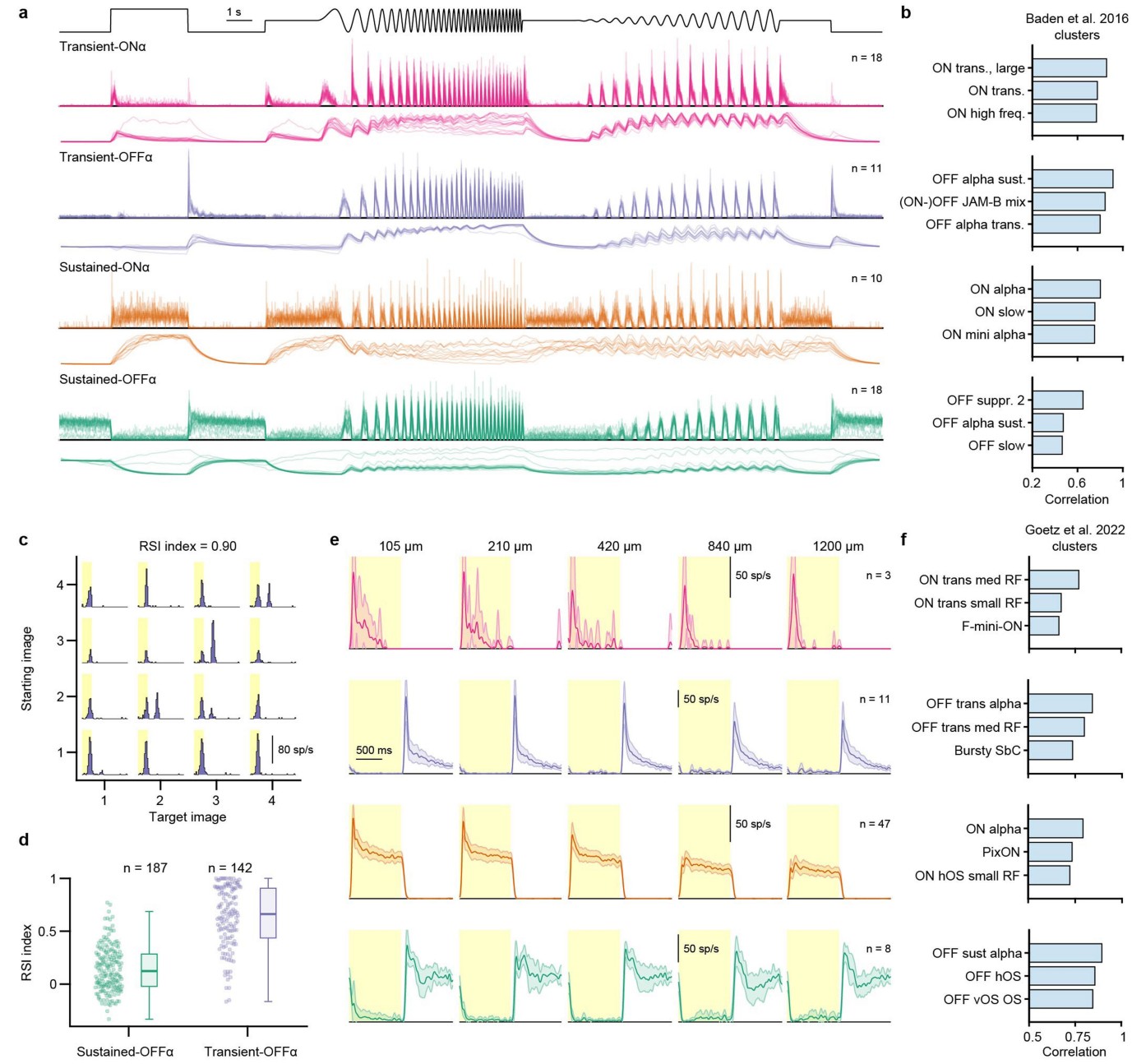

**Extended Data Fig. 4 | Mapping identified ganglion cell mosaics to alpha types in the mouse retina. a**, Responses of four identified ganglion cell types to the chirp stimulus (top), previously used to classify mouse retinal ganglion cells[24]. We compared these four functional types, comprised of ON and OFF transient and sustained cells, with standard databases and confirmed their assignment to mouse alpha cells. Spiking responses were converted to a calcium-equivalent signal (shown below the firing-rate profiles). **b**, Comparison with reported cell types. We calculated the average Pearson correlation of the cells' calcium-equivalent signals with the reported templates[24]. Shown are the top three hits. **c**, An OFF transient cell showing image recurrence sensitivity, measured with saccade gratings[71]. This sensitivity was quantified with the recurrence sensitivity index (RSI), as done previously[71]. **d**, OFF transient cells had significantly higher RSI indices than OFF sustained cells (mean ± SD vs. mean ± SD, mean ± SD, p < 10[−50], two-sided Wilcoxon rank-sum test), and the indices were significantly larger than 0.5 (p = 0.038, two-sided Wilcoxon sign-rank test), the threshold used for the original characterization. This corroborates that the identified OFF transient cells correspond to transient-OFFα cells. **e**, Average responses of the four main types to flashed spots of five different sizes. The spots were flashed either within, or very close to the receptive field centers of the selected cells. Shaded error bars are 95% confidence intervals. **f**, Spot responses were compared (via correlations) to a functional database[23]. Top three hits are shown, generally matching the alpha types. For ON transient cells, the match is ON transient medium RF, hypothesized to match the original description of the transient-ONα cell[25].

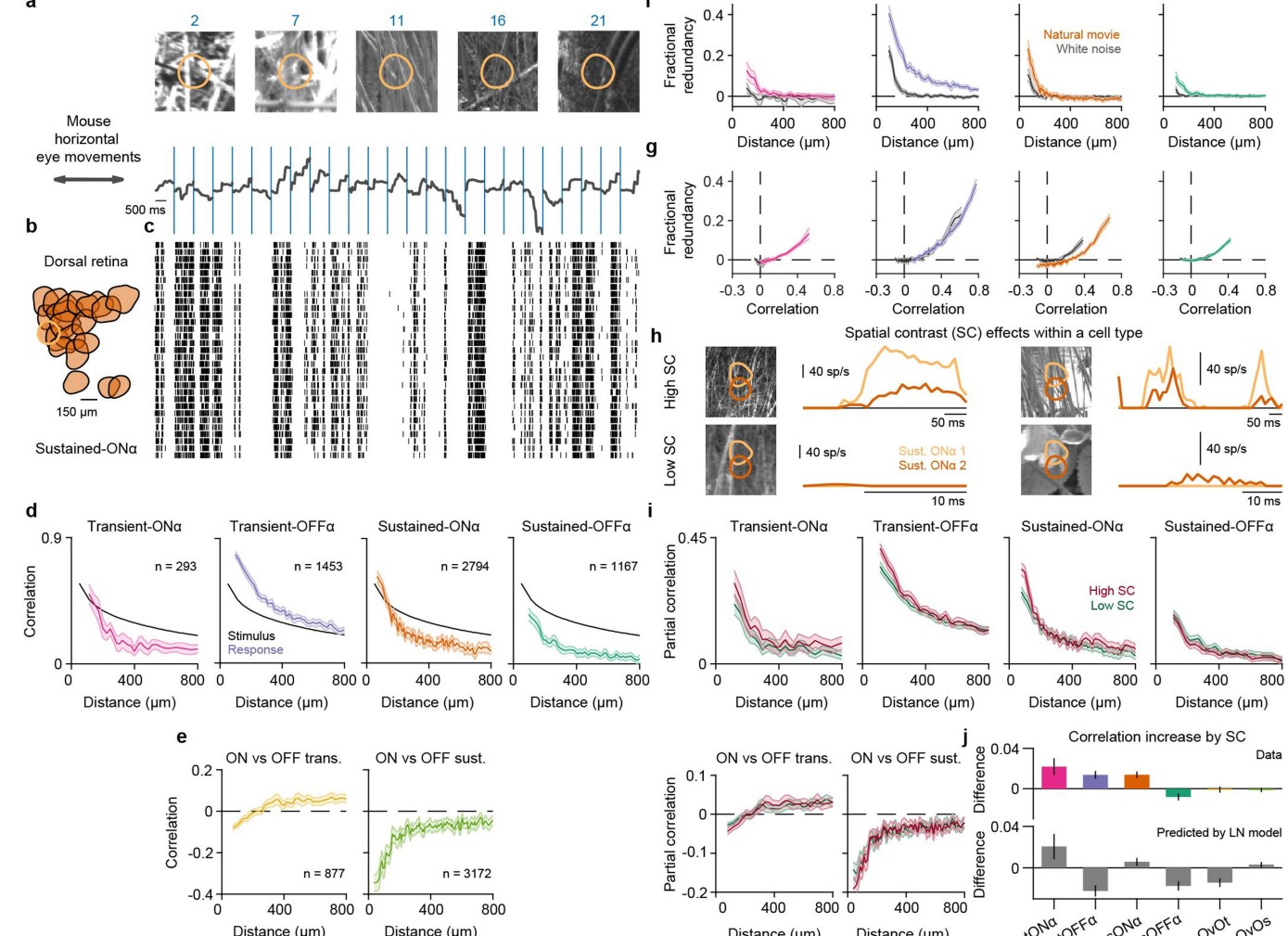

**Extended Data Fig. 5 | Mouse natural movies, pairwise correlations, and spatial contrast analysis. a**, We generated mouse-specific movies by shifting natural images according to horizontal gaze traces recorded from freely-moving mice. Each image was presented for 1 s (annotated by the blue lines) and displaced along a cardinal direction that was randomly assigned per image. The receptive field of a sample sustained-ONα cell is overlaid on the displayed five sample images (order in image sequence given by the blue numbers). **b**, Receptive-field mosaic of simultaneously recorded sustained-ONα cells, with the outline of a sample cell highlighted. **c**, Spike-raster of the sample cell for 30 trials of the stimulus in **a**. **d**, Pearson correlation coefficient for the natural-movie responses of ganglion cell pairs as a function of their distance. Colored lines represent average correlation for pairs at similar distance (with 95% confidence intervals) within the same ganglion cell type from three retinas. For reference, black lines show the correlation between stimulus pixels. **e**, Same as **d**, but for pairs of ON and OFF ganglion cells. **f**, Fractional redundancy for cell pairs as a function of receptive field distance. **g**, The relationship between correlation and fractional

redundancy under natural movies for cell pairs. **h**, Responses of two neighboring sustained-ONα cells to fixations with similar light intensity, but either high (top) or low (bottom) spatial contrast. **i**, Pairwise partial correlations, obtained for high- and low-spatial-contrast fixations, respectively. **j**, Median differences between high- and low-spatial-contrast partial correlations in the data (top) as well as their predictions from fitted linear-nonlinear (LN) models (bottom). The measured correlation increases by spatial contrast were statistically significant (one-sided Wilcoxon sign-rank test) for transient-ONα (tONα, p = 5.5·10⁻⁹), transient-OFFα (tOFFα, p = 9·10⁻¹¹), and sustained-ONα (sONα, p = 1.4·10⁻²⁰) cell pairs, but not for sustained-OFFα (sOFFα, p > 0.99; here, correlations decreased slightly but significantly) ON vs. OFF transient α (OvOt, p = 0.39) or ON vs. OFF sustained α (OvOs, p = 0.90) cell pairs. Number of cells (cell pairs) n = 57(293)/135(1453)/164(2794)/123(1167) for tONα/tOFFα/sONα/sOFFα types, and n = 877/3172 cell pairs of OvOt/OvOs type pairs. Error bars are robust confidence intervals (95%), and data are from 8 retina pieces.

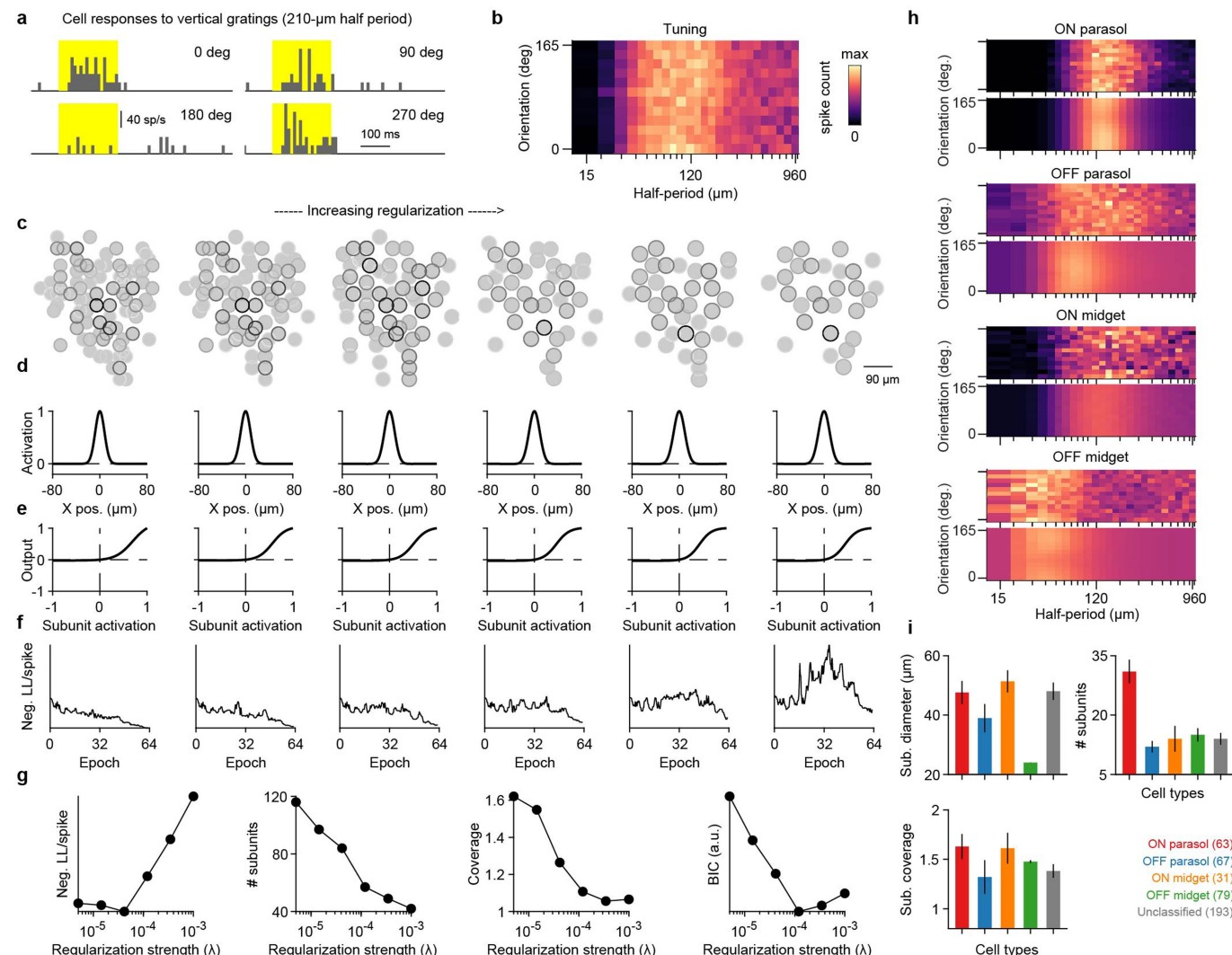

**Extended Data Fig. 6 | Weight density regularization and ganglion cell responses to flashed gratings. a**, Responses of a mouse retinal ganglion cell to flashed gratings of different spatial phases. **b**, Tuning surface summary of the responses for the same cell, responses for each orientation/spatial period pair were averaged over phases and trials. Note that for fitting subunit grid models, additional important information is contained in the response differences for different grating phases, which is not visible in this summary response plot. **c**, Subunit layouts from model fits of the sample cell for six different regularization values. **d**, Subunit receptive field profile for each fit. **e**, Corresponding nonlinearity. **f**, Parameters were fitted by minimizing the negative Poisson log-likelihood (Neg. LL). Training curves were smoothed with a moving median filter (length of ten points) for plotting. **g**, Effects of varying

regularization strength on the cost function at the end of the optimization, on the number of subunits, on subunit coverage, and on the Bayesian Information Criterion (BIC), which was used to select the best model. The number of parameters in the BIC was here given by the number of subunits with non-zero weights. **h**, Comparison of the actual, measured tuning surface (top) vs. its prediction from the SG model (bottom) for four sample cells of the marmoset retina (same as in Fig. 3), exemplifying that subunit grid models could fit responses to the flashed gratings reasonably well. Color map as in b. **i**, Median values of subunit diameter, number of subunits per cell, and subunit coverage of receptive field for the identified four marmoset ganglion cell types as well as for unclassified marmoset ganglion cells. Error bars are median ± 95% robust confidence interval, and data are from 3 retina pieces.

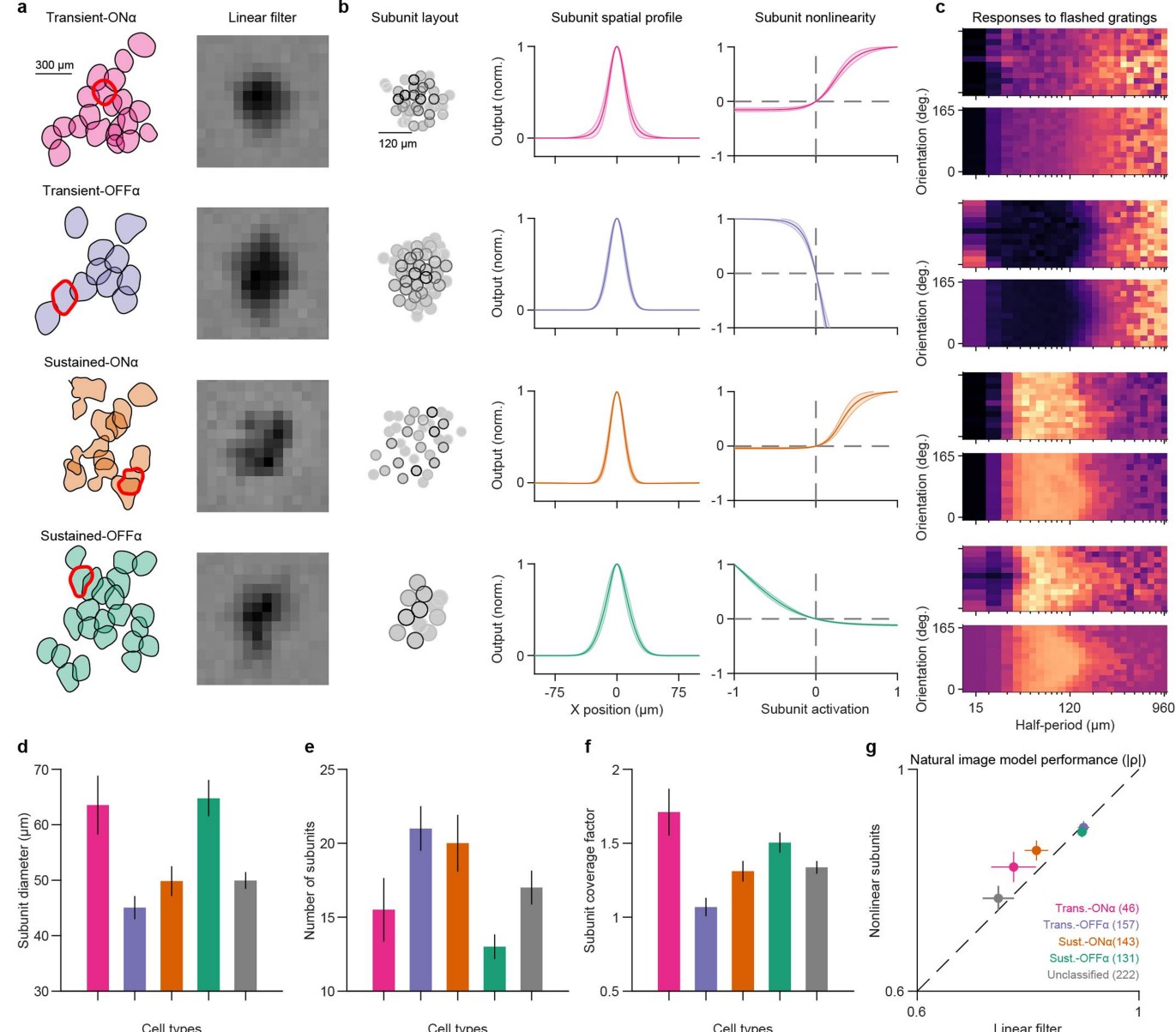

**Extended Data Fig. 7 | Cell type analysis of model parameters for the mouse retina. a**, Left: Receptive-field mosaics for the four identified mouse ganglion cell types from a sample recording with a good representation of all four types. For clarity of the display, contours are here shrunk by 20%. Right: White-noise spatial filters for the sample cells highlighted in the mosaics on the left. Darker pixels in spatial filters denote larger (positive) values. **b**, Subunit layouts for the sample cells in **a** (left), and average spatial profiles (middle) and nonlinearities (right) of the obtained subunit grid models for all ganglion cells of the corresponding type in the recording. Shaded areas show 95% confidence intervals around the mean. **c**, Comparison of the actual, measured tuning surface (top) vs. its prediction from the SG model (bottom) for the four sample cells), exemplifying that subunit grid models could fit mouse ganglion cell responses to the flashed gratings reasonably well. Colormap of the tuning

surfaces as in Extended Data Fig. 6b. **d**, Median subunit diameters for all four types. The values are consistent with previous receptive field measurements of bipolar cells that provide input to alpha-type ganglion cells[29,82] (40–70 μm). **e**, Median numbers of subunits for all types. **f**, Median coverage of the subunit mosaics. **g**, Performances of the subunit grid model ("nonlinear subunits") and the LN model ("linear filter") in predicting responses to natural images, calculated as the absolute value of Spearman's ρ between predictions and measured average spike counts. Improvements in model performance by using the obtained subunits over linear receptive fields can be seen for most cell types except for sustained-OFFα cells, which typically displayed rather linear subunit outputs, as well as transient-OFFα cells, which were well-predicted by a linear receptive field, as reported previously[36]. Error bars in **d-g** are median ± 95% robust confidence interval, and data are from 8 retina pieces.

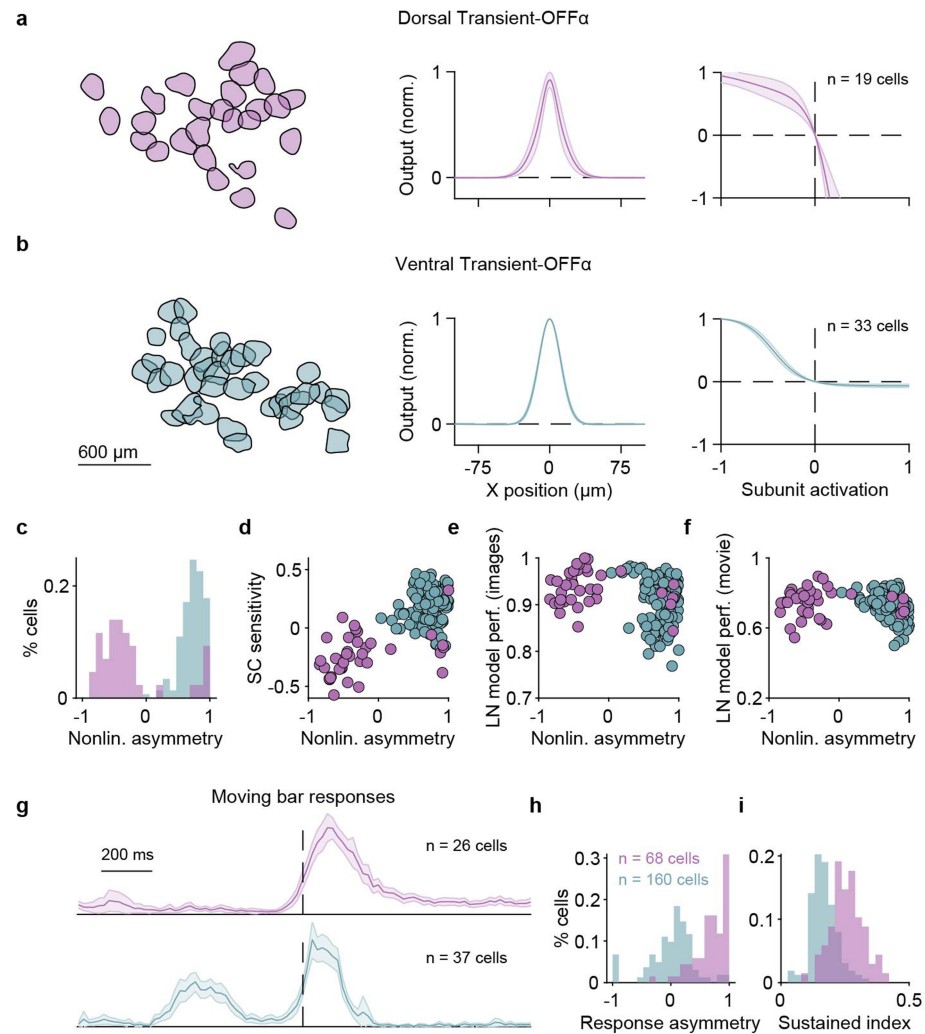

**Extended Data Fig. 8 | Subunit nonlinearities differ between dorsal and ventral transient-OFFα cells. a**, Subunit model parameters for a transient-OFFα cell mosaic in the dorsal retina. **b**, Same as **a**, but for a recording from the ventral part of the same retina. **c**, Asymmetry in the nonlinearities is evident for all recorded transient-OFFα cells. (See Methods for calculation of nonlinearity asymmetry. Positive asymmetry values correspond to rectification of negative output by the nonlinearity, negative values to stronger negative than positive outputs of the nonlinearity). **d**, The asymmetry in the nonlinearities was related to the cells' sensitivity to spatial contrast (SC), as measured from responses to natural images. Spatial-contrast sensitivity was computed as described previously[36]. Negative spatial-contrast sensitivity corresponds to a preference for spatially homogenous light intensity inside the receptive field. **e-f**, The asymmetry was also related to linear-nonlinear (LN) model performance both for natural images (**e**) and natural movies (**f**). Both dorsal and ventral cells had generally better LN model predictions if their nonlinearity asymmetries were close to zero. **g**, Firing rate responses (normalized) of dorsal (top) and ventral (bottom) transient-OFFα cells to a moving bar stimulus, averaged over eight

different directions. The bars had a positive (ON) contrast and approximately entered the receptive field of the cells at the start of the displayed traces and left the receptive field approximately at the time point marked by the dashed lines. The responses in the ventral retina showed a delayed peak following the onset of the bar. This peak may reflect different center-surround receptive field structures for transient-OFFα cells over the retinal surface. **h**, The relative strength of ON and OFF responses in **g** were measured with a response asymmetry index defined as $(R_{off} - R_{on})/(R_{off} + R_{on})$, where $R_{on}$ and $R_{off}$ are the average responses before and after the bar leaves the receptive field center (using time windows of 0.9 s, the time that it takes the bar to cross a point in space). This index was significantly larger for the dorsal retina (0.67 ± 0.31 vs. 0.00 ± 0.38, mean ± SD, $p < 10^{-23}$, two-sided Wilcoxon rank-sum test). **i**, Moving bar offset responses in the dorsal retina were more sustained compared to the ventral retina (0.26 ± 0.06 vs. 0.23 ± 0.22, mean ± SD, $p < 10^{-12}$, two-sided Wilcoxon rank-sum test). The sustained index was defined as the ratio of the average response over the maximum response in the time window (0.9 s) following the bar leaving the receptive field center. Data from 8 retina pieces.

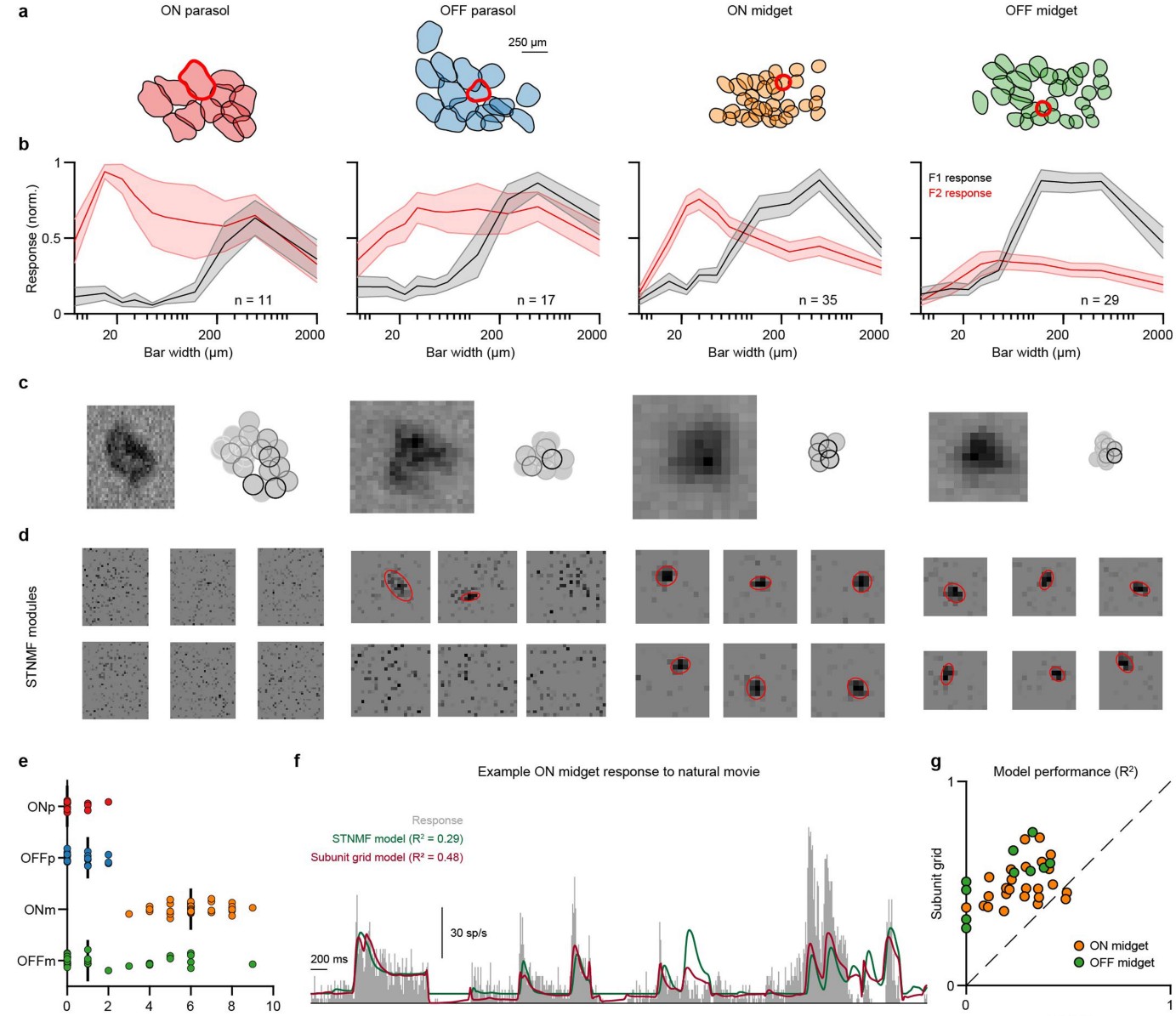

**Extended Data Fig. 9 | Method comparison with spike-triggered non-negative matrix factorization (STNMF). a**, Receptive-field mosaics from a single peripheral marmoset retina recording. Sample cells are marked with red outlines. **b**, Summary of responses to contrast-reversing gratings of different spatial frequencies (square-wave gratings of 100% contrast, reversal frequency 5 Hz, with one to eight equidistant spatial phases per bar width). Spatial frequency tuning curves for the first Fourier harmonic (F1; black) and the second Fourier harmonic (F2; red). F1 is calculated as the maximum and F2 as the mean harmonic amplitude of the responses of the cells over all spatial phases. The error bars represent the SEM. For all four types, the effect of a suppressive surround is clear as both F1 and F2 components decay with increasing bar width. Except for OFF midget cells, the spatial nonlinearity is evident in the strong F2 component for small stimulus scales. **c**, Spatial filter from white-noise responses of a sample cell for each cell type and the corresponding subunit layout, fitted with flickering gratings. Darker pixels in spatial filters denote larger (positive) values.

**d**, STNMF[28] applied to a one-hour-long recording of spatiotemporal white-noise stimulation with high spatial resolution. STNMF recovers subunits for midget cells, but here fails for parasol cells, likely owing to the large number of pixels included for these cells. **e**, Number of subunits recovered by STNMF for cells of all four types. Vertical black lines mark the medians. **f**, Using responses to the natural movie, we compared the prediction performance of the subunit grid model and a subunit model derived from STNMF. For the latter model, STNMF subunit outputs were rectified and summed to obtain generator signals. Summation weights were determined by fitting a linear combination of subunit filters to match the overall spatial filter, using non-negative least squares. Generator signals were then related to spiking responses by fitting a logistic output nonlinearity using the non-repeated part of the natural movie (as for the subunit grid model). **g**, Model performance comparison for midget cells between the two nonlinear subunit methods. Parasol cells were omitted because STNMF failed to recover meaningful subunit layouts.

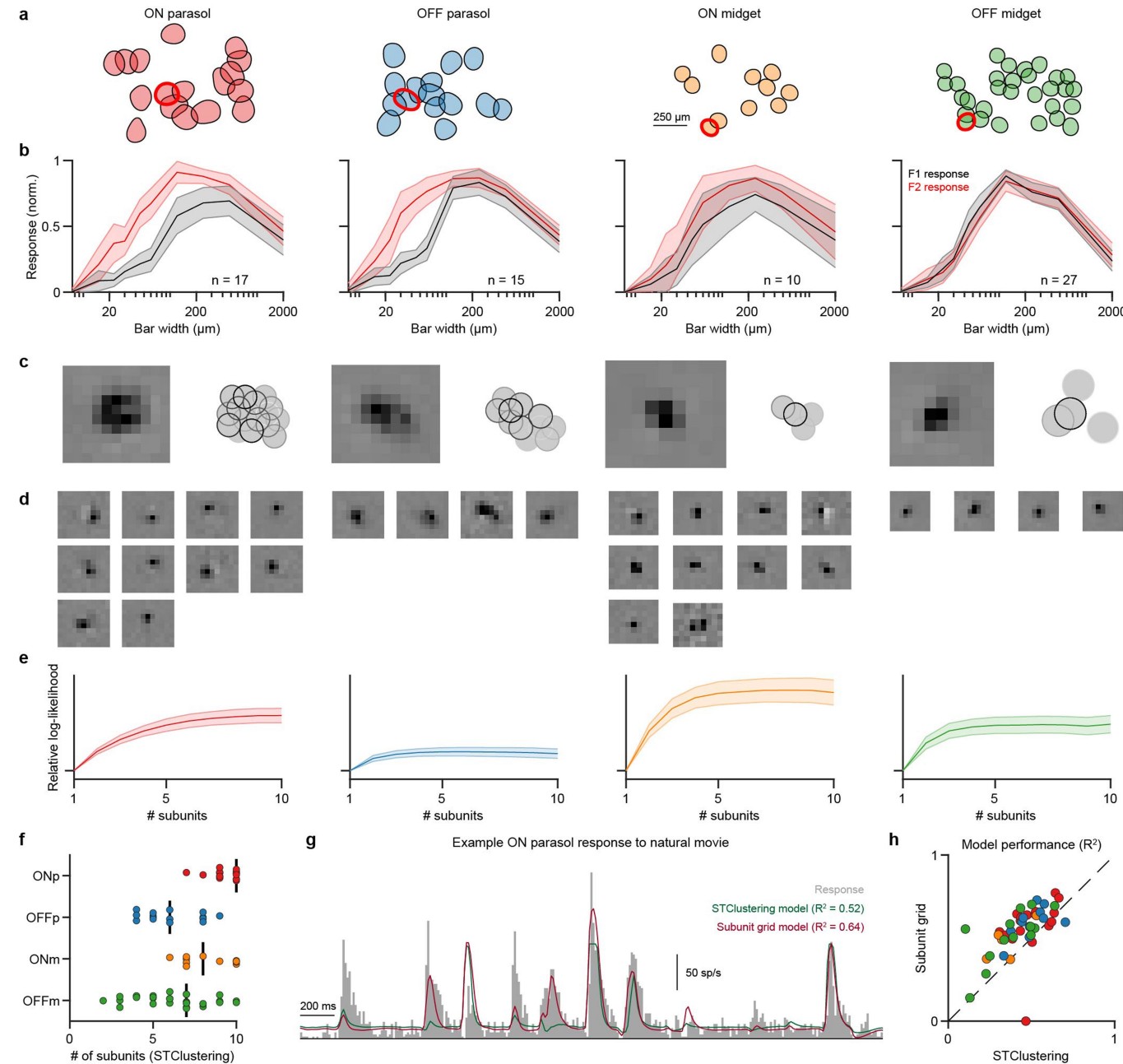

**Extended Data Fig. 10 | Method comparison with spike-triggered clustering (STClus). a**, Receptive-field mosaics from a single peripheral marmoset retina recording. Sample cells are marked with red outlines. **b**, Summary of cell type responses to contrast-reversing gratings. **c**, Spatial filter from white noise of a sample cell for each cell type and the corresponding nonlinear subunit layout obtained by the subunit grid method. Darker pixels in spatial filters denote larger (positive) values. **d**, Nonlinear subunits obtained by STClus[30] applied to white-noise responses. The selected number of subunits maximized the likelihood of a validation set. **e**, The log-likelihood for different numbers of subunits for all cells of the same type. Error bars are 95% confidence intervals.

**f**, Number of subunits that maximized the validation likelihood for each cell. Black bars are medians over cells belonging to the same type. **g**, We compared the prediction performance of the two models using the natural movie. STClus subunit outputs were exponentiated and summed to obtain generator signals. Summation weights were determined by the STClus fitting procedure using the white-noise data. Generator signals were then related to spiking responses by fitting a model-specific output nonlinearity[30], using the non-repeated part of the natural movie (as for the subunit grid model). **h**, Model performance comparison between the two nonlinear subunit methods.

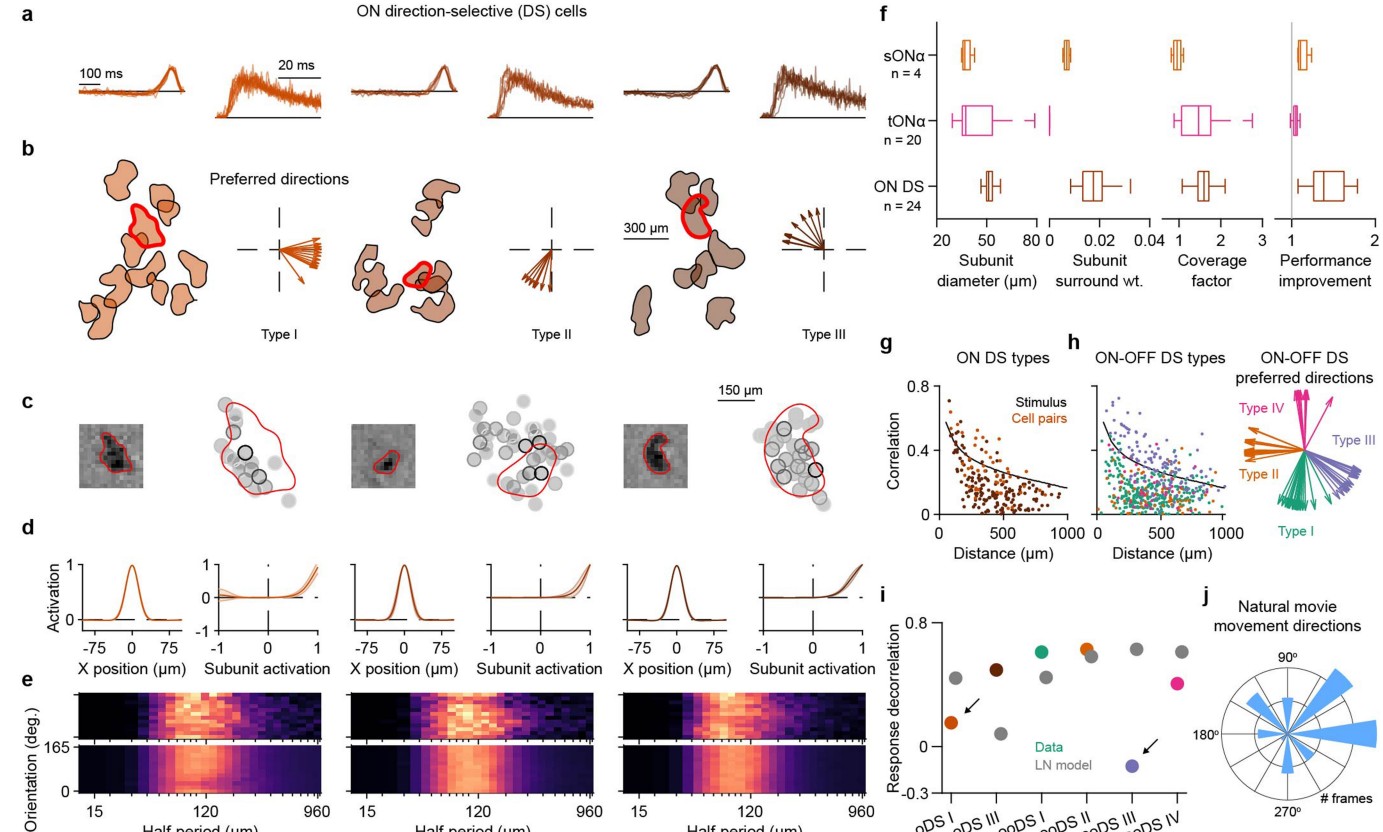

**Extended Data Fig. 11 | Direction-selective cells show correlated responses when driven by natural movies. a**, Temporal filters and autocorrelations of ON direction-selective (DS) cells from a single recording, separated into three groups according to preferred motion direction in response to a drifting grating stimulus. Temporal filters are typically monophasic and autocorrelations suggest sustained spiking responses. **b**, Receptive field contours and preferred motion directions of the three groups of ON DS cells. **c**, Spatial filters from white noise and subunit layouts obtained from the subunit grid model for three sample ON DS cells. Note that the subunit map only roughly matches the receptive-field contour. Darker pixels in spatial filters denote larger (positive) values. **d**, Average spatial profiles and nonlinearities of subunits for the three groups of ON DS cells, revealing strong rectification. Shaded error bars depict 95% confidence intervals. **e**, Tuning surfaces of the three ON DS cells from **c**, revealing strong suppression for large spatial scales. Colormap of the tuning surfaces as in Extended Data Fig. 6b. **f**, Comparison of model fits for ON DS cells with sustained- (sONα) and transient-ONα cells (tONα) from the same recording. Compared to the other two ON types, ON DS cells had larger subunit diameters,

stronger subunit surround weights (wt.), comparable coverage factors, and larger model performance improvement over DoG LN models for natural images. **g**, Pairwise correlations for ON DS cells from two subtypes (I and III from **b**) under natural movie stimulation. Cells from subtype II were excluded because they had unreliable responses to the movie. Each data point corresponds to a pair. The black line shows the correlation between stimulus pixels. **h**, Same as **g**, but for ON-OFF DS cell pairs (left). ON-OFF DS cells were clustered into four types based on their preferred directions in response to drifting gratings (right). **i**, For both ON DS (oDS) and ON-OFF DS (ooDS) cells, one subtype (type I from **b** and type III from **h**) showed particularly low decorrelation in our data (colored circles, arrows), compared to what would be predicted by an LN model (grey circles). These are cells with preferred directions approximately towards the right. **j**, Large gaze shifts (> 75 μm/frame) in our constructed natural movie typically caused global movement with a strong rightward component, approximately matching the preferred directions of cell types with relatively low decorrelation values. We hypothesize that this prevalence of motion in the preferred direction led to the increased correlations of these DS cell types.

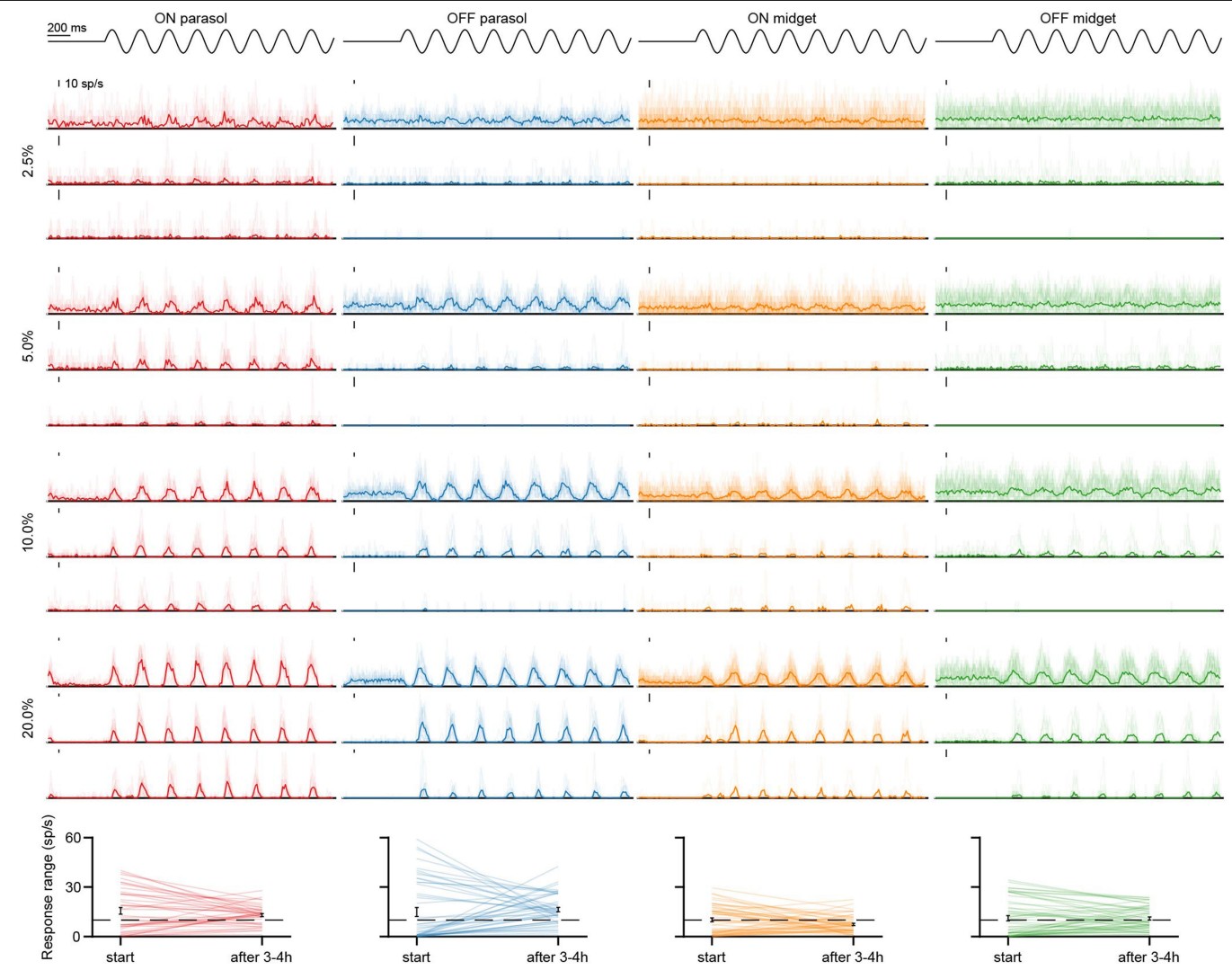

**Extended Data Fig. 12 | Contrast sensitivity of marmoset retinal ganglion cells.** Spike rate responses of individual cells (thin traces) to full-field sinusoidal modulation of light intensity (shown schematically on top) at 4 Hz and different contrast values (2.5 to 20%). For each contrast, data from three recordings (3 retina pieces) are shown separately. The thick solid lines mark the average responses over cells, and data come from the beginning of each recording. Bottom: Response range for each cell at the start of the recording vs. after 3 to 4 h. The response range was calculated as the difference between the maximal and minimal firing rate during the sinusoidal modulation at 5% contrast. Error bars are mean ± SEM. Dashed lines mark 10 spikes per second (sp/s).

# Reporting Summary

## Statistics

For all statistical analyses, confirm that the following items are present in the figure legend, table legend, main text, or Methods section.

| n/a | Confirmed | |
|---|---|---|
| ☐ | ☒ | The exact sample size (*n*) for each experimental group/condition, given as a discrete number and unit of measurement |
| ☐ | ☒ | A statement on whether measurements were taken from distinct samples or whether the same sample was measured repeatedly |
| ☐ | ☒ | The statistical test(s) used AND whether they are one- or two-sided<br>*Only common tests should be described solely by name; describe more complex techniques in the Methods section.* |
| ☒ | ☐ | A description of all covariates tested |
| ☒ | ☐ | A description of any assumptions or corrections, such as tests of normality and adjustment for multiple comparisons |
| ☐ | ☒ | A full description of the statistical parameters including central tendency (e.g. means) or other basic estimates (e.g. regression coefficient) AND variation (e.g. standard deviation) or associated estimates of uncertainty (e.g. confidence intervals) |
| ☐ | ☒ | For null hypothesis testing, the test statistic (e.g. *F*, *t*, *r*) with confidence intervals, effect sizes, degrees of freedom and *P* value noted<br>*Give P values as exact values whenever suitable.* |
| ☒ | ☐ | For Bayesian analysis, information on the choice of priors and Markov chain Monte Carlo settings |
| ☒ | ☐ | For hierarchical and complex designs, identification of the appropriate level for tests and full reporting of outcomes |
| ☐ | ☒ | Estimates of effect sizes (e.g. Cohen's *d*, Pearson's *r*), indicating how they were calculated |

*Our web collection on statistics for biologists contains articles on many of the points above.*

## Software and code

Policy information about availability of computer code

| Data collection | Extracellular voltage signals were acquired with Multichannel Systems amplifiers. Visual stimuli were generated and controlled through custom-made software, based on Visual C++ and OpenGL. Raw data files and stimulus generation code can be made available upon reasonable request to the corresponding author. |
|---|---|
| Data analysis | Spike sorting was performed using a modified version of Kilosort (Pachitariu et al., 2016), available at https://github.com/dimokaramanlis/KiloSortMEA, and curated with the phy software (https://github.com/cortex-lab/phy). All analyses and generation of figures were done with MATLAB (versions 9.10 to 9.12). Code used to analyze spiking data and fit computational models is available on GitHub: https://github.com/dimokaramanlis/subunit_grid_model. |

For manuscripts utilizing custom algorithms or software that are central to the research but not yet described in published literature, software must be made available to editors and reviewers. We strongly encourage code deposition in a community repository (e.g. GitHub). See the Nature Portfolio guidelines for submitting code & software for further information.

## Data

Policy information about availability of data

All manuscripts must include a data availability statement. This statement should provide the following information, where applicable:

- Accession codes, unique identifiers, or web links for publicly available datasets
- A description of any restrictions on data availability
- For clinical datasets or third party data, please ensure that the statement adheres to our policy

> All spike-sorted data used for this study are available at G-Node: https://doi.org/10.12751/g-node.ejk8kx (doi: 10.12751/g-node.ejk8kx). The applied natural images from the van Hateren database are available at https://pirsquared.org/research/vhatdb/full/.

## Research involving human participants, their data, or biological material

Policy information about studies with human participants or human data. See also policy information about sex, gender (identity/presentation), and sexual orientation and race, ethnicity and racism.

| | |
|---|---|
| Reporting on sex and gender | N/A |
| Reporting on race, ethnicity, or other socially relevant groupings | N/A |
| Population characteristics | N/A |
| Recruitment | N/A |
| Ethics oversight | N/A |

Note that full information on the approval of the study protocol must also be provided in the manuscript.

# Field-specific reporting

Please select the one below that is the best fit for your research. If you are not sure, read the appropriate sections before making your selection.

☒ Life sciences ☐ Behavioural & social sciences ☐ Ecological, evolutionary & environmental sciences

For a reference copy of the document with all sections, see nature.com/documents/nr-reporting-summary-flat.pdf

# Life sciences study design

All studies must disclose on these points even when the disclosure is negative.

| | |
|---|---|
| Sample size | No sample size calculations were used for this study. The sample size (here at least 3 retinas per species; each retina yields an unpredictable, but large number of recorded cells, typically several tens to hundreds) is consistent with the standards in the field for retinal multielectrode-array recordings and is comparable to sample sizes reported in similar publications (e.g., Roy et al 2021, Nature 492:409-413; Shah et al 2020, eLife 9:e45743). The retinal response properties investigated in this study were consistent across different recordings, and the sample size was sufficient to demonstrate the repeatability of the effects observed in both marmoset monkeys and mice. |
| Data exclusions | For all stimulus-specific population analyses, we excluded individual cells which did not reliably respond to the corresponding stimulus. Specific exclusion criteria are reported in the manuscript. For marmoset tissue, we only used retinas for which a 5% contrast full-field modulation at 4 Hz produced at least a 10 spikes/s modulation in the average ON parasol spike rate at the beginning of the recording. |
| Replication | All measurements in the study were performed on multiple cells of each type in multiple animals; cell numbers are reported in each of the relevant figure legends. There were no unsuccessful replication attempts; results were consistent across all recordings, and alls datasets that passed the reliability criterion stated above under Data exclusions were included in the final analysis. |
| Randomization | The study did not involve any traditional experimental groups and thus there was no requirement for randomization. |
| Blinding | The study did not involve traditional experimental groups that could be blinded. Data from all retinal ganglion cells were analyzed with the same code without selection. The researchers were blind to any group-dependent bias during data collection, because of the laborious offline analyses required for grouping cells into types. |

# Reporting for specific materials, systems and methods

We require information from authors about some types of materials, experimental systems and methods used in many studies. Here, indicate whether each material, system or method listed is relevant to your study. If you are not sure if a list item applies to your research, read the appropriate section before selecting a response.

## Materials & experimental systems

| n/a | Involved in the study |
|-----|----------------------|
| ☒ | ☐ Antibodies |
| ☒ | ☐ Eukaryotic cell lines |
| ☒ | ☐ Palaeontology and archaeology |
| ☐ | ☒ Animals and other organisms |
| ☒ | ☐ Clinical data |
| ☒ | ☐ Dual use research of concern |
| ☒ | ☐ Plants |

## Methods

| n/a | Involved in the study |
|-----|----------------------|
| ☒ | ☐ ChIP-seq |
| ☒ | ☐ Flow cytometry |
| ☒ | ☐ MRI-based neuroimaging |

## Animals and other research organisms

Policy information about <u>studies involving animals</u>; <u>ARRIVE guidelines</u> recommended for reporting animal research, and <u>Sex and Gender in Research</u>

| Laboratory animals | The study was performed with retinal tissue obtained from adult male marmoset monkeys (Callithrix jacchus), aged 12-18 years, and wild-type female mice (C57BL/6J), aged 7-23 weeks. Mice were housed in a 12-hour light/dark cycle.The ambient conditions in the animal housing room were kept at around 21°C (20–24°C) temperature and near 50% (45–65%) humidity. |
|---|---|
| Wild animals | The study did not involve wild animals. |
| Reporting on sex | The study did not involve sex-based analyses, because previous literature suggests that the electrophysiological properties of retinal tissue is relatively homogeneous between sexes. |
| Field-collected samples | The study did not involve collection of samples from the field. |
| Ethics oversight | Experimental procedures were in accordance with national and institutional guidelines and approved by the institutional animal care committee of the University Medical Center Göttingen, the German Primate Center and by the responsible regional government office (Niedersächsisches Landesamt für Verbraucherschutz und Lebensmittelsicherheit, permit number 33.19-42502-04-20/3458). |

Note that full information on the approval of the study protocol must also be provided in the manuscript.

