## [Peer Review File · Nature]

Nonlinear receptive fields evoke redundant retinal coding of natural scenes

Corresponding Author: Professor Tim Gollisch

A version of this paper was originally rejected for publication by Nature, however that decision was reconsidered after appeal by the authors.

Version 0:

Reviewer comments:

Referee #1

(Remarks to the Author)

The study by Karamanlis and Colleagues investigates the nonlinear receptive field structure of retinal ganglion cells (RGCs) in the marmoset and mouse retina. In particular, they look at how gaze shifts on natural scenes influence the pairwise correlations present in nearby RGCs. They observed relatively high pairwise correlations that were largely present when high spatial frequency and high contrast content was present in the receptive fields of nearby RGCs. They observe these correlations are not consistent with predictions from an linear-nonlinear (LN) model for some RGC types, like ON Parasol cells. They develop a very nice modeling framework that incorporates a grid of subunits that introduce spatially localized rectification within the receptive fields. This model is able to explain much about how the RGCs are responding during the mimicked saccades around natural scenes. Finally, the authors use the observation of high correlations to argue that efficient coding theory can't be at play for at least some ganglion cells types that have these receptive field subunits and that are signaling 'features' to the brain.

The paper is technically sound, and the modeling effort to explain the nonlinear RGC responses is impressive. However, in my view the way the study is framed is conceptually flawed and builds what I view as a straw man argument against efficient coding theory that is misleading. The basic claim by the authors is that because efficient coding theory postulates low correlations and the authors observe high correlations, that efficient coding theory can't be correct. However, this misses the other half of what efficient coding theory postulates, which is that the system is trying to signal as much information as possible about natural scenes. Thus, efficient coding theory is balancing two objectives, reducing redundancy while signaling as much information as possible in the presence of noise. Because the authors make no measurements of noise or of information, it is unclear to me how they can conclude anything about how efficient coding theory does or does not apply in the case of saccades and natural scenes. Furthermore, the authors are equating correlated activity between two cells for redundancy. I don't think this is justified. Efficient coding theory doesn't say anything about correlations specifically, it says redundancy should be reduced. If two cells are communicating something useful about the scene when they both spike (for example, that a high frequency texture extends into the receptive fields of both cells) then the correlated activity would be fine.

I agree with the authors that efficient coding theory applied to an LN model — e.g. as was done in the Karklin and Simoncelli 2011 paper — cannot explain their results. But one must then ask, is that a problem with the theory or with the model that was optimized under the theory. The authors clearly show (as many labs dating back to Hochstein and Shapley 1974) that an purely LN model isn't sufficient to explain RGC responses. So the real enterprise that must be engaged with to decide if efficient coding theory is or is not obeyed, is to optimize the subunit grid model or some other LN-LN cascade under efficient coding theory and see what the prediction is. The authors haven't presented anything remotely close to this and thus I don't believe they can conclude anything about efficient coding theory and its relevance to this nonlinear system.

What is left in the paper is a nice modeling study showing that the subunit grid model can capture the responses of RGCs in this context of saccades around a natural scene. However, several other modeling efforts have been similarly developed both by the Gollisch Lab and other labs (e.g., Rieke Lab, Chichilnisky Lab, Butts Lab, Ganguli Lab) in recent years to develop similar nonlinear subunit models (Freedland and Rieke 2022, Shah et al, 2020, Liu et al, 2022, Liu et al, 2017,

McFarland et al 2013, Maheswaranathan et al 2018), and it isn't clear that this model dramatically outperforms those others. Thus, it is difficult for me to see how the study would be of sufficiently broad interest to be published in Nature.

Referee #2

(Remarks to the Author)

This manuscript entitled "Correlated nonlinear responses driven by natural gaze shifts counteract efficient coding in the retina" by Karamanlis et al examines the impact of gaze shifts on retinal coding while using natural visual stimuli. The authors setup the problem as a conflict between the efficient coding theory, which predicts that the retina discards spatiotemporal correlations in natural scenes, and their novel findings showing strong stimulus-dependent correlations on primate and mouse retinas under naturalistic stimuli with gaze shifts. The authors include the impact of gaze shifts in stimulating the retinas, while recording population spike codes of distinct ganglion cell types on mouse and monkey retinas and show that distinct ganglion cell types disrupt efficient coding theory in these stimulus conditions by showing strong stimulus correlations. The authors further show that the observed correlations and disruptions of efficient coding theory can be explained by a model where nonlinear subunits are pooled in particular ganglion cell type inputs.

This is a well-designed study with a clear rationale. The methodology is sound, and the recordings are of high quality and the data analysis is solid. All major claims are well supported by the data. Particularly impressive is the clear demonstration that retinal subunits are activated by gaze shifts in cell-type-specific manner in some of the major ganglion cell classes of the primate and mouse retinas. These findings have major implications in understanding the retinal population codes in the vertebrate retina under naturalistic stimulus conditions. This is a fundamental contribution in understanding how the vision works. Furthermore, this paper gives great insights in understanding how nonlinear neural circuit elements can enhance stimulus-driven correlations. I have one major comment and several minor ones.

Major:

The authors show that ON parasols on marmoset retinas and ON-sustained alpha RGCs on mouse retinas show particularly strongly the stimulus-dependent correlations under gaze shifts. The stimulus conditions used in this study (for mouse ~4000 R*/rod/s) line up with previous findings (Schwartz et al, Nat. Neurosci, 2012), where subunit models seem to be important for explaining ON-sustained alpha RGC coding around these light levels. However, previous studies have also reported (Grimes, Schwarz & Rieke, Neuron, 2014) that the existence of these subunits in ON-sustained alpha RGCs depends strongly on the background light levels such that ON-sustained alpha RGCs follow linear coding scheme at scotopic light levels. While it would be unreasonable to request repeating these measurements across light levels, it would, however, be at least important to discuss in the paper how these previous findings on the background-light level dependency of subunit appearance in ON-sustained alpha cells fit into functional speculations that the authors present in their discussion section.

Minor

Fig.1B and related text: Natural movies consist of each image being presented for 1 s and displaced in x and y directions. Can you justify how "natural" this image is? It would be important to understand what underlies the selection of these stimulus parameters.

Fig 1F, K, left: Is the positive correlation significant for ON-parasols and ON-OFF parasols still at retinal distances up to 800 micrometers? How does this match with the spatial extent of the neighboring ganglion cell RFs and/or does this involve spatial mechanisms extending beyond neighboring RGC receptive fields?

P4: It would be good to briefly justify the selection of ganglion cell types selected for this study for a general reader and particularly introduce the similarity of ON and OFF parasols vs. mouse ON and OFF sustained alpha cells. This is particularly interesting taken into account how both ON parasols and mouse ON-sustained alpha RGCs show strong correlations and have been previously considered to be homologous cell types.

P4, line 110: "..On-alpha as well as transient-OFF-alpha": Does this mean On-sustained alpha cells in this case?

Fig 2E: The mosaics of ON parasols are more complete in the example as for the OFF parasols. What is the implication for this on the deduced results?

Fig 3: How well is the spatial extent of the subunits in line with the previous findings and/or morphological assumptions of the bipolar cell RFs?

p 15, line 489: Specify the wavelength of the IR light.

p 16, Visual Stimulation: What was the overall level of visual pigment bleaching (maximum level) in these recording conditions taken into account the recording duration. Were there any signs of light adaptation and/or how was this controlled in these recordings?

p 17: How complete were the obtained cell-type specific mosaics? How does the completeness of the mosaic impact the results particularly related to correlation analyses. Main Figs of the paper seem to indicate differences in mosaic completeness across cell types.

Referee #3

(Remarks to the Author)

In this study, Karamanlis and colleagues measure the responses of populations of retinal ganglion cells (RGC) to natural movies. Intriguingly, they find that certain cell classes On and Off cells are correlated with one another. The authors hypothesize that this behavior occurs from nonlinear subunits within the RGC receptive fields; they fit a subunit model to capture the size and nonlinearities of subunits and find that the unexpected behavior of the cells to natural scenes correlates to the degree of subunit nonlinearity. Further, they find that On cell subunits are more nonlinear than Off cell subunits. Overall, these findings are interesting and important for understanding natural vision, but I have several concerns about inconsistencies in the results relative to previous studies and how the current study relates to previously published work.

Major concerns:

1) Significant discrepancies in findings relative to previous studies.

The subunit model used to find the ganglion cell responses in marmoset found that On parasol and midget cell subunits were more rectified (i.e., nonlinear) than Off parasol/midget cells at photopic backgrounds (Fig. 2). Yet, several studies in primate retina, including MEA work and direct measurements of the subunit nonlinearities using voltage clamp recordings have repeatedly found the opposite to be the case – at these light levels, Off parasol/midget cell subunits (bipolar cells) are significantly more rectified than their On counterparts (Chichilnisky and Kalmar, 2001; Turner and Rieke; Manookin; Appleby; Sinha). Further, this finding has been replicated across several vertebrate and invertebrate species, and normative studies have provided a theoretical explanation for why this On/Off asymmetry ought to be the case across taxonomic groups.

– In lines 156-159, the authors point out this discrepancy, but no explanation is given for why the current study deviates so significantly from previous work.

– In Fig. 2, the On midget subunit size is bigger than Off parasol subunits. However, this does not agree with published measurements of bipolar cells in the mid-peripheral/peripheral primate retina. Anatomical studies have shown that, at any given eccentricity, diffuse bipolar dendritic trees are significantly larger than midget bipolar dendritic trees (Boycott and Wässle, 1991; Dacey et al., 2000; Tsukamoto and Omi, 2015, 2016). This discrepancy likely indicates that the subunit model fit in the current study poorly estimates the subunit properties, including their size and nonlinear properties (see above).

– Lines 252:255: The current work indicates that spatial contrast dependence was largest in On parasol cells, but previous work has shown that Off parasols have the greatest sensitivity to spatial contrast both to artificial and naturalistic inputs (Turner and Rieke, 2016; Turner et al., 2018; Appleby and Manookin, 2020).

This discrepancy with previous studies could arise from at least two sources: 1) Imbalances in the activity in the On and Off pathways due to issues with tissue health and/or 2) excessive free parameters in the subunit model that cause convergence on unrealistic values. Tracking this down will be critical to the overall impact of the study.

2) Acknowledgment of previous work/larger context of the current study

In my reading, the major findings of the current study can be summarized as follows:

A) Contrary to the predictions of efficient coding the spike outputs of On and Off parasol cells are strongly correlated to naturalistic visual inputs. This correlation occurs both within and across types. On/Off midget cells, however, are anti-correlated on local spatial scales and decorrelated on wider spatial scales, which is consistent with the predictions of efficient coding (Fig. 1).

This finding that the retinal output is strongly correlated/redundant has been shown previously in several studies including in primates (Schnitzer and Meister, 2003; Schneidman et al., 2006; Shlens et al., 2009; Trong and Rieke, 2008; Ala Laurila et al., 2011; and others...). Further, the previous literature has identified several functional roles for such correlations. The current study would benefit greatly from properly acknowledging this previous work and explaining how it contributes to our understanding of these correlations in visual processing.

B) This break from efficient coding arises from and can be fully explained by nonlinear subunits within the receptive field.

The observation that On and Off parasol cells are correlated with each other to natural inputs is an interesting and important contribution to the field. However, as mentioned above, how does this finding fit into the larger picture? The current work intimates broadly at the functional consequences for this, but having this spelled out more explicitly and placed in the general context of our current understanding would be very helpful.

Version 2:

Reviewer comments:

Referee #1

(Remarks to the Author)

The authors have clearly taken the initial round of reviews seriously and the manuscript has improved substantially. However, I still think there are ambiguities in interpretation and I ultimately do not think the authors have demonstrated the main conclusions of the paper: nonlinear processing counteracts efficient coding. The continuing disagreement here is about the extent to which correlations (or lack of decorrelation) indicates violations of efficient coding theory. The authors claim it does, but I disagree.

The Authors state on line 98 “Decorrelation is a strong prediction of the efficient coding hypothesis in the case of noiseless transmission channels (Atick and Redlich, 1992; Barlow, 1961).” In my view, this statement is incomplete. The decorrelation prediction also assumes an approximately linear encoding model (e.g., an LNP model). This is actually clearly articulated in the Atick and Redlich (1992) study that the authors cite, see just before and after equation 1.2 in that manuscript. Thus, a different encoding model can make different predictions with respect to decorrelations. For example, one difference in the predictions made by the Karklin and Simoncelli model from that of the Atick and Redlich model is how receptive field structure depends on noise. In a purely linear model optimized to efficient coding (Atick and Redlich), changing the noise causes a change in receptive field structure (center-only under high noise and center-surround under low noise). However, when one introduces a rectifying nonlinearity and then optimizes according to the efficient coding, the prediction changes. The surround doesn’t go away at low light levels (high noise conditions); instead the center AND surround both increase their diameters so that the population of cells have greater overlap. The two models (optimized according to efficient coding theory) thus make different predictions about the extent and manner of decorrelation. This illustrates the point that the specific encoding model under consideration can have a major impact on what efficient coding theory predicts.

The authors are implicitly assuming a pseudo-linear encoding model when comparing retinal responses to efficient coding predictions. Prior work shows a pseudo-linear model holds better, in general, for midget ganglion cells than for parasol ganglion cells. And the authors observe better agreement with the Atick and Redlich model for midget cells (more linear) than for parasol cells (less linear). In my view, this pretty strongly suggests this is a flaw in the model, not a flaw in the theory that is optimizing the model.

Here is an illustration of the point I’m trying to make. The Atick and Redlich work doesn’t predict the presence of ON and OFF cells. So one could take their work and compare it to the retina and say that efficient coding theory is wrong because the theory doesn’t predict ON and OFF cells. But then Karklin and Simoncelli come around and accounted for the rectifying nonlinearity in spike generation in a linear-nonlinear model and then optimized that new model according to the theory. Suddenly, optimizing a more accurate model of retinal encoding produces ON and OFF cells, mosaics, the slight numerical dominance of OFF cells over ON cells, and a few other features of retinal processing. But of course, it doesn’t predict (yet) midget and parasol cells, or color opponent cells. So, is efficient coding theory wrong or right? Given the work on elaborating encoding models and the stimulus sets used to train those models (Karklin and Simoncelli 2011; Ocko, Lindsey, Ganguli, and Deny, 2018; Jun, Field and Pearson, 2022), it seems much more likely that the divergence of the results in this study from prior efficient coding predictions has more to do with overly interpreting ‘decorrelation’ as a specific signature of efficient coding theory, when in fact it is just a prediction from efficient coding theory under a linear model that we already know is wrong.

Thus, I continue to view this study as not challenging the idea of efficient coding theory, but a case where the authors are exploring conditions under which these nonlinearities of parasol cell signaling are amplified — i.e. greater deviations from the predictions of an LNP model. I suspect one major reason for this is the known subunits present in the receptive fields of parasol cells. These subunits are not accounted for in Atick and Redlich or the Karklin and Simoncelli models, hence these models cannot be trusted to make accurate predictions about how the parasol cells should respond to these stimuli nor do they make accurate prediction with respect to efficient coding.

The authors also create some straw-man models in their manuscript, that I think are misleading. For example, on line 67, “Furthermore, when considering pairs of either parasol or midget cells with opposing contrast preference, that is, an ON and an OFF cell, one might expect these pairs to be negatively correlated, as one cell should respond to increases and the other to decreases in light intensity.” Again, this statement assumes a linear model. If the ON and OFF cells do not exhibit subunit rectification, then the above statement is true. But we know parasol cells exhibit subunit rectification and so do peripheral midget RGCs to a lesser extent. Thus, anytime a natural image with high spatial frequency content (e.g., a high contrast edge) falls in the receptive field of overlapping ON and OFF parasol cells, they will both fire action potentials because their encoding is not linear. This has been known since the 1970s. None of this is a violation of efficient coding theory per se, but it is a violation of linear models.

The authors also added an explicit computation of information rates and redundancy (Fig S3 and Figs 11-J). I liked this analysis, but I still think it suffers from some issues. There is pretty clearly redundancy across cells, as the authors indicate. The question is, ‘does this violate efficient coding theory?’ It isn’t actually clear to me if it does or doesn’t in this case for a couple of reasons. First, efficient coding theory isn’t just about redundancy reduction, as the authors state, it is about maximizing information transmission too, in the face of noise. Thus, the question becomes, ‘could the cumulative information rates across all cell types be retained if the redundancies were reduced?’ Clearly, in a noiseless system, the answer would be yes, but when noise is spread across the system and common to multiple channels, it is less clear. Second, the

illustration by the authors that ON parasol cells are the most reliable and yet still exhibit high levels of redundancy has the issue that the ON parasol cells are not acting independently of the other cell types in the system and thus perhaps redundancy in their signals is helping to compensate for noise in OFF parasol or midget cell signals. Admittedly, these are very difficult issues to untangle, but given these difficulties in interpretation, I think the manuscript comes down too hard on the side against efficient coding.

A more conservative interpretation of these results (that I think is also more accurate) is that this study demonstrates some combination of the following — 1) the retina is not well described by pseudo-linear models; 2.) redundancy reduction does not appear to be the primary goal of retinal signaling, and 3) observation 2 could be because efficient coding theory is inadequate to describe retinal coding OR because sufficiently accurate models of retinal processing (e.g., models that include subunit rectification) that also include the magnitude and sources of retinal noise (e.g., input, output, and correlated noise), have not been optimized according efficient coding theory. Thus, we don't have a strong comparison between the predictions of efficient coding theory when applied to circuits of similar structure to the retina and the actual signaling performed by the retina. This tension is demonstrated in the title of the manuscript — paraphrasing: "Correlated nonlinear responses counteract efficient coding." But this apparent "counteracting" is (in my view) more likely to be a result of exploring efficient coding in terms of linear instead of nonlinear models.

In summary, I think this is a high quality study, but it is not entirely decisive in what can be concluded. This, to some extent, diminishes the impact. The work is also of sufficient complexity, that it would likely benefit from some more space. Reading it at times feels like topics are being oversimplified to manage the space constraints.

Referee #2

(Remarks to the Author)

The authors have addressed sufficiently all my previous concerns. The already technically sound and interesting manuscript has now gained significant additional presentation power and depth. Overall, the manuscript has improved significantly in revision particularly related to its framing. In my view, the authors have done an excellent job in improving the presentation of the paper in relation to the efficient coding theory and in bringing up the novelty aspects of their study in reference to the previous literature. In addition, important improvements have been made in addressing the possibility to assess the tissue condition as well as e.g. more clearly relating the spatial scale of the modeled subunits to the morphological dimensions of the cone bipolar subunits. Particularly useful aspect of the study is also the comparison of the findings on the primate retina to those on the mouse retina, as a model system. I have no additional comments except of a minor note related to the methods section. It would be important to state clearly the assumptions underlying the conversion of light intensities into isomerization rates (i.e. state clearly the assumed parameter values, particularly the photoreceptor collecting area parameters) such that the stimulus conditions can be reproduced in the future studies.

Referee #3

(Remarks to the Author)

The authors have adequately addressed my previous concerns. This is a very nice study.

Author Rebuttals to Initial Comments:

We thank the Reviewers for their thorough and constructive comments. We have addressed all aspects raised by the Reviewers as detailed in the point-by-point responses below. In particular, among other changes to the manuscript, we have added several new analyses for directly computing redundancy between cell pairs, for assessing response reliability, and for demonstrating quality and stability of the recorded data. We believe that the additions and alterations of the presented material have substantially strengthened and clarified the findings of our manuscript.

Referee #1:

The paper is technically sound, and the modeling effort to explain the nonlinear RGC responses is impressive. However, in my view the way the study is framed is conceptually flawed and builds what I view as a straw man argument against efficient coding theory that is misleading. The basic claim by the authors is that because efficient coding theory postulates low correlations and the authors observe high correlations, that efficient coding theory can't be correct. However, this misses the other half of what efficient coding theory postulates, which is that the system is trying to signal as much information as possible about natural scenes. Thus, efficient coding theory is balancing two objectives, reducing redundancy while signaling as much information as possible in the presence of noise. Because the authors make no measurements of noise or of information, it is unclear to me how they can conclude anything about how efficient coding theory does or does not apply in the case of saccades and natural scenes. Furthermore, the authors are equating correlated activity between two cells for redundancy. I don't think this is justified. Efficient coding theory doesn't say anything about correlations specifically, it says redundancy should be reduced. If two cells are communicating something useful about the scene when they both spike (for example, that a high frequency texture extends into the receptive fields of both cells) then the correlated activity would be fine.

We agree with the Reviewer that the connections between the measured neural correlations and the violation of efficient coding had not been sufficiently substantiated in the previous manuscript version. We thank the Reviewer for pointing out this disconnect. In the revision, we have rectified this in several ways. First, we more clearly motivate the analysis of activity correlations between cells through the often-considered hypothesis that center-surround receptive fields decorrelate signals in the encoding of natural stimuli. This is often taken as a proxy for redundancy reduction, and indeed one can expect a certain level of decorrelation toward redundancy reduction (Atick and Redlich 1992), at least to the point where noise becomes relevant. Our finding, however, is that some cell types display hardly any decorrelation compared to the input signal and that there is a cell-type dependence of decorrelation that is not consistent with general redundancy reduction.

Second, to more directly assess the nature of the correlations and their relation to noise, we now 1) compare the level of noise in the responses of different cell types and find that the most correlated ones (in particular ON parasol cells) are also the most reliable ones (cf. lines 98-109), excluding that correlations are there to counteract noise, and 2) quantify noise correlations between cells and find these to be minor compared to stimulus-induced correlations (cf. new Figure S2 and lines 110-118), thus excluding the possibility that correlations contribute to information transmission through an intrinsic (synergistic) population coding scheme (Schneidman et al 2003). Note also that our measure of correlations in the main analysis is based on firing rates and thus independent of noise correlations, as now pointed out more clearly (lines 116-118).

Third, and most importantly, we now explicitly compute information rates and redundancy measures for cell pairs and compare them across types. Despite the technical difficulties often associated with computing information rates, especially when going beyond single cells, we here found that reliable estimates can be obtained by an approach that assesses activity patterns in Fourier space. These

analyses clearly show that redundancy between cell pairs is cell-type specific in the same way as correlations and can be substantial, in particular, for ON parasol cells in the marmoset and transient OFF α cells in the mouse. Moreover, we find that the level of correlations is strongly indicative of the level of redundancy for a cell pair of a given type (e.g., new Fig. 1J), thus confirming correlations as a good proxy for redundancy in the present context.

Together, the new analyses strongly corroborate our original conclusion: that redundancy is cell-type-specific in the retina, that it is substantial for certain (correlated, nonlinear) cell types, and that the efficient coding hypothesis is thus partially violated. Note that, for this conclusion and for our analysis, it is not relevant whether the cells encode, for example, luminance or texture information. Redundancy is computed independently of stimulus information, and failure to reduce redundancy stemming from correlated texture signals in the stimulus is a violation of efficient coding in the same way as a failure to reduce redundancy in luminance signals. This is now explained in more detail in the Discussion (lines 392-400).

I agree with the authors that efficient coding theory applied to an LN model — e.g. as was done in the Karklin and Simoncelli 2011 paper — cannot explain their results. But one must then ask, is that a problem with the theory or with the model that was optimized under the theory. The authors clearly show (as many labs dating back to Hochstein and Shapley 1974) that a purely LN model isn't sufficient to explain RGC responses. So the real enterprise that must be engaged with to decide if efficient coding theory is or is not obeyed, is to optimize the subunit grid model or some other LN-LN cascade under efficient coding theory and see what the prediction is. The authors haven't presented anything remotely close to this and thus I don't believe they can conclude anything about efficient coding theory and its relevance to this nonlinear system.

We thank the Reviewer for the suggestion. We agree that optimizing subunit models under an efficient coding objective would be an interesting endeavor. However, this would be a rather different research question from the one asked in our work. Our goal is not to explore models under the assumption of efficient coding, but rather probe efficient coding in a model-independent way. Our finding of strong correlations and high redundancy is independent of whether cells can be described by an LN or a subunit model. Subsequently, our model explorations are used to assess which functional features contribute to the observed redundancy. To clarify this, we now take up the question of subunit models that would be optimal for efficient coding in the Discussion and relate this to our findings (lines 400-405).

What is left in the paper is a nice modeling study showing that the subunit grid model can capture the responses of RGCs in this context of saccades around a natural scene. However, several other modeling efforts have been similarly developed both by the Gollisch Lab and other labs (e.g., Rieke Lab, Chichilnisky Lab, Butts Lab, Ganguli Lab) in recent years to develop similar nonlinear subunit models (Freedland and Rieke 2022, Shah et al, 2020, Liu et al, 2022, Liu et al, 2017, McFarland et al 2013, Maheswaranathan et al 2018), and it isn't clear that this model dramatically outperforms those others. Thus, it is difficult for me to see how the study would be of sufficiently broad interest to be publish in Nature.

As pointed out above, our goal was not to develop a model for the purpose of better performance, but to be able to connect it to the analysis of natural stimuli. Most previous works face serious roadblocks here, for example, because they are not (yet) applicable to fully extract two-dimensional spatial subunit layouts (McFarland et al 2013, Maheswaranathan et al 2018) or because they need to apply finely structured white-noise stimuli requires long recordings that are difficult to combine with long data acquisition under natural movies (Shah et al 2020, Liu et al 2017) or because of the lack of temporal integration (Liu et al 2022, Freedland and Rieke 2022). Our method thus goes beyond these approaches

by allowing model fits with full two-dimensional subunit layouts and temporal kernels in reasonable experimental time. Moreover, we do demonstrate superior performance relative to comparable previous approaches in two supplementary figures (Figs. S10 and S11). Also, our inclusion of surround effects in the subunits goes beyond what has so far been considered for subunit models, and the evaluation on natural movies (which had not been attempted in most of the previous works) underscores the importance of this model feature. Thus, we believe that our model does indeed outperform and, more importantly, conceptually advances over previous work. In particular, it allowed us to dissect the contributions to neuronal correlations, which was the primary goal of our endeavor.

Referee #2:

The authors show that ON parasols on marmoset retinas and ON-sustained alpha RGCs on mouse retinas show particularly strongly the stimulus-dependent correlations under gaze shifts. The stimulus conditions used in this study (for mouse $\sim 4000 R^/rod/s$) line up with previous findings (Schwartz et al, Nat. Neurosci, 2012), where subunit models seems to be important for explaining ON-sustained alpha RGC coding around these light levels. However, previous studies have also reported (Grimes, Schwarz & Rieke, Neuron, 2014) that the existence of these subunits in ON-sustained alpha RGCs depends strongly on the background light levels such that ON-sustained alpha RGCs follow linear coding scheme at scotopic light levels. While it would be unreasonable to request repeating these measurements across light levels, it would, however, be at least important to discuss in the paper how these previous findings on the background-light level dependency of subunit appearance in ON-sustained alpha cells fit into to functional speculations that the authors present in their discussion section.*

Thank you for the suggestion. We chose the average light level for this study in the low photopic range, which may represent a good midpoint between driving the nonlinear receptive field and retaining light responsivity throughout our hours-long recordings. Thus, our findings do not represent effects at an extreme range of light intensities, but fall in the physiological range of expected natural light-intensity exposure. Nonetheless, we agree that assessing the effect of light intensity on the correlations would be an interesting endeavor, which is, however, beyond the range of the current work. As suggested, we have therefore taken up this question in the Discussion (lines 440-444), where we also speculate that effects of decreased spatial nonlinearities may be counterbalanced by increased noise correlations under scotopic conditions (Ruda et al 2020).

Fig.1B and related text: Natural movies consist of each image being presented for 1 s and displaced in x and y directions. Can you justify how "natural" this image is? It would be important to understand what underlies the selection of these stimulus parameters.

We have added explanatory text (lines 38-49) to emphasize that a particular focus of this study is natural gaze shifts. Briefly, the goal was to present a variety of visual scenes to provide for variability in stimulus context and activity patterns of the neurons while also containing natural gaze dynamics. The one-second presentations of individual images provide a good compromise, as they allow us to apply a fairly large number of different images together with extended sequences of natural (actually measured) gaze shifts. Note that a typical 1-s image presentation for the recordings from the marmoset retina contain several saccades (plus fixational eye movements) and that the neurons respond strongly and reliably to these gaze dynamics (e.g., Fig. 1A-D). The images themselves were mostly taken from the van Hateren database, which is something like a standard choice for natural (monochromatic) images. For the marmoset, a few indoor scenes were included that had previously been used in a vision experiment. All the applied images used can be viewed in the accompanying repository supplied by us.

Fig 1F, K, left: Is the positive correlation significant for ON-parasols and ON-OFF parasols still at retinal distances up to 800 micrometers? How does this match with the spatial extent of the neighboring ganglion cell RFs and/or does this involve spatial mechanisms extending beyond neighboring RGC receptive fields?

Yes, indeed, correlations extend to distances up to 800 micrometers and beyond. (And these are significant in the sense that the mean correlations are separated from zero by more than the 95% confidence interval.) Note, though, that the stimulus also contains considerable correlations at these distances (black lines in Fig. 1F). As we explore further in subsequent parts of the manuscript, the neuronal response correlations arise from an interaction of nonlinear spatial integration by the cells (acting within the receptive field) and the spatiotemporal structure of the natural stimulus (leading to joint activation of the nonlinear receptive fields even across large distances). Thus, no neuronal mechanisms beyond the spatial scale of receptive fields are required, as also corroborated by the fact that we can reproduce the correlation values by our subunit grid model (see Figure 4H). We now clarify this point by explicitly pointing out the long-range correlations (lines 59-68) and relating them later to the proposed mechanism (lines 372-376).

P4: It would be good to briefly justify the selection of ganglion cell types selected for this study for a general reader and particularly introduce the similarity of ON and OFF parasols vs. mouse ON and OFF sustained alpha cells. This is particularly interesting taken into account how both ON parasols and mouse ON-sustained alpha RGCs show strong correlations and have been previously considered to be homologous cell types.

Thanks for the suggestions. We have added corresponding explanations of the cell selection (lines 51-54 for marmoset and lines 179-182 for mouse) and now also discuss the potential homology between the primate and mouse ganglion cell types (lines 182-184).

P4, line 110: “..On-alpha as well as transient-OFF-alpha”: Does this mean On-sustained alpha cells in this case?

Yes, what we wrote was not clear. Now corrected (lines 186-188).

Fig 2E: The mosaics of ON parasols are more complete in the example as for the OFF parasols. What is the implication for this on the deduced results?

Mosaics are incomplete owing to cells missed by the multielectrode arrays. There is likely a recording bias that may lead to an unequal sampling of ON and OFF parasol cells (as well as other cell types). For the present analysis, however, we do not require complete mosaics, as our analysis focuses on cell pairs, and even with incomplete mosaics, there are sufficient samples of pairs to explore correlations and redundancy at different spatial arrangements. This is now explained in the Methods section (lines 766-772).

Fig 3: How well is the spatial extent of the subunits in line with the previous findings and/or morphological assumptions of the bipolar cell RFs?

For the peripheral marmoset retina, the evidence mainly comes from anatomy. Dendritic fields have been reported as 15-20 μm in diameter for flat midget bipolar cells (Chan et al., 2001, Telkes et al. 2008), the inputs to OFF midget cells, and can be estimated to be around 30 μm for type-3 diffuse bipolar cells (Chan et al 2001), the inputs to OFF parasol cells, approximately in line with our obtained subunit sizes of around 20 and 30-40 μm , respectively (Fig. S7I). For ON midget cells, however, subunits are found to be surprisingly large (~50 μm), larger than midget bipolar cell dendritic trees. We speculate that this may reflect groups of midget bipolar cells or input from type-6 diffuse bipolar cells, which have dendritic fields of 40-80 μm in the marmoset peripheral retina (Chan et al 2001) and which also connect

to ON midget cells (Tsukamoto and Omi, 2016). In the mouse retina, our estimates of 40-65 μm for alpha cell subunit diameters match previous physiological measurements of bipolar cell receptive fields with calcium imaging (50-70 μm ; Franke et al, 2017; Strauss et al 2022) or with electrophysiology from the type 6 ON bipolar cell (40-50 μm ; Schwartz et al 2012). The comparisons of subunit sizes and bipolar cell dendritic fields are now included in the text (lines 241-250) and in the legend of Fig. S8.

p 15, line 489: Specify the wavelength of the IR light.

During the dissection, the samples were illuminated by LED lamps with peak intensity at 850 nm. We have now included this information in the Methods section (line 685).

p 16, Visual Stimulation: What was the overall level of visual pigment bleaching (maximum level) in these recording conditions taken into account the recording duration. Were there any signs of light adaptation and/or how was this controlled in these recordings?

In the marmoset retina, we controlled the level of tissue quality and visual pigment bleaching by periodically measuring contrast sensitivity of retinal ganglion cells (now presented in the new Fig. S13). Contrast sensitivity at the beginning of the recording slowly decreased after three hours, but stayed reasonably high throughout the presentation of natural and artificial stimuli used in this study. The mouse retina was also minimally affected during the presentation of relevant stimuli, given that we used similar light levels as in the marmoset and animals were dark-adapted before the isolation of the retinal tissue. We confirmed the recording stability by stimulating the retina with full-field white noise about every 1-2 hours. We compared temporal filters and static nonlinearities and mostly found only small changes of filter latency (~ 10 ms) in some retinal pieces within the first three to four hours of the recording. Furthermore, in both species, we found stable responses of ganglion cells to repeated segments of natural movies, indicating that there was no rapid decline of sensitivity or recording quality.

p 17: How complete were the obtained cell-type specific mosaics? How does the completeness of the mosaic impact the results particularly related to correlation analyses. Main Figs of the paper seem to indicate differences in mosaic completeness across cell types.

As also mentioned above, differences in mosaic completeness likely arise from recording biases in the extracellular multielectrode-array recordings. However, our analyses do not require complete mosaics. The correlation and redundancy analyses rely on cell pairs, and for all analyzed cell types, we have a large number of samples of cell pairs at different distances from each other. We now clarify this in the Methods section (lines 766-772).

Referee #3:

Major concerns:

1) *Significant discrepancies in findings relative to previous studies.*

The subunit model used to find the ganglion cell responses in marmoset found that On parasol and midget cell subunits were more rectified (i.e., nonlinear) than Off parasol/midget cells at photopic backgrounds (Fig. 2). Yet, several studies in primate retina, including MEA work and direct measurements of the subunit nonlinearities using voltage clamp recordings have repeatedly found the opposite to be the case – at these light levels, Off parasol/midget cell subunits (bipolar cells) are significantly more rectified than their On counterparts (Chichilnisky and Kalmar, 2001; Turner and Rieke; Manookin; Appleby; Sinha). Further, this finding has been replicated across several vertebrate and

invertebrate species, and normative studies have provided a theoretical explanation for why this On/Off asymmetry ought to be the case across taxonomic groups.

– In lines 156-159, the authors point out this discrepancy, but no explanation is given for why the current study deviates so significantly from previous work.

We were also originally puzzled by this finding, but found this to be a robust experimental result, consistent over retinas as well as over stimuli (flashed gratings, as seen in the subunit nonlinearities, Fig. 3; reversing gratings, as seen in the comparison of F2 frequency components, Fig. S10; natural movies and spatiotemporal white noise, as seen in the relative performance of models with linear receptive fields, Fig. 4). Given that there is no prior data of subunit nonlinearities in the marmoset, we believe that it is feasible that there is a species difference between the previously studied macaque and the marmoset. This might reflect different behavioral demands and viewing strategies as previously discussed (Mitchell and Leopold 2015). As an example that differences between macaque and marmoset do indeed exist in the relative composition of ON and OFF pathways, one can consider the relative number of retinal cells stained for choline acetyltransferase (ChAT) in the ON versus OFF ChAT bands. While there are considerably more ON ChAT cells than OFF in the macaque retina (with a ratio of about 70:30), this is reversed in the marmoset with 30:70 fewer ON than OFF ChAT cells (Grünert and Martin, 2020).

Note also that differences in spatial nonlinearities between ON and OFF channels across species are not as systematic and theoretically grounded as one might think (and as they are, for example, in the case of the size of receptive fields, where a good normative basis has been established, Ratliff et al. 2010). Among the different types of alpha ganglion cells in the mouse retina, ON cells show nonlinear responses under reversing gratings at least as strong as OFF cells (Krieger et al. 2017). Furthermore, the subunit nonlinearity represents a major contribution to a ganglion cell's contrast-response function, yet these are not systematically more nonlinear for OFF cells. For example, the asymmetry of nonlinearities in contrast-response functions in the rat retina has been found to be cell-type-specific with some types showing more nonlinear responses for ON cells than for their OFF counterparts (Ravi et al 2018), and in the degu retina, ON cells seem rather more nonlinear than OFF cells (Escobar et al 2018).

For the macaque retina, we further note that the spatial nonlinearities of ON and OFF parasol cells are complex and context-dependent. Nonlinearities in OFF parasol cells, for example, are attenuated by stimulation of the receptive field surround (Turner and Rieke, 2018), and recent work from the Rieke lab (Hong and Rieke, abstract at European Retina Meeting 2023) suggests that the opposite may be the case for ON parasols, whose nonlinear receptive field surround can enhance nonlinear response properties rather than suppressing them. This means, for example, that measurements of spatial nonlinearities obtained with localized stimuli inside receptive-field centers cannot easily be generalized to stimuli that span larger regions, as is the case for the natural movies applied in our work. The stimulus-dependent variability of nonlinear integration can even be observed on an image-by-image basis. For macaque ON parasol cells, linear spatial integration is observed for some natural scenes, while others evoke clear nonlinear responses (Freedland and Rieke 2022). Similarly, receptive field models with linear spatial integration fail to predict natural movie responses for both ON and OFF parasol cells, often with worse performance for ON parasol cells (Heitman et al 2016). The context dependence of spatial nonlinearities may also be tied to the observation that nonlinearities are sensitive to the excitation/inhibition balance in the inner plexiform layer and that small parameter changes may shift this balance and change the spatial-contrast sensitivity of the retinal output (Yu et al 2022), which could be a simple mechanistic underpinning for species-specific differences.

In the revised manuscript, we now discuss the difference in the relative nonlinearity of ON and OFF parasol cells between our findings and those in the macaque in more detail (lines 406-416). Note also, that for our primary results, it is not important whether ON or OFF parasols show stronger nonlinearities. In our data, both cell types are highly nonlinear and strongly correlated and redundant in their pairwise responses.

– *In Fig. 2, the On midget subunit size is bigger than Off parasol subunits. However, this does not agree with published measurements of bipolar cells in the mid-peripheral/peripheral primate retina. Anatomical studies have shown that, at any given eccentricity, diffuse bipolar dendritic trees are significantly larger than midget bipolar dendritic trees (Boycott and Wassle, 1991; Dacey et al., 2000; Tsukamoto and Omi, 2015, 2016). This discrepancy likely indicates that the subunit model fit in the current study poorly estimates the subunit properties, including their size and nonlinear properties (see above).*

We agree that the large subunits of ON midget cells are surprising and unlikely to represent the layout of individual midget bipolar cells. This may reflect the fact that the subunit layout is not well restricted by the data and by the fitting procedure. Alternatively, subunits could correspond to groups of bipolar cells rather than individual ones. Furthermore, ON midget cells can receive inputs not only from midget bipolar cells, but also from type-6 diffuse bipolar cells (DB6; Tsukamoto and Omi, 2016), which have relatively large dendritic fields (40-80 μm ; Chan et al 2001). Thus, one can speculate that the large subunits of ON midget cells could also be governed by their DB6 inputs. To clarify these points, we have added a discussion of the comparison of subunit sizes and bipolar cell dendritic fields (lines 241-250).

Note, though, that the goal of our model was not to identify subunits as bipolar cell layouts, but to identify good subunit-based models to capture nonlinear spatial integration, which the models do, as demonstrated by several comparisons (Fig. 3K,L; Fig. 4E-G; Fig. S8J; Fig. S10G; Fig. S11H). This forms the basis for the use of the models to analyze the effects of nonlinearities on correlations under natural stimuli (Fig. 5), the actual purpose of the models here. We note that, for model performance, the actual subunit layout seems not to be critical, as also observed elsewhere (e.g., Freedland and Rieke 2022). Other parameters of the model, such as subunit nonlinearities, are more important and much better constrained in the model fits. For example, unlike the subunit layouts, they do not vary much with changes in regularization parameters (Fig. S7). We now discuss these aspects in the revised manuscript (lines 257-271).

– *Lines 252:255: The current work indicates that spatial contrast dependence was largest in On parasol cells, but previous work has shown that Off parasols have the greatest sensitivity to spatial contrast both to artificial and naturalistic inputs (Turner and Rieke, 2016; Turner et al., 2018; Appleby and Manookin, 2020).*

See also our response to the comment above, regarding the question about discrepancies to other studies. Briefly, we see the stronger spatial-contrast sensitivity of ON parasols as a robust finding and speculate that this reflects a species difference between macaque and marmoset. Furthermore, most previous measurements of spatial nonlinearities in macaque have focused on the receptive-field center, but the receptive-field surround can differentially affect nonlinearities in macaque ON and OFF parasols (Turner and Rieke 2018; Hong and Rieke, abstract from European Retina Meeting 2023). And finally, for the purpose of the present study, both parasol types show substantial spatial nonlinearities as well as correlations and redundancy, supporting our key finding of compromised efficient coding, regardless of which type displays the stronger spatial nonlinearities.

This discrepancy with previous studies could arise from at least two sources: 1) Imbalances in the activity in the On and Off pathways due to issues with tissue health and/or 2) excessive free parameters

in the subunit model that cause convergence on unrealistic values. Tracking this down will be critical to the overall impact of the study.

We do not believe that issues with tissue health were a major factor in potential discrepancies with previous studies. We have applied strict criteria for recording quality by requiring sensitivity to low-contrast light-intensity modulation, tested with 4-Hz sinusoidal modulations and contrast levels from 20% down to 2.5%, similar to the monitoring of recording quality by, for example, the Rieke lab and others. To demonstrate the quality of our recordings, we now show this data in a new supplementary figure (Fig. S13; see also lines 669-671), revealing that parasol cells generally responded reliably to 5% and often even 2.5% contrast and that this sensitivity was maintained over the course of several hours.

Regarding the potential source of excessive parameters in the models, we note that the strong nonlinearity of ON parasol cells were also directly observed in the responses to reversing gratings (Fig. S10A-B), independent of the model analysis. Furthermore, the models yield good model fits for held-out data, thus giving no direct sign of overfitting. We therefore believe that it is likely that the stronger spatial nonlinearity in ON parasol cells is true for the marmoset under the applied stimulus conditions.

2) Acknowledgment of previous work/larger context of the current study

This finding that the retinal output is strongly correlated/redundant has been shown previously in several studies including in primates (Schnitzer and Meister, 2003; Schneidman et al., 2006 Shlens et al., 2009; Trong and Rieke, 2008; Ala Laurila et al., 2011; and others...). Further, the previous literature has identified several functional roles for such correlations. The current study would benefit greatly from properly acknowledging this previous work and explaining how it contributes to our understanding of these correlations in visual processing.

Thanks for pointing this out. As suggested, we have extended the Discussion to relate our findings to previous analyses of correlated activity (e.g., lines 377-388). Note, though, that many of these earlier studies typically targeted other questions (e.g., noise correlations) and mostly used spontaneous activity or artificial white-noise stimulation. We do point out some earlier work on correlations under natural movies in the salamander retina (lines 384-386) and now refer to prior functional hypotheses regarding correlated ganglion cell activity (lines 417-421).

The observation that On and Off parasol cells are correlated with each other to natural inputs is an interesting and important contribution to the field. However, as mentioned above, how does this finding fit into the larger picture? The current work intimates broadly at the functional consequences for this, but having this spelled out more explicitly and placed in the general context of our current understanding would be very helpful.

This is an interesting question about which we can only speculate at this point. We have extended the discussion around the correlation of ON and OFF parasol cells and now more clearly draw analogies with correlations observed in other cell types of differential tuning. In particular, we speculate that the signals of ON and OFF parasol cells could be jointly used to disentangle luminance and spatial contrast information (lines 432-440).

Author Rebuttals to First Revision:

We thank the Reviewers and the Editor for their thorough and constructive comments. We have addressed all aspects raised by the Reviewers as detailed in the point-by-point responses below. In particular, we have revised the exposition regarding the relation between decorrelation, redundancy reduction, and efficient coding and have clarified the influence of potential assumptions, such as linear receptive fields and noise level. Furthermore, we have extended the Discussion to consider scenarios of joint encoding by multiple cell types. Through these changes, we also provide important contextualization and acknowledgment of alternative hypotheses, as suggested in the editorial response. We believe that the revision of the presented material have substantially clarified the findings of our manuscript and their interpretation. The changes to the text have been highlighted in the manuscript.

Referee #1:

The authors have clearly taken the initial round of reviews seriously and the manuscript has improved substantially. However, I still think there are ambiguities in interpretation and I ultimately do not think the authors have demonstrated the main conclusions of the paper: nonlinear processing counteracts efficient coding. The continuing disagreement here is about the extent to which correlations (or lack of decorrelation) indicates violations of efficient coding theory. The authors claim it does, but I disagree.

Thank you for your feedback. We take this as an opportunity to further clarify, as detailed below, how the observed correlations provide a challenge for efficient coding, what assumptions underlie this, and, as suggested by the Reviewer, what alternative hypotheses and implications might be of consequence. In particular, we have aimed at better explaining that the measured correlations truly challenge the efficient coding hypothesis in the low-noise scenario and that there is no assumption of a specific encoding model underlying this argument.

The Authors state on line 98 "Decorrelation is a strong prediction of the efficient coding hypothesis in the case of noiseless transmission channels (Atick and Redlich, 1992; Barlow, 1961)." In my view, this statement is incomplete. The decorrelation prediction also assumes an approximately linear encoding model (e.g., an LNP model). This is actually clearly articulated in the Atick and Redlich (1992) study that the authors cite, see just before and after equation 1.2 in that manuscript. Thus, a different encoding model can make different predictions with respect to decorrelations. For example, one difference in the predictions made by the Karklin and Simoncelli model from that of the Atick and Redlich model is how receptive field structure depends on noise. In a purely linear model optimized to efficient coding (Atick and Redlich), changing the noise causes a change in receptive field structure (center-only under high noise and center-surround under low noise). However, when one introduces a rectifying nonlinearity and then optimizes according the efficient coding, the prediction changes. The surround doesn't go away at low light levels (high noise conditions); instead the center AND surround both increase their diameters so that the population of cells have greater overlap. The two models (optimized according to efficient coding theory) thus make different predictions about the extent and manner of decorrelation. This illustrates the point that the specific encoding model under consideration can have a major impact on what efficient coding theory predicts.

The authors are implicitly assuming a pseudo-linear encoding model when comparing retinal responses to efficient coding predictions. Prior work shows a pseudo-linear model holds better, in general, for midget ganglion cells than for parasol ganglion cells. And the authors observe better agreement with the Atick and Redlich model for midget cells (more linear) than for parasol cells (less linear). In my view, this pretty strongly suggests this is a flaw in the model, not a flaw in the theory that is optimizing the model.

Thank you for your insightful comments and clear exposition. This allows us to better explain the underlying assumptions of our analyses. Most importantly, there is no assumption of a pseudo-linear

encoding model in our work (or any other specific encoding model). This is apparently a misunderstanding, which may have originated in our use of the Atick and Redlich work as a starting point and their heavy reliance on a pseudo-linear model for deriving center-surround receptive fields. Our work, however, is not concerned with receptive field shape or with a similar model-based prediction of retinal processing features. Instead, decorrelation and redundancy reduction are model-free predictions of efficient coding, at least in the case of noiseless transmission channels. That decorrelation follows from efficient coding in the noiseless case is simply a consequence of the fact that any dependencies in the output of a communication channel reduce the output entropy compared to the channel's capacity. This is the scenario of Barlow's original hypothesis (Barlow 1961) and also evident from the 1992 Atick and Redlich paper, see their section 2.1 (and maybe laid out more clearly in Atick's review in Network from 1992, which we now cite), whereas the text around equation 1.2 in that paper deals with the specific model this is used for deriving receptive field structure. To clarify this aspect in our manuscript, we have revised the statement pointed out by the Reviewer to better explain that decorrelation is a direct consequence of efficient coding in the noiseless case (lines 95-101) and later discuss more clearly how the noise present in retinal activity affects our analysis (lines 385-392). We also now point out explicitly in the Discussion that the analysis of decorrelation and redundancy reduction is independent of whether receptive fields are linear or nonlinear (lines 414-415).

Here is an illustration of the point I'm trying to make. The Atick and Redlich work doesn't predict the presence of ON and OFF cells. So one could take their work and compare it to the retina and say that efficient coding theory is wrong because the theory doesn't predict ON and OFF cells. But then Karklin and Simoncelli come around and accounted for the rectifying nonlinearity in spike generation in a linear-nonlinear model and then optimized that new model according to the theory. Suddenly, optimizing a more accurate model of retinal encoding produces ON and OFF cells, mosaics, the slight numerical dominance of OFF cells over ON cells, and a few other features of retinal processing. But of course, it doesn't predict (yet) midget and parasol cells, or color opponent cells. So, is efficient coding theory wrong or right? Given the work on elaborating encoding models and the stimulus sets used to train those models (Karklin and Simoncelli 2011; Ocko, Lindsey, Ganguli, and Deny, 2018; Jun, Field and Pearson, 2022), it seems much more likely that the divergence of the results in this study from prior efficient coding predictions has more to do with overly interpreting 'decorrelation' as a specific signature of efficient coding theory, when in fact it is just a prediction from efficient coding theory under a linear model that we already know is wrong.

Thus, I continue to view this study as not challenging the idea of efficient coding theory, but a case where the authors are exploring conditions under which these nonlinearities of parasol cell signaling are amplified — i.e. greater deviations from the predictions of an LNP model. I suspect one major reason for this is the known subunits present in the receptive fields of parasol cells. These subunits are not accounted for in Atick and Redlich or the Karklin and Simoncelli models, hence these models cannot be trusted to make accurate predictions about how the parasol cells should respond to these stimuli nor do they make accurate prediction with respect to efficient coding.

Thank you for clarifying and elaborating on this aspect. However, as already explained above, our approach is different from the model-based optimization approaches discussed here by the Reviewer. Our analyses of decorrelation and redundancy do not rely on an underlying model and therefore make no assumption about linear integration or signal rectification. Thus, deviations from efficient coding found in our data cannot be explained by unaccounted model components, such as subunits. We turn to specific models only after having established the lack of decorrelation and redundancy reduction in our model-free analyses and then use the models not to derive predictions that would follow from efficient coding, but to investigate the origins of the correlations. We then find, as the Reviewer suspected, that subunits are a crucial component, together with the natural stimulus statistics for

generating correlations (but not for the connection of correlations to efficient coding, as explained above). Yet, for normative approaches that aim at deriving model-based predictions from efficient coding, complementary to our work, this means that including subunits would be a worthwhile challenge, as mentioned in the Discussion (lines 423-429). However, we do understand that our interpretation of decorrelation as a specific signature of efficient coding may not have been clear enough, in particular with respect to the role of noise. We have therefore aimed throughout the text to better differentiate between our conclusions regarding decorrelation and redundancy reduction and the consequences for efficient coding (e.g., lines 95-107, lines 385-398, lines 423-429., as already stated above).

The authors also create some straw-man models in their manuscript, that I think are misleading. For example, on line 67, "Furthermore, when considering pairs of either parasol or midget cells with opposing contrast preference, that is, an ON and an OFF cell, one might expect these pairs to be negatively correlated, as one cell should respond to increases and the other to decreases in light intensity." Again, this statement assumes a linear model. If the ON and OFF cells do not exhibit subunit rectification, then the above statement is true. But we know parasol cells exhibit subunit rectification and so do peripheral midget RGCs to a lesser extent. Thus, anytime a natural image with high spatial frequency content (e.g., a high contrast edge) falls in the receptive field of overlapping ON and OFF parasol cells, they will both fire action potentials because their encoding is not linear. This has been known since the 1970s. None of this is a violation of efficient coding theory per se, but it is a violation of linear models.

We apologize for this misunderstanding. The statement regarding the expected negative correlation between ON and OFF parasol cells was not meant as a hypothesis related to efficient coding, and the observed positive correlation was not used as evidence for or against efficient coding or redundancy reduction in our manuscript. Rather, we do believe that this is an insightful and unexpected observation that helps underscore the general correlation-boosting effect of nonlinear receptive fields in the context of natural stimuli. We have correspondingly revised the statement to make this aspect clearer and to put less emphasis on the potential expectation of negative correlations (lines 69-73).

It is also worth mentioning that, although the positive correlations between ON and OFF parasol cells can be understood based on the nonlinear receptive fields, as pointed out by the Reviewer, the correlations might still be counterintuitive, at least for a more general audience. They demonstrate that the nonlinear response properties are not only present, but have, under natural stimuli, the stronger effect on the response correlations as compared to the cells' linear response characteristics that give them their name as ON and OFF cells, respectively. And while we agree that the nonlinear response characteristics have been known for decades, we are not aware of any prior evidence that this leads to positive response correlations between ON and OFF cells under natural stimuli.

The authors also added an explicit computation of information rates and redundancy (Fig S3 and Figs 1I-J). I liked this analysis, but I still think it suffers from some issues. There is pretty clearly redundancy across cells, as the authors indicate. The question is, 'does this violate efficient coding theory?' It isn't actually clear to me if it does or doesn't in this case for a couple of reasons. First, efficient coding theory isn't just about redundancy reduction, as the authors state, it is about maximizing information transmission too, in the face of noise. Thus, the question becomes, 'could the cumulative information rates across all cell types be retained if the redundancies were reduced?' Clearly, in a noiseless system, the answer would be yes, but when noise is spread across the system and common to multiple channels, it is less clear. Second, the illustration by the authors that ON parasol cells are the most reliable and yet still exhibit high levels of redundancy has the issue that the ON parasol cells are not acting independently of the other cell types in the system and thus perhaps redundancy in their signals is helping to compensate for noise in OFF parasol or midget cell signals. Admittedly, these are very difficult

issues to untangle, but given these difficulties in interpretation, I think the manuscript comes down too hard on the side against efficient coding.

We agree that considering larger populations, potentially containing multiple cell types and the effect of noise is an important point. We have addressed this first by expanding our analyses of the reliability of the cells' responses to the presented natural stimuli: We now show that also for mice, the more redundant cell types are also the more reliable and thus less noisy ones (see new figure panel S3H). This indicates, like for the marmoset retina, that a role of correlations for efficient encoding by averaging out noisy responses is unlikely, at least when considering single-cell-type encoding. Furthermore, we now present an analysis of Fano factors (see new figure panels S2E and S3G), which for all considered cell types lie around 0.1 or below when computed for the relevant time scale of individual fixations. This underscores the high reliability of the cells' responses, consistent with encoding in a low-noise regime. Second, we take up the idea that correlations among, for example, ON parasol cells could serve to counteract noise in ON midget cells. This is a scenario that we cannot exclude, as experimentally assessing efficient information transmission in large populations is difficult to fully answer and likely beyond our current methodology. Yet, whether, for example, ON parasol and midget cells should be considered part of a joint information channel is also questionable, given that their downstream information pathways in the brain are to a large degree separated and that they substantially differ in their signal conduction velocities. Nonetheless, to acknowledge and discuss the possibility that redundancies in one cell type may help efficient encoding in conjunction with other cell types in the presence of noise, we have extended the Discussion accordingly (lines 392-398).

A more conservative interpretation of these results (that I think is also more accurate) is that this study demonstrates some combination of the following — 1) the retina is not well described by pseudo-linear models; 2.) redundancy reduction does not appear to be the primary goal of retinal signaling, and 3) observation 2 could be because efficient coding theory is inadequate to describe retinal coding OR because sufficiently accurate models of retinal processing (e.g., models that include subunit rectification) that also include the magnitude and sources of retinal noise (e.g., input, output, and correlated noise), have not been optimized according efficient coding theory. Thus, we don't have a strong comparison between the predictions of efficient coding theory when applied to circuits of similar structure to the retina and the actual signaling performed by the retina. This tension is demonstrated in the title of the manuscript — paraphrasing: "Correlated nonlinear responses counteract efficient coding." But this apparent "counteracting" is (in my view) more likely to be a result of exploring efficient coding in terms of linear instead of nonlinear models.

In summary, I think this is a high quality study, but it is not entirely decisive in what can be concluded. This, to some extent, diminishes the impact. The work is also of sufficient complexity, that it would likely benefit from some more space. Reading it at times feels like topics are being oversimplified to manage the space constraints.

Thank you for these suggestions. They have helped us refine our line of argumentation and interpretation. We have correspondingly aimed at clarifying our interpretation throughout the manuscript and at acknowledging alternative hypotheses, as pointed out in the responses above. Let us re-iterate that there is no assumption of pseudo-linear models in our assessment of redundancy reduction and its connection to efficient coding and that our approach is not based on optimizing a particular model structure. Given the model-free, straightforward assessment of decorrelation and redundancy reduction and their connection to efficient coding, we also do not believe that our presentation oversimplifies the topic. Thus, our interpretation might be summarized as the following — 1) redundancy reduction does not appear to be the primary goal of retinal signaling, in particular when considering certain cell types; 2) the redundancy in these cell types arises from nonlinear receptive fields, which lead to highly correlated activity in response to natural gaze shifts; and 3) given the low-

noise conditions, evident by the high response reliability in particular of nonlinear cell types, the lack of redundancy reduction appears inconsistent with efficient coding; yet, we cannot exclude that a full evaluation of information by joint encoding of multiple cell types that includes a detailed assessment of (correlated) input noise may modify this picture. Our revisions have aimed at stating these conclusions and the underlying arguments more clearly. In line with these refinements, we have changed the title of our manuscript to “Redundant retinal coding from nonlinear receptive fields during natural gaze shifts” to better reflect our key findings rather than the interpretation of the results.

Referee #2:

The authors have addressed sufficiently all my previous concerns. The already technically sound and interesting manuscript has now gained significant additional presentation power and depth. Overall, the manuscript has improved significantly in revision particularly related to its framing. In my view, the authors have done an excellent job in improving the presentation of the paper in relation to the efficient coding theory and in bringing up the novelty aspects of their study in reference to the previous literature. In addition, important improvements have been made in addressing the possibility to assess the tissue condition as well as e.g. more clearly relating the spatial scale of the modeled subunits to the morphological dimensions of the cone bipolar subunits. Particularly useful aspect of the study is also the comparison of the findings on the primate retina to those on the mouse retina, as a model system. I have no additional comments except of a minor note related to the methods section. It would be important to state clearly the assumptions underlying the conversion of light intensities into isomerization rates (i.e. state clearly the assumed parameter values, particularly the photoreceptor collecting area parameters) such that the stimulus conditions can be reproduced in the future studies.

Thank you for your positive feedback and the suggestion regarding the estimation of isomerization rates. We have now included details in the Methods section (lines 756-762) that summarize how we performed the isomerization rate calculations and that, in particular, state the applied collecting areas and peak sensitivity values.

Referee #3:

The authors have adequately addressed my previous concerns. This is a very nice study.

Thank you for your assessing our manuscript and for your positive feedback.